# Homeostatic plasticity fails at the intersection of autism-gene mutations and a novel class of common genetic modifiers

Özgür Genç[1], Joon-Yong An[2,3], Richard D Fetter[1], Yelena Kulik[1], Giulia Zunino[1], Stephan J Sanders[2], Graeme W Davis[1]*

[1]Department of Biochemistry and Biophysics Kavli Institute for Fundamental Neuroscience University of California, San Francisco, San Francisco, United States; [2]Department of Psychiatry UCSF Weill Institute for Neurosciences University of California, San Francisco, San Francisco, United States; [3]School of Biosystem and Biomedical Science, College of Health Science, Korea University, Seoul, Republic of Korea

**Abstract** We identify a set of common phenotypic modifiers that interact with five independent autism gene orthologs (*RIMS1*, *CHD8*, *CHD2*, *WDFY3*, *ASH1L*) causing a common failure of presynaptic homeostatic plasticity (PHP) in *Drosophila*. Heterozygous null mutations in each autism gene are demonstrated to have normal baseline neurotransmission and PHP. However, PHP is sensitized and rendered prone to failure. A subsequent electrophysiology-based genetic screen identifies the first known heterozygous mutations that commonly genetically interact with multiple ASD gene orthologs, causing PHP to fail. Two phenotypic modifiers identified in the screen, *PDPK1* and *PPP2R5D,* are characterized. Finally, transcriptomic, ultrastructural and electrophysiological analyses define one mechanism by which PHP fails; an unexpected, maladaptive up-regulation of *CREG*, a conserved, neuronally expressed, stress response gene and a novel repressor of PHP. Thus, we define a novel genetic landscape by which diverse, unrelated autism risk genes may converge to commonly affect the robustness of synaptic transmission.

*For correspondence:
graeme.davis@ucsf.edu

## Introduction

Autism Spectrum Disorder (ASD) is a polygenic disorder with a complex underlying genetic etiology (*Bourgeron, 2015*). Advances in whole genome sequencing and genome-wide association studies have dramatically expanded our understanding of the genetic architecture of ASD. In particular, the identification of rare de novo mutations that confer high risk for ASD has generated new molecular insight (*De Rubeis et al., 2014*; *Iossifov et al., 2014*; *Sanders et al., 2015*). Yet, even in cases where a rare de novo mutation confers risk for ASD, additional processes are likely to contribute to the ASD phenotype (*Leppa et al., 2016*; *Peter et al., 2019*) including the engagement of adaptive physiological mechanisms (*Gaugler et al., 2014*; *Gibson, 2009*; *Hartman et al., 2001*; *Hou et al., 2019*; *Kitano, 2007*; *Plomp et al., 1992*; *Sackton and Hartl, 2016*; *Sardi and Gasch, 2018*; *Bourgeron, 2015*).

Homeostatic plasticity, in particular, has garnered considerable attention as an adaptive physiological process that might be relevant to the phenotypic penetrance of ASD mutations (*Antoine et al., 2019*; *Bourgeron, 2015*; *Mullins et al., 2016*; *Nelson and Valakh, 2015*; *Ramocki and Zoghbi, 2008*). Yet, very little is known at a mechanistic level regarding the interface of homeostatic plasticity and ASD genetics. There remains ongoing debate regarding whether

homeostatic plasticity is normally induced or whether it is impaired in the context of rare de novo mutations that confer risk for ASD (*Antoine et al., 2019*; *Bourgeron, 2015*; *Ramocki and Zoghbi, 2008*). And, there is no mechanistic information regarding how rare de novo mutations that confer risk for ASD might be connected to the signaling mechanisms that are essential for the induction and expression of homeostatic plasticity.

It is well established that homeostatic signaling systems function throughout the central and peripheral nervous systems to stabilize neural function following a perturbation that can be of genetic, immunological, pharmacological or environmental origin (*Davis, 2006*; *Marder, 2011*; *Turrigiano, 2011*). Evidence for this has accumulated by measuring how nerve and muscle respond to the persistent disruption of synaptic transmission, ion channel function or neuronal firing. In systems ranging from *Drosophila* to human, cells have been shown to restore baseline function in the continued presence of these perturbations by rebalancing ion channel expression, modifying neurotransmitter receptor trafficking and modulating neurotransmitter release (*Davis, 2013*; *Hengen et al., 2013*; *Maffei and Fontanini, 2009*; *Watt and Desai, 2010*). There is evidence that homeostatic signaling systems function at the level of individual cells and synapses (*Davis, 2013*). There is also evidence that homeostatic signaling systems influence the function of neural circuitry (*Deeg and Aizenman, 2011*; *Hengen et al., 2013*; *Maffei and Fontanini, 2009*; *Nelson and Valakh, 2015*).

We set out to determine whether there exists a molecular interface between mutations in ASD gene orthologs in *Drosophila* and the induction or expression of presynaptic homeostatic plasticity. Presynaptic homeostatic plasticity (PHP) is an evolutionarily conserved form of homeostatic plasticity, observed in *Drosophila*, mice and humans (*Davis, 2013*). PHP has been documented at both central and peripheral synapses in response to differences in target innervation (*Liu and Tsien, 1995*) altered postsynaptic excitability (*Davis, 2006*; *Marder and Goaillard, 2006*; *Mullins et al., 2016*), following chronic inhibition of neural activity (*Kim and Ryan, 2010*; *Zhao et al., 2011*) and following disruption of postsynaptic neurotransmitter receptor function (*Henry et al., 2012*; *Jakawich et al., 2010*). The mechanisms of PHP have a remarkable ability to modulate and stabilize synaptic transmission, with an effect size that can exceed 200% (*Müller and Davis, 2012*; *Ortega et al., 2018*).

Many of the rare de novo mutations that confer high risk for ASD are considered to be heterozygous loss of function (LOF) mutations (*Bourgeron, 2015*; *De Rubeis et al., 2014*; *Iossifov et al., 2014*; *Sanders et al., 2015*). Therefore, we examine the phenotype of heterozygous LOF mutations in five different ASD gene orthologs. We make several fundamental advances. First, we demonstrate that these individual heterozygous LOF mutations have no overt effect on baseline transmission or PHP. However, we demonstrate that PHP is sensitized to failure. Next, we sought to define the molecular mechanisms that connect ASD gene orthologs to the mechanisms of PHP. A genome-scale screen and subsequent systems-genetic analyses yielded unexpected insight. We do not simply identify genes that, when mutated, enhance the phenotype of individual ASD gene mutations. We discovered genes that, when their function is diminished by heterozygous LOF mutations, *commonly modify* multiple ASD gene orthologs, causing a selective failure of homeostatic plasticity. Thus, we define the first class of common phenotypic modifiers of ASD genes in any system. Finally, we do not stop with the identification of a novel class of ASD gene modifiers. We proceed to characterize *how* homeostatic plasticity fails in one such condition. The mechanism we discovered is also unexpected and illuminates the complexity by which double heterozygous gene-gene interactions can generate a cellular or organismal phenotype. We demonstrate maladaptive, enhanced expression of a gene known as *Cellular Repressor of E1A Stimulated Genes (CREG)*, a gene that is conserved from *Drosophila* to human and expressed in the brain (*Yang et al., 2011*).

Taken together, we define a novel, unexpected genetic architecture that connects heterozygous LOF mutations in ASD-associated gene orthologs with the mechanisms of homeostatic plasticity. In particular, the observation that PHP is commonly sensitized by multiple, different ASD genes, and the fact that we identify and characterize common phenotypic modifiers of five different ASD genes, defines a novel means by which a diversity of ASD-associated risk genes may converge to affect synaptic transmission. We propose that this information may be relevant to new therapeutic approaches that might someday modify ASD phenotypic severity, regardless of the underlying genetic mutation (s) that confer risk for ASD.

## Results

We began an investigation of ASD gene orthologs in *Drosophila* by acquiring heterozygous null mutations in five genes; *RIMS1*, *CHD8*, *CHD2*, *WDFY3* and *ASH1L* (*Figure 1A*; Note: throughout we use the human nomenclature). Heterozygous null mutations were analyzed, as opposed to use of RNAi-mediated gene knockdown, in order to more precisely reflect the proposed genetic perturbations in human.

All five of these genes are considered high confidence 'category 1' ASD-associated genes based on SFARI Gene (*Simons Simplex Collection, 2020*). All five of these genes have clear *Drosophila* orthologs. Further, we demonstrate that all five genes are expressed in *Drosophila* third instar motoneurons based on a Patch-Seq analysis of gene expression (*Figure 1—figure supplement 1*). The five ASD gene orthologs were also chosen to reflect a broad range of biological activities that are associated with the numerous ASD-associated genes identified to date. The *RIMS1* gene is a synaptic scaffolding protein that localizes to and organizes sites of neurotransmitter release, termed active zones. The *CHD8* and *CHD2* genes encode chromatin remodeling factors that localize to the cell nucleus. *WDFY3* encodes a phosphatidylinositol 3-phosphate-binding protein and regulator of autophagy and intracellular signaling. *ASH1L* encodes a member of the trithorax group of transcriptional activators and is found in the cell nucleus. A supplemental table (*Supplementary file 1,* Supplemental Table S1) includes known disease associations for each of these five human genes, and links to web-based genetic and genomic resources. A survey of biochemical and genetic interaction networks in *Drosophila* demonstrates no known interactions among these five genes (Flybase). In humans, there appear to be no known direct biochemical interactions among these genes. Yet, heterozygous LOF mutations in each of these genes are associated with risk for ASD in humans.

### Heterozygous ASD gene mutations have normal synaptic transmission and PHP

We analyzed baseline neurotransmission and presynaptic homeostatic plasticity (PHP) at the *Drosophila* neuromuscular junction (NMJ) as a model glutamatergic synapse. At the *Drosophila* NMJ, PHP is induced by application of sub-blocking concentrations of the postsynaptic glutamate receptor antagonist philanthotoxin-433 (PhTx; 5–10 µM), diminishing the average postsynaptic depolarization caused by the release of single synaptic vesicles (miniature excitatory postsynaptic potential; mEPSP). Decreased mEPSP amplitude initiates a potentiation of presynaptic neurotransmitter release that precisely offsets the magnitude of the PhTx perturbation and, thereby, maintains evoked excitatory postsynaptic potential amplitude (EPSP) at baseline levels prior to the application of PhTx (*Figure 1B–E*; *Davis, 2013*; *Frank et al., 2009*).

First, we characterized baseline synaptic transmission and the rapid induction of PHP in heterozygous null mutations of all five ASD-associated genes, defined above. We find no significant change in baseline neurotransmission, including average mEPSP amplitude, average EPSP amplitude and quantal content (*Figure 1B–E*). Following application of PhTx, we find that heterozygous null mutations in all five ASD gene orthologs do not alter the expression of PHP (*Figure 1B–E*). Specifically, PhTx significantly diminished the average mEPSP amplitude in each heterozygous mutant and induced a statistically significant increase in quantal content that restored EPSP amplitudes toward wild type values. We conclude that all five heterozygous mutations express normal PHP.

### Genetic interaction of *RIMS1* with either *CHD8*, *ASH1L* or *CHD2* impairs PHP

Tests of genetic interaction are commonly used to determine if two genes have a function that converges on a specific biological process. While genetic interactions cannot be interpreted to reflect participation in a linear signaling pathway, such an analysis can define signaling relationships among genes that are independent of whether the encoded proteins interact biochemically. Thus, genetic interactions have been a powerful means to explore new signaling systems in model organisms, an approach that is being increasingly utilized in cancer biology (*Ashworth et al., 2011*; *Mair et al., 2019*; *Chan and Giaccia, 2011*; *O'Neil et al., 2017*; *Baetz et al., 2004*; *Bharucha et al., 2011*) One approach, formally termed '*second site non-complementation* (SSNC)' or '*non-allelic non-complementation*', is particularly powerful when a gene of interest is essential for cell or organismal viability, such as *CHD8* and *CHD2*. In brief, if two heterozygous null mutations, each having no observable

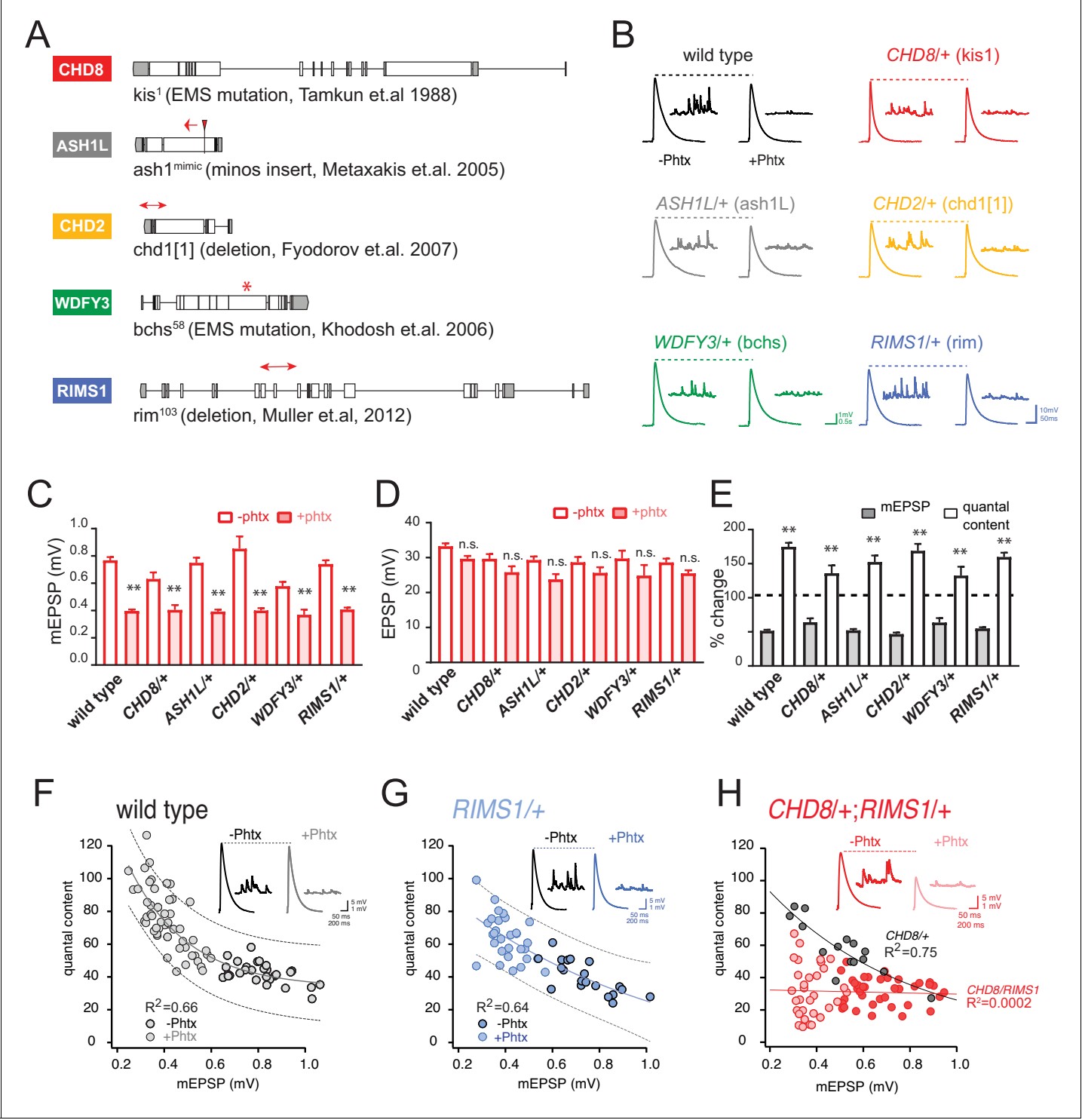

**Figure 1.** Heterozygous ASD gene mutations do not affect baseline transmission or PHP. (**A**) Schematic of the *Drosophila* locus for *CHD8*, *ASH1L*, *CHD2*, *WDFY3* and *RIMS1* with gene disruptions indicated. (**B**) Representative EPSP and mEPSP traces for indicated genotypes (+ / - PhTx for each genotype, left traces and right traces respectively) (**C–D**) Quantification of mEPSP amplitude (**C**) and EPSP amplitude (**D**) in the absence and presence of PhTx (open and filled bars respectively). (**E**) The percent change of mEPSP and quantal content as indicated, comparing the presence and absence of PhTx for each genotype with Student's t-test (two tail), *p<0.05, **p<0.01. Sample sizes for data reported (**C–E**) are as follows (n reported for each genotype -/+ PhTx): *wild type*: n = 36/47; *CHD8/+*: n = 7/8; *ASH1L/+*: n = 15/25; *WDFY3/+*: n = 8/7; *CHD2/+*: n = 8/19; *RIMS1/+*: n = 20/30. (**F–H**) Scatter plots of quantal content (*y* axis) versus mEPSP amplitude (*x* axis) for wild type (left), *RIMS1/+* mutant (middle) and the *CHD8/+; RIMS1/+* double heterozygous mutant. Each symbol represents an individual muscle recording. Inset: representative traces (+ / - PhTx). Exponential data fit (black line,

*Figure 1 continued on next page*

*Figure 1 continued*

R$^2$-value inset, calculated based on a linear fit). Dashed lines encompass 95% of all data (absent in (H) for clarity). Below each graph (F–H), boxes display percent PHP (+ / - PhTx for each genotype), statistical values compared to baseline (H).

The online version of this article includes the following figure supplement(s) for figure 1:

**Figure supplement 1.** Patch-Seq analysis of gene expression in type 1b and type 1 s motoneurons.
**Figure supplement 2.** Double-heterozygous gene mutation combinations impair homeostatic plasticity.

phenotype when tested alone, create a phenotype when combined in a single organism, then the genes are said to genetically interact according to SSNC. We apply this approach here.

The *Drosophila RIMS1* ortholog was previously demonstrated to be a central component of the presynaptic machinery necessary for PHP (*Müller et al., 2012*). Genetic interactions with heterozygous null mutations in *Drosophila RIMS1* have been used to link genes to the mechanism of PHP (*Harris et al., 2018*; *Hauswirth et al., 2018*; *Ortega et al., 2018*). First, we confirm that PHP is robustly expressed in the *RIMS1/+* heterozygous null mutant (*Figure 1E–G*). The average magnitude of homeostatic compensation is indistinguishable from wild type (p>0.1). When we plot the relationship between mEPSP amplitude and quantal content for every individual recording, there is a strong negative correlation observed in both wild type (R$^2$ = 0.66) and *RIMS1/+* (R$^2$ = 0.64) (*Figure 1F and G*, respectively).

Next, we analyzed the heterozygous *CHD8/+* mutant, which also shows robust PHP (*Figure 1C–E*) and a strong negative correlation between mEPSP amplitude and quantal content (R$^2$ = 0.75; *Figure 1H*, gray points and black line). However, animals harboring heterozygous mutations in both *RIMS1* and *CHD8* (*CHD8/+; RIMS1/+*) show a complete failure of PHP (*Figure 1H*). The correlation of mEPSP amplitude and quantal content is abolished (*Figure 1H*, red points and red line; R$^2$ = 0.01). The percent homeostasis in the double heterozygote is decreased to less than 10%, not statistically different from baseline (*Figure 1H*, box; p=0.6), and highly statistically different from both *CHD8/+* and *RIMS1/+* alone (p<0.01). We conclude that *CHD8* can be linked, directly or indirectly, to the mechanisms of PHP. We propose that the heterozygous LOF mutation in *CHD8* weakens the robustness of PHP, thereby associating an ASD-associated chromatin remodeling factor with homeostatic mechanisms that ensure robust synaptic transmission.

Next, to test the generality of this effect, we created double heterozygous mutant combinations of *RIMS1/+* with the remaining ASD orthologs that we examine in this study (*ASH1L*, *CHD2* and *WDFY3*) (*Figure 1—figure supplement 2*). The *ASH1L/+, RIMS1/+* double heterozygous animal shows a complete failure of PHP (*Figure 1—figure supplement 2*). The percent PHP expression is decreased from 152% in the *ASH1L/+* mutant, to 114% in the double heterozygote, which is not different from baseline (p=0.2) and represents a highly significant suppression (p<0.01) compared to the *ASH1L/+* mutant alone (p<0.01) (*Figure 1—figure supplement 2B*). A similar analysis of the *CHD2/+; RIMS1/+* double heterozygous animal shows a significant suppression of PHP (p=0.01), although significant PHP remains expressed in the double heterozygous animals (*Figure 1—figure supplement 2C*). Finally, the *WDFY3/+; RIMS1/+* double heterozygous animal shows robust PHP (*Figure 1—figure supplement 2D*) that is indistinguishable from either the *WDFY3/+* or the *RIMS1/+* single heterozygotes. Taken together, these results suggest that there may be an unexpected connection between three unrelated ASD gene orthologs (*CHD8* and *CHD2* and *ASH1L*) and the mechanisms of PHP, given that all three genes interact with *RIMS1*. Based on these data, we pursued a genome-scale forward genetic screen to interrogate and better define the molecular interface of these ASD gene orthologs and the rapid induction of PHP.

## Forward genetic screen for altered baseline transmission and PHP

The screen that we performed is diagrammed in *Figure 2A*. We took advantage of a collection of small chromosomal deficiencies (5–50 genes per deficiency, each with known chromosomal breakpoints) that tile the 3$^{rd}$ chromosome, uncovering approximately 6000 genes in total. For every double heterozygous combination of *RIMS1/+* with a heterozygous deficiency, we performed multiple (n = 3–15) intracellular recordings, quantifying mEPSP amplitude, EPSP amplitude, quantal content (EPSP/mEPSP), resting membrane potential and input resistance. Recordings were made in the presence of PhTx to induce PHP. If the baseline EPSP is normal and quantal content is increased

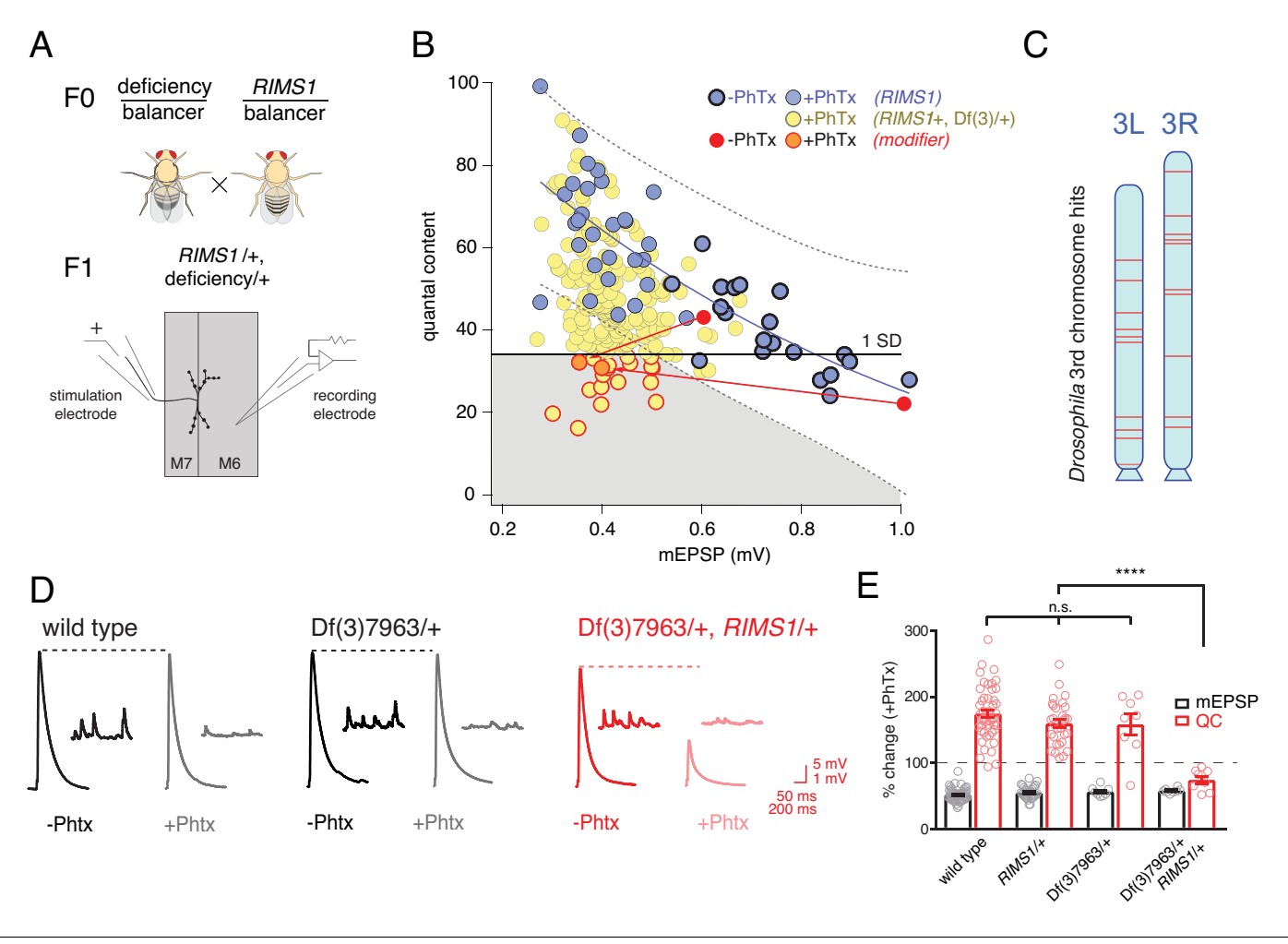

**Figure 2.** Screen for common genomic modifiers of ASD-associated gene mutations. (A) Diagram of genetic screen. (B) Screen results are shown with yellow circles representing average data per genotype. Fit (solid blue line) and confidence interval (dotted lines encompassing 95% of all data) from *RIMS1/+* are overlaid. Black horizontal line defines one standard deviation from population mean (yellow circles). Gray area encompasses potential hits residing outside the *RIMS1/+* confidence interval and below the solid line. Two modifiers are shown in the absence (dark red circles) and presence of PhTx (light red circles, dark outline) (C) Approximate location of hits (red lines) on chromosome 3. (D) Representative traces for indicated genotypes in the presence and absence of PhTx as indicated. (E) Average percent change in mEPSP amplitude (gray bars) and quantal content (red bars) in presence of PhTx compared to baseline. One-way ANOVA and posthoc Tukey's multiple comparisons; ****p<0.0001 for quantal content (QC).

compared to wild type, then we can conclude that PHP is normally expressed. In these instances, we expect that baseline transmission was also normal in the absence of PhTx. However, if EPSPs are diminished in a given genetic combination (*RIMS1/+, Df/+*) and quantal content is not increased compared to wild type, then there are two possible origins: 1) the double mutant impairs baseline transmission or 2) baseline transmission is normal and PHP is selectively impaired. In these instances, the double heterozygous mutant combinations were re-assessed in the absence of PhTx to test for altered baseline transmission.

Double heterozygous combinations that strongly affected muscle resting potential or input resistance were not observed. We uncovered two instances where a mutant combination (*RIMS1/+, Df/+*) caused a specific deficit in baseline transmission, without altering PHP. We did not isolate any double heterozygous conditions with dramatic defects in mEPSP amplitude. The majority of double mutant combinations specifically affected the expression of PHP. This was unexpected.

Double mutant combinations were determined to disrupt PHP by satisfying one of two criteria. First, the average quantal content (+PhTx) had to be more than one standard deviation below the population mean of all genotypes (*Figure 2B*, solid horizontal black line). Second, average quantal

content had to reside outside a boundary that encompasses 95% of all individual recordings made in the *RIMS1/+* mutant alone (*Figure 2B*, black dashed lines). Two example 'hits' are shown in red (*Figure 2B*; dark red point shows data in the absence of PhTx and light red point shows data recorded in the presence of PhTx, and the red lines simply connect the points for a given genotype for the purposes of data display). We also present a complete data set for a single hit from the screen as a standard format bar graph with representative traces (*Figure 2D,E*). Note that the heterozygous deficiency has normal baseline transmission and PHP (p>0.1 One-way ANOVA with post-hoc Tukey multiple comparison), but when combined with a heterozygous mutation in *RIMS1*, PHP completely and selectively fails.

In total, our screen identified, and we subsequently confirmed, 20 small deficiencies that cause PHP to fail when combined with *RIMS1/+*. To achieve a final list of 20 hits, each potential hit was re-validated in a second set of experiments, increasing sample sizes (generally 7–16 NMJ). During the process of re-validation, we rule out approximately one third of the potential hits selected from the screen. The identified deficiencies are randomly distributed across the 3rd chromosome (*Figure 2C*). The screen was empirically validated by the identification, blind to genotype, of deficiencies that uncovered the *RIMS1* locus, as well as the *Pi3K68D* locus (not included in hit list), previously shown to interact as a double heterozygous mutant with *RIMS1/+* (*Hauswirth et al., 2018*). Furthermore, the *rim binding protein* (RBP) locus was not identified as disrupting PHP, consistent with the previously published observation that a *rbp/+* mutant does not interact with *RIMS1/+* for PHP (*Müller et al., 2015*). However, *rbp/+* did interact with *RIMS1/+* for baseline neurotransmitter release as expected based on previously published data (not shown) (*Müller et al., 2015*). No other genes previously implicated in the mechanisms of PHP were present in the deficiencies isolated in our screen. It is important to note that, according to a formal genetic analysis, no strong conclusion can be made regarding the negative result of a double heterozygous genetic interaction (see Supplemental Tables S2-S5 for further detailed information on the screen results).

Finally, we assessed whether there was any relationship between the number of genes that were deleted within a given deficiency and the robustness of PHP. One hypothesis is that the additive effects of multiple, heterozygous gene mutations would increase for larger deficiencies and PHP would be increasingly compromised. That was not the case (*Figure 3*). There was no correlation between the number of genes uncovered by a given deficiency and EPSP amplitude recorded in the presence of PhTx ($R^2$ = 0.003; *Figure 3A*). Thus, impaired PHP cannot be accounted for by a simple additive accumulation of genetic mutations within a given deficiency.

## Identification of common phenotypic enhancers of multiple unrelated ASD genes

The results of our forward genetic screen, encompassing approximately one third of the *Drosophila* genome, might identify genetic interactions specific to *RIMS1*. However, we reasoned that because *RIMS1* also showed a strong genetic interaction with *CHD8*, as well as *CHD2* and *ASH1L* (*Figure 1*; *Figure 1—figure supplement 2*), a portion of the hits from our screen might also interact with these genes. When initial experiments confirmed that this was the case, we expanded our analysis to encompass all five of the ASD-associated gene orthologs from *Figure 1*. Thus, we performed a systems-genetic test of all possible double heterozygous genetic interactions, using wild type and the five ASD gene orthologs introduced in *Figure 1* combined with wild type and five hits (deficiencies) randomly selected from our forward genetic screen. In total, we tested 36 genetic combinations for baseline transmission and PHP, recording every genotype in the presence and absence of PhTx (*Figure 4*).

To facilitate visual interpretation, genetic interaction data are presented as a heat map superimposed on a matrix representing all genetic combinations, in the presence or absence of PhTx (*Figure 4A*). All but one genotype responded to the application of PhTx with decreased mEPSP amplitudes (*Figure 4A*, mEPSP; compare top left matrix with top right matrix, the transition from blue to red indicates diminished average mEPSP for each genetic combination). Thus, we induced homeostatic pressure in 35 out of 36 genetic combinations (*CHD8/+* with *Df(3)7562/+* being the exception). Next, we demonstrate that all heterozygous deficiencies (x-axis) or heterozygous ASD-associated gene mutations (y-axis), when crossed to the wild type strain (*w1118*) showed normal EPSP amplitudes in the absence and presence of PhTx, demonstrating robust induction of PHP (*Figure 4A*, EPSP, bottom right matrix). Next, nearly all (23 out of 25) of the double heterozygous

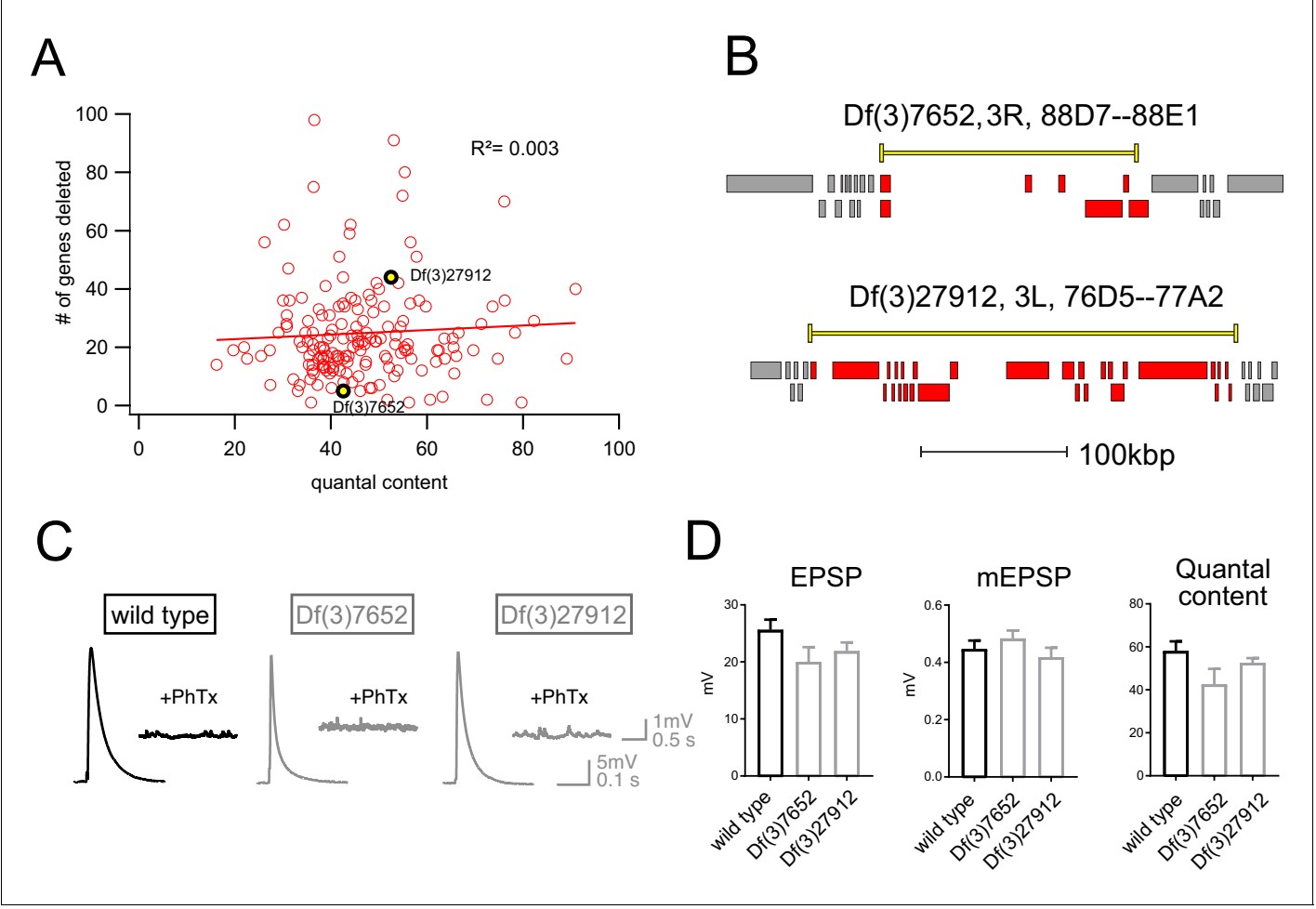

**Figure 3.** Absence of an additive effect of gene heterozygosity on synaptic transmission or PHP. (**A**) Scatter plot showing the number of genes deleted (*y* axis) versus quantal content (*x* axis) in the presence of PhTx for all deficiencies tested. Each circle represents average data from an individual muscle recording for an individual deficiency. Red line shows the fit with a Pearson coefficient of 0.003. (**B**) Schematic of two deficiency alleles showing the extent of the deletion (yellow bars) and the genes deleted (red boxes) (**C**) Representative EPSP and mEPSP traces for indicated genotypes **D**) Quantification of EPSP, mEPSP amplitude and quantal content for the indicated genotypes. All deficiencies recorded as heterozygous mutations in the presence of *RIMS1/+*).

combinations show normal EPSP amplitudes in the absence of PhTx, demonstrating normal baseline neurotransmission (*Figure 4A*, bottom left matrix). However, a majority (16 of 25) of the double heterozygous genetic combinations showed a failure of PHP in the presence of PhTx (*Figure 4A*, red and light-red boxes, bottom right matrix). In *Figure 4B*, we also plot the induction of PHP for each double heterozygous combination by calculating the percent change in quantal content following PhTx application (*Figure 4B*, top matrix). Here, if quantal content does not change (<15% change; gray), then PHP is impaired or blocked. Moderate increases in quantal content (15–30% change; light orange) suggest suppression of PHP, in some instances being statistically significant suppression (see below).

We performed statistical analyses for each double mutant combination, asking whether there was a statistically significant increase in quantal content for a given double mutant in the presence of PhTx compared to that same double mutant combination in the absence of PhTx (*Figure 4B*, bottom matrix.) Note that we are testing whether PHP is induced in a given double heterozygous mutant combination (an individual square in the matrix), comparing quantal content in the absence and presence of PhTx. We do not compare PHP expression among different double heterozygous mutant combinations. The colors gray and 'light pink' each reflect a complete block of PHP, an effect that is observed in the majority of double mutant combinations. As a complementary statistical

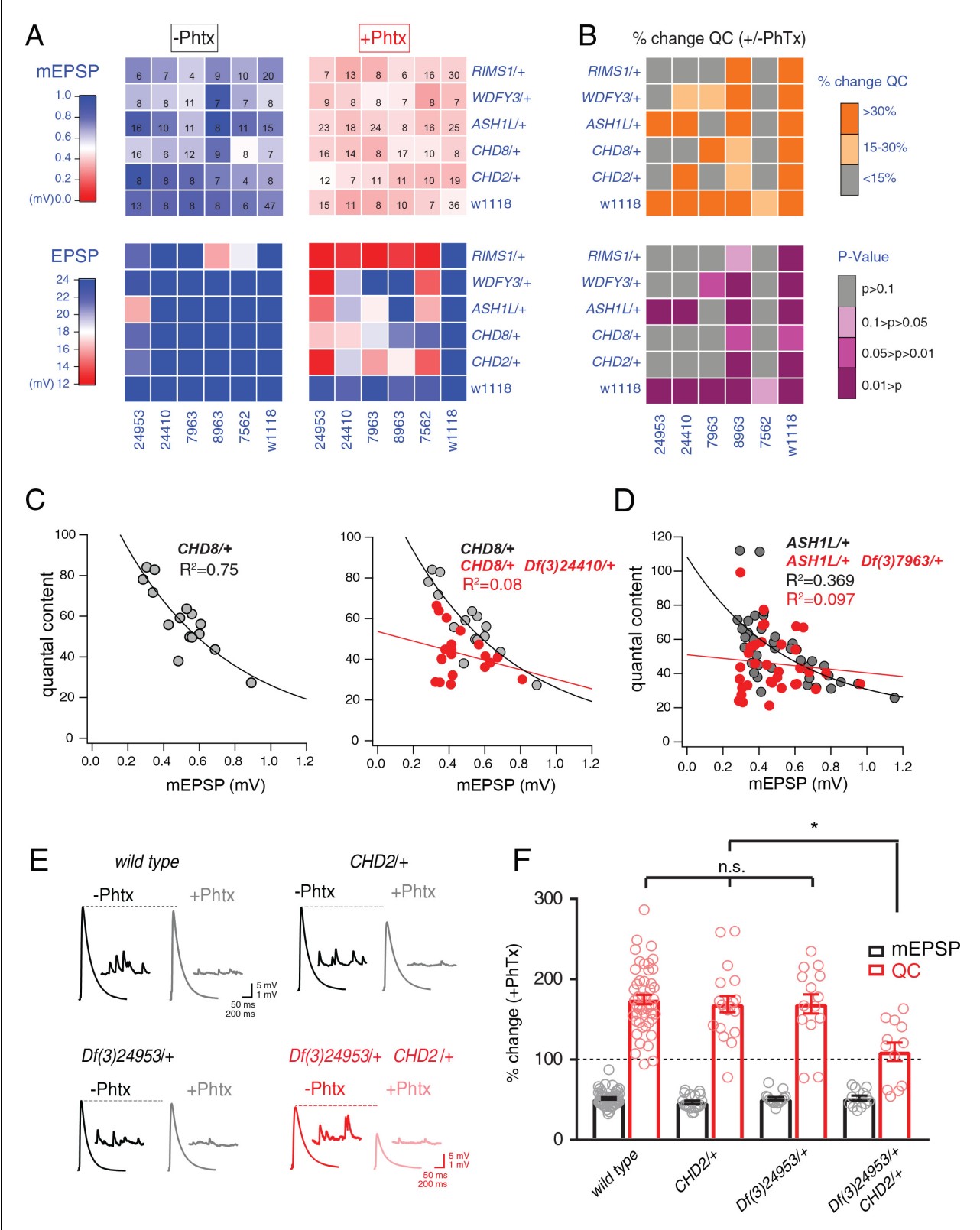

**Figure 4.** Identification of common modifiers of diverse ASD-associated mutations. (**A**) Genetic interaction matrix showing average mEPSP (top two matrix) and EPSP (bottom two matrix) amplitudes in the absence (left) and presence (right) of PhTx, as indicated. Values are according to lookup codes at left. Each individual box represents average data for a double-heterozygous mutant at intersection of x and y axes. Sample size (number of NMJ recordings) is displayed for each box (top) and are identical below (bottom). (**B**) Top matrix (orange and gray) is organized as in (**A**). Average percent

*Figure 4 continued on next page*

*Figure 4 continued*

change in quantal content (+ PhTx) compared to baseline (- PhTx), values according to lookup code. Bottom panel, shows data from top panel re-plotted diagramming p-values for the observed percent change in quantal content (+ / - PhTx), values according to lookup code. Student's t-test (two tail) comparing each genotype + / - PhTx. (C) Scatter plots of quantal content (*y* axis) versus mEPSP amplitude (*x* axis) for *CHD8/+* (left), and *CHD8/+;Df(3)24410/+*. (D) Scatter plot as in (C) for *ASH1L/+* and *ASH1L/+,/Df(3)7963/+*. Each dot represents average data from an individual muscle recording. Fits as indicated. $R^2$ values as indicated (calculated based on linear fit). (E) Representative traces for indicated genotypes (+ / - PhTx) (F) Percent change in mEPSP (gray bars) and quantal content (red bars) in presence of PhTx compared to baseline. One-way ANOVA and posthoc Tukey's multiple comparisons; *p<0.05.

The online version of this article includes the following figure supplement(s) for figure 4:

**Figure supplement 1.** One-way ANOVA with Dunnett's multiple comparisons test (compared to *w1118*).

analysis, we tested the differences between individual genotypic conditions (quantal content in each box) versus the wild type quantal content (One-way ANOVA with Dunnett's multiple comparisons) (*Figure 4—figure supplement 1*). In this case, if PHP is blocked, then there will be a statistically significant difference compared to wild type. Again, 18 of 25 comparisons are significantly different. The genotypic comparison against wild type verified the analysis based on individual genotypic comparisons (*Figure 4B*). It should be noted that, in a few instances, minor differences were observed caused by a change in quantal content that was significant (-/+PhTx), but which remained smaller compared to wild type and therefore became significant. Thus, comparisons within genotypes (-/+PhTx) seem to assess the presence or absence of PHP most accurately (*Figure 4B*).

Our data demonstrate that four out of five deficiencies, isolated in our forward genetic screen as interacting with *RIMS1*, also cause PHP to fail when combined with any one of four different heterozygous ASD-associated gene mutations (*Figure 4B*, bottom). The pattern of PHP blockade is not uniform. *WDFY3*, *CHD8* and *CHD2* show a common pattern of interactions with the same three deficiencies. However, *ASH1L* interacts with only two out of the five tested deficiencies. To our knowledge, this is the first demonstration, in any system, of common phenotypic enhancement for multiple, independent and unrelated ASD gene orthologs.

In *Figure 4C–F*, we elaborate on three of the genetic interactions with data presentations that are more detailed. We show evidence of normal PHP in *CHD8/+* (*Figure 4C*, replicated from *Figure 1*). In the adjacent graph (*Figure 4C*, right), we show evidence of a strong disruption of PHP in the double heterozygous combination of *CHD8/+* with a heterozygous deficiency (*Df(3)24410/+*) isolated in genetic screen. A similar analysis is presented for the *ASH1L/+* heterozygous gene mutation and the interaction with a different heterozygous deficiency (*Figure 4D*). Finally, a third genetic interaction is presented in a format that is standard for the field of homeostatic plasticity (*Figure 4E, F*), inclusive of representative traces (*Figure 4E*) and bar graphs with associated statistical analyses (*Figure 4F*). Note that values for all recordings are presented (*Supplementary file 1* Supplemental Table S6). Several additional controls were performed to validate and extend the findings reported for our genetic interaction data set. First, we note that all double heterozygous mutant combinations are adult viable. Thus, it was possible to inspect adult animals for phenotypes that might indicate altered signaling. Inspection of the compound eye and wings (bristles, wing veins and size) demonstrate wild type tissue morphogenesis (data not shown).

## *PDPK1* and *PPP2R5D* are common phenotypic enhancers of multiple ASD gene orthologs

We isolated the causal single gene mutations within two of the deficiencies isolated from our screen. To do so, we tested smaller sub-deficiencies that mapped within the originally isolated deficiencies. Sub-deficiency mapping either identified the causal gene, or a limited number of candidates. We subsequently tested individual gene candidates with established single gene mutations or RNAi. The process of mapping to single genes, therefore, included several rounds of independent phenotype verification. The first two instances in which we have isolated single causal genes are presented. Each candidate gene was tested individually against all five ASD gene orthologs, using previously published mutations (*Figure 5*). For both genes, we confirmed the same set of genetic interactions that occurred when analyzing the deficiency that included the identified gene (*Figure 5*).

The first gene that we identified encodes a serine threonine kinase encoded by the *PDPK1* gene (*PDK1* in *Drosophila*). *PDPK1* is a master controller of cellular metabolism, as well as cellular and

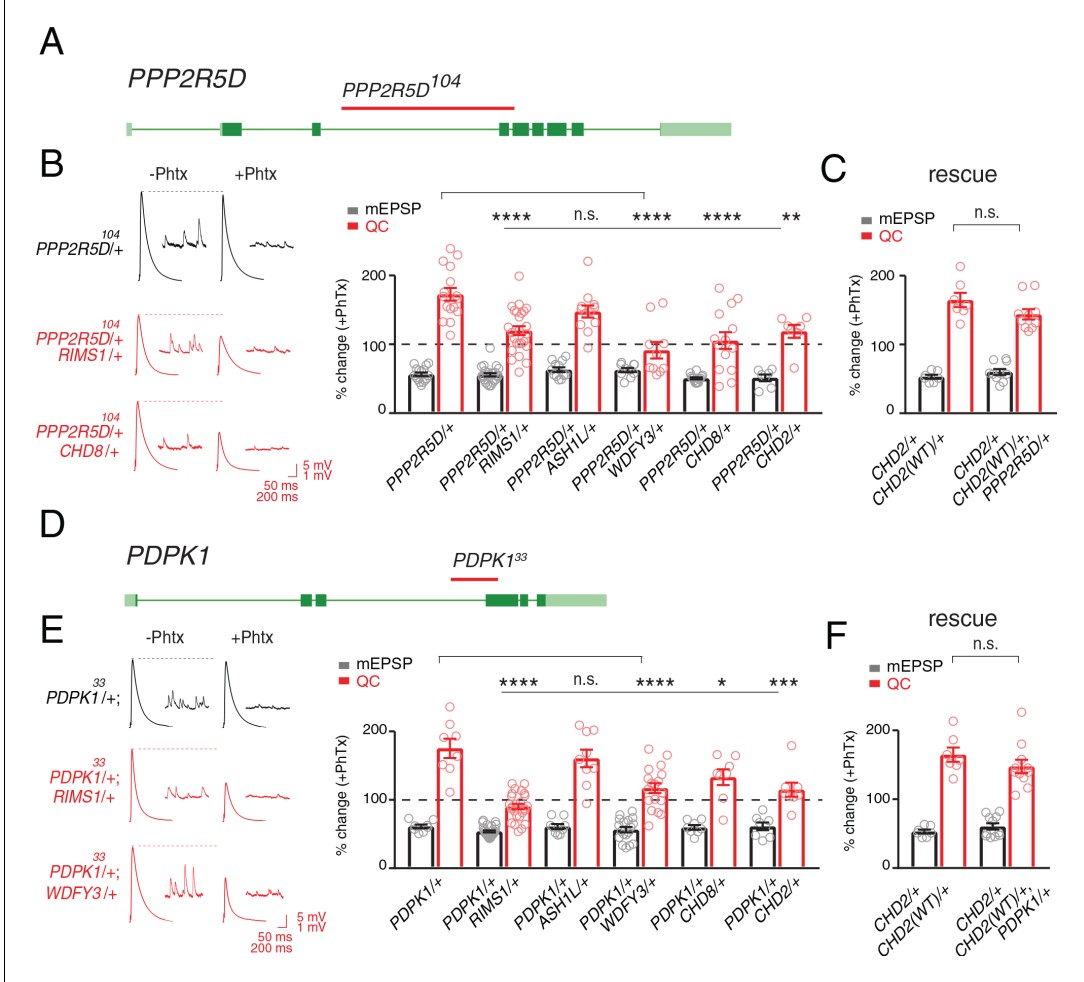

**Figure 5.** Single genes are common modifiers of diverse ASD-associated mutations. (A) Schematic of the *PPP2R5D* gene locus and the *PPP2R5D*[104] deletion mutation (red horizontal bar). (B) Representative traces for indicated genotypes. Bar graph (right) shows percent change in mEPSP (gray) and quantal content (red) (+ / - PhTx). (C) Data as in (B) for rescue of the double heterozygous *CHD2*/+ and *PPP2R5D*[104]/+ mutant by incorporation of a *CHD2* translocation (*CHD2*[WT]/+). (D) Schematic of the *PDPK1* gene locus with the *PDPK1*[33] deletion mutation (red horizontal bar). (E) Representative traces for indicated genotypes. Bar graph (right) as in B. (F) Data as in (C) for the genomic rescue of double heterozygous *CHD2*/+ and *PDPK1*[33]/+ mutants. One-way ANOVA, Dunnett's multiple comparisons *p<0.05, **p<0.01, ***p<0.001, ****p<0.0001 for (B) and (E), Student's t-test, two-tailed for (C) and (F), n.s. p>0.05).

The online version of this article includes the following figure supplement(s) for figure 5:

**Figure supplement 1.** Analysis of the NMJ morphology.

**Figure supplement 2.** A *PPP2R5D* loss-of-function mutation disrupts PHP, but PDPK1 does not.

**Figure supplement 3.** Firing properties of motoneurons are not different in a double heterozygous mutant.

synaptic growth at the *Drosophila* NMJ (*Cheng et al., 2011*). The second gene encodes a regulatory subunit of the PP2A phosphatase encoded by the *PPP2R5D* gene (*wrd* in *Drosophila*) (*Viquez, 2006*). *PPP2R5D* is also a master controller of cellular metabolism (*Bernal et al., 2014*), as well as cellular and synaptic growth at the *Drosophila* NMJ (*Viquez, 2006*). Both proteins are present at the *Drosophila* NMJ (*Cheng et al., 2011*; *Viquez et al., 2009*; *Viquez, 2006*) Single-cell Patch-Seq experiments confirmed the expression of these genes in *Drosophila* motoneurons at third-instar (*Figure 1—figure supplement 1*). Intriguingly, the *PPP2R5D* gene has recently been associated with intellectual disability and autism in human (*Loveday et al., 2015*; *Shang et al., 2016*).

Next, we demonstrate that both genes, *PDPK1* and *PPP2R5D,* are common modifiers of multiple heterozygous ASD-associated gene mutations (*Figure 5*). To underscore the specificity of the double heterozygous genetic interactions (*Figure 5B,E*), we restored the expression of the *CHD2* gene to

wild type levels in the *CHD2/+; PPP2R5D/+* double heterozygous combination. This was achieved using a previously published translocation of the *CHD2* gene locus that allowed us to put back one copy of the *CHD2* gene (*CHD2^WT^/+*) in the background of the *CHD2/+; PPP2R5D/+* double mutant. We demonstrate that PHP is fully restored (*Figure 5C*). An identical series of experiments was performed with a previously characterized *PDPK1* mutation (*Figure 5E,F*). Thus, *PDPK1* and *PPP2R5D* represent the first known common phenotypic modifiers of ASD gene orthologs, causing PHP to fail (see also *Supplementary file 1* Supplemental Table S7 for values and statistics). The data also underscore that deficiencies isolated in our forward genetic screen can be resolved to the activity of single genes. As such, the screen may have identified a novel class of common phenotypic modifier.

Another series of control experiments were performed. We asked whether neuronal morphology was substantially altered in the heterozygous ASD-associated gene mutations and in select double heterozygous genetic interactions (*Figure 5—figure supplement 1*). We do find evidence that the heterozygous *CHD8/+* mutation predisposes the NMJ to modest overgrowth, consistent with *CHD8* influencing brain development in other systems (*Gompers et al., 2017*). But, this effect does not become more severe when combined with either the *PDPK1* or *PPP2R5D* mutation as double heterozygotes. Thus, we conclude that altered synaptic growth is not highly correlated with the block of PHP in these double heterozygous combinations.

## Deletion of *PPP2R5D* impairs the robustness of PHP

Given that *PPP2R5D* and *PDPK1* both genetically interact with multiple ASD-associated gene mutations, and given that several of the ASD-associated genes sensitize PHP toward failure, we considered whether *PPP2R5D* and *PDPK1* are also directly involved with the induction of PHP. Unlike many of the ASD genes, both *PPP2R5D* and *PDPK1* are viable as homozygous deletion mutations, allowing a direct test of their involvement in PHP. To our surprise, neither *PPP2R5D* nor *PDPK1* can be classified as strictly essential for the mechanisms of PHP based on analysis of homozygous LOF mutants (*Figure 5—figure supplement 2*). PHP is fully expressed in the homozygous *PDPK1* mutant (*Figure 5—figure supplement 2*), demonstrating that this gene is not required. There is a statistically significant suppression of PHP in the homozygous LOF mutation in *PPP2R5D* suggesting a role for this gene in the rapid induction of PHP, but without being strictly necessary (*Figure 5—figure supplement 2*).

We note that both *PDPK1* and *PPP2R5D* control signaling that directly intersects with the AKT/mTOR pathway, a signaling system that is associated with ASD in human (*Alessi et al., 1997*; *Manning and Toker, 2017*; *Yeung et al., 2017*). The mTOR signaling proteins *S6K* and *Tor* have both been implicated in the long-term maintenance of PHP. However, both are dispensable for the rapid, PhTx-dependent induction of PHP (*Cheng et al., 2011*; *Penney et al., 2012*). Never-the-less, the possible connection to Tor signaling prompted us to revisit our screen data and ask whether mutations affecting the broader AKT/mTOR signaling system might also be common ASD-gene modifiers. The genes *Akt, S6K, TSC1, TSC2,* and *PTP61F* are all encoded on the *Drosophila* third chromosome. All of these genes were present within the deficiencies that were tested in our screen. But, none were identified as a hit in our unbiased forward genetic screen. Although the lack of a genetic interaction cannot be used to conclude the absence of a role for these genes in the PHP effects that we observe, it seems likely that *PPP2R5D* and *PDPK1* have other targets relevant to the intersection of ASD-gene mutations and the rapid induction of PHP. Consistent with this possibility, *PDPK1* and *PPP2R5D* are predicted to have opposing actions on AKT, yet both genes participate in the blockade of PHP when combined with a mutation in one of the five ASD-associated gene mutations (see discussion). Furthermore, as demonstrated below, one mechanism by which PHP is blocked is novel and unexpected.

## Dissecting the mechanism of impaired PHP in a single double heterozygous mutant combination

It is rare for a genetic study to define, precisely, how a double heterozygous interaction creates a synthetic phenotype if the two genes do not encode proteins that biochemically interact. Simply put, there are a vast number of possible mechanisms by which SSNC could occur (*Yook et al., 2001*). None-the-less, we attempted to do so for at least one double heterozygous combination. Although this represents only a single mechanism of SSNC, it could provide proof of principle for

how PHP is affected in other ASD gene interactions. We chose the genetic interaction of *PPP2R5D/ +* with *CHD8/+*. This combination was chosen because *CHD8* is among the most common ASD de novo gene mutations. Furthermore, the genetic interaction is highly penetrant.

We began by pursuing additional phenotypic analyses, looking for clues in a wider variety of cellular and electrophysiological measures. It is possible that the genetic interaction of *PPP2R5D/+* with *CHD8/+* could indirectly affect PHP expression by altering motoneuron firing properties. Therefore, we analyzed intrinsic excitability and neuronal firing by patch clamp electrophysiology of larval motoneurons. There is no change in motoneuron firing frequency in response to a series of step current pulse injection. Likewise, there are no changes in action potential amplitude, cell input resistance or rheobase comparing wild type with each single heterozygous mutation and the double heterozygote (*Figure 5—figure supplement 3*). Thus, aberrant excitability is not linked to impaired PHP.

## Ultrastructural correlate of impaired PHP: altered presynaptic membrane trafficking

Next, we turned to electron microscopy to determine whether the genetic interaction of *PPP2R5D/+* with *CHD8/+* affects the presynaptic release site. Ultrastructural changes have previously been linked to impaired PHP (*Harris et al., 2018*). Thin section transmission electron microscopy (EM) was used to examine the synapse, defined as a characteristic increase in pre- and postsynaptic membrane electron density, opposing clustered presynaptic vesicles and a characteristic presynaptic density, termed a T-bar. We find that the ultrastructure of *CHD8/+* alone was wild type (*Figure 6B,D,E*). The ultrastructure of *PPP2R5D/+* alone was wild type (*Figure 6A,D,E*). However, the double heterozygous mutant showed evidence of large membrane structures surrounding the presynaptic release site and apparent stalled endocytic events, appearing adjacent to sites of neurotransmitter release where compensatory synaptic vesicle endocytosis occurs (*Figure 6C*, insets). Quantification of vesicle size reveals a large increase in average intracellular vesicle diameter for all vesicles within 150 nm of the base of the presynaptic release site, defined by the T-bar structure (*Figure 6D,E*), again selective to the double heterozygous mutant. These data provide a striking visual confirmation of the genetic interaction between *PPP2R5D/+* and *CHD8/+*. And, this is further evidence linking the action of a chromatin-remodeling factor (CHD8) to the stability of synaptic transmission.

Given the appearance of enlarged vesicles at or near the presynaptic release site, we repeated our ultrastructural analysis of the double heterozygous mutant, fixing the synapse immediately (~1–5 s) after strong stimulation of presynaptic release (50 Hz stimulation, 10 s). In wild type, there was no change in the number or appearance of presynaptic vesicles when fixed immediately following the stimulus. However, in the double heterozygous mutant condition (*CHD8/+; PPP2R5D/+*) we found that intracellular vesicles were further increased in size and took on a crenulated appearance (*Figure 6F–I*). These data are consistent with the enlarged vesicles being endosomal intermediates, arguing that the process of vesicle recycling is altered in the double heterozygous mutant. In further support of this idea, we demonstrate enhanced synaptic depression in response to high frequency (50 Hz) stimulation (*Figure 6—figure supplement 1*). Recently, homozygous *CHD8* loss of function mutations have been linked to defects in synaptic vesicle endocytosis at the *Drosophila* NMJ (*Latcheva et al., 2019*). Our data underscore that that an endocytosis phenotype can be uncovered in the heterozygous *CHD8/+* mutant in the context of the *PPP2R5D/+* mutant. Regardless of the underlying molecular mechanism leading to this EM phenotype and associated physiological deficits (a topic for future study), these data present a striking, visual confirmation of a strong synthetic genetic interaction between *PPP2R5D/+* and the *CHD8/+* heterozygous mutations. Furthermore, these data link the activity of a chromatin remodeling factor, present in the nucleus (CHD8), to a profound synaptic defect. Experiments detailed below, including genetic rescue, confirm the specificity of this EM phenotype.

## Differential gene expression analyses

One possible reason that genes isolated from our screen are common modifiers of diverse ASD genes is that each modifier is a direct transcriptional target of the ASD mutants. It is possible to assess this by RNAseq. To our knowledge, side-by-side differential gene expression analysis has yet to be performed for multiple heterozygous ASD-associated gene mutant backgrounds. We performed whole genome RNAseq analysis for wild types and the four heterozygous ASD mutants (four

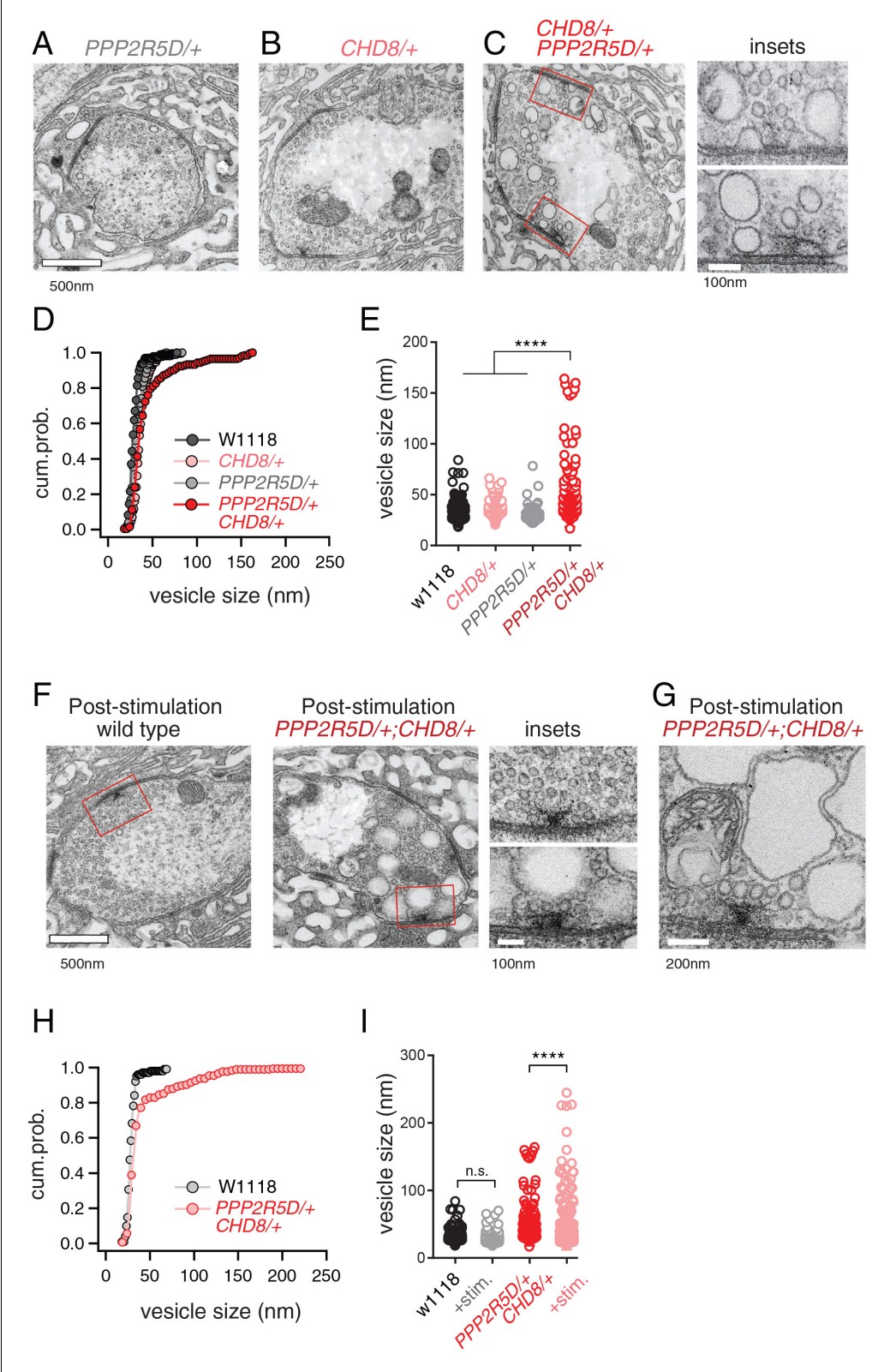

**Figure 6.** ASD gene-modifier interaction causes impaired synaptic membrane organization. (**A–C**) Representative electron microscopy images of individual boutons inclusive of (**A**) *PPP2R5D/+*, (**B**) *CHD8/+* and (**C**) *CHD8/+; PPP2R5D/+* double heterozygous mutant. Insets (**C**) show individual active zones taken from the image on the left (red rectangles) (**D**) Cumulative probability distribution of the vesicle size for wild type (*w1118*) and *CHD8/+* and *PPP2R5D/+* single mutants, as well as the *CHD8/+; PPP2R5D/+* double heterozygous mutants. Each point reflects the average at a single active zone.
*Figure 6 continued on next page*

*Figure 6 continued*

(E) Plot of individual data points for each genotype as shown in (D). Sample sizes (D, E): Animal number: w1118 N = 2, *CHD8/+* N = 2; *PPP2R5D/+* N = 2; *CHD8/+; PPP2R5D/+* N = 3. Active zone number: in same genotypic order n = 12, n = 23, n = 10, n = 14; Vesicle number n = 97, n = 171, n = 85; n = 89. (F) Representative electron microscopy images for individual boutons for indicated genotypes after stimulation with 50 Hz for 10 s and rapid fixation. Insets show active zones for *wild type* (top) and the *CHD8/+; PPP2R5D/+* double heterozygous mutant (bottom) taken from the images on the left (red rectangles) (G) An example image from the *CHD/+; PPP2R5D+* double heterozygous mutant with larger vesicles having a crenulated appearance after stimulation. (H) Cumulative probability distribution of the vesicle size for wild type (*w1118*) and the double heterozygous mutant *CHD8/+; PPP2R5D/+* after stimulation and rapid fixation. Each point reflects the average at a single active zone. (I) Plot of individual data points for data in (H). Sample sizes (H, I): Animal number: *w1118* with stimulation N = 2, *CHD8/+; PPP2R5D/+* with stimulation N = 2; Active zone number n = 10, 21. Vesicle number n = 101, n = 175. One-way ANOVA Tukey's multiple comparisons, ****$p < 0.001$, n.s. $p > 0.05$.

The online version of this article includes the following figure supplement(s) for figure 6:

**Figure supplement 1.** Analysis of short-term depression in *CHD8/+; PPP2R5D/+* double heterozygote.

biological replicates) (*Figure 7A*). We asked whether any of the genes contained within the 20 deficiencies identified in the screen (37 genes; a number arrived at following sub-deficiency mapping and sub-selection based on gene expression in nerve or muscle) are commonly altered in all four of ASD-associated mutants (*Figure 7A*, orange data points). None were commonly differentially regulated (p-value=0.096 for *ASH1L/+*; p-value=0.636 for *WDFY3/+*; p-value=0.392 for *CHD2/+*; p-value=0.112 for *CHD8/+*; Wilcoxon sign rank test two-sided). We conclude that common down-regulation of identified genetic modifiers cannot account for the common impairment of PHP that we observe electrophysiologically.

Next, we asked whether the ASD-associated gene mutations might cause common changes in gene expression, with potential relevance to a common disruption of PHP. We define all differentially expressed genes common to at least two ASD mutations (*Figure 7B,C*). While there are individual genes that are commonly differentially regulated, a GO database analysis of differentially expressed genes did not reveal any consistent change in a gene category across all four genotypes. The patterns of gene dysregulation do not predict any pattern of genetic interactions documented in our systems-genetic analysis. Finally, while there are genes that are commonly dysregulated in multiple ASD gene orthologs, there are only two genes that are commonly down-regulated in all four ASD mutants (*FBgn0027578* [*Nepl21*] and *FBgn0037166* [*CG11426*]) (*Figure 7C*). *FBgn0027578* encodes a metalloprotease of the Neprilysin family, with homology to endothelin converting enzyme one in human, of unknown function in the nervous system. *FBgn0037166* encodes phosphatidic acid phosphatase type 2, which is expressed in the *Drosophila* nervous system, but of unknown function. There is no obvious means to connect the down regulation of these two genes to impaired homeostatic signaling, although future experiments will explore these genes in greater depth. Furthermore, there is no clear connection between FBgn0027578 or FBgn0037166 and the roles of either PDPK1 or PPP2R5D in the nervous system. Thus, a transcriptional analysis of heterozygous ASD gene mutations alone did not allow us to make clear progress toward understanding the mechanisms of impaired PHP.

## Candidate mechanisms for impaired PHP based on differential gene expression analysis

Next, we continued with our focus on characterizing the homeostasis defect in the *CHD8/+; PPPR25D/+* double heterozygous mutant combination. We repeated the RNAseq differential gene expression analysis comparing the double heterozygous condition to three control conditions, inclusive of wild type and each single heterozygous mutant alone. In this manner, we sought to identify synergistic effects on gene expression that could not be accounted for in either single heterozygous mutant alone (*Figure 7D,E*). As expected, many of the differentially expressed genes documented in the double heterozygous mutant, when compared to wild type, could be accounted for by subsequent comparisons to each single heterozygous mutant. However, a small number of genes (14 genes; 5 upregulated and nine downregulated) appear to be synergistically differentially expressed in the double heterozygous mutant compared to all three control conditions (*Figure 7D,E*). We successfully replicated altered expression of four genes in the double heterozygous mutant combination by quantitative RT-PCR (*Figure 8A,B*). Of these genes, *CREG* stood out as being robustly and dramatically up-regulated. Upon closer inspection, *CREG* showed a slight, but significant, up-regulation

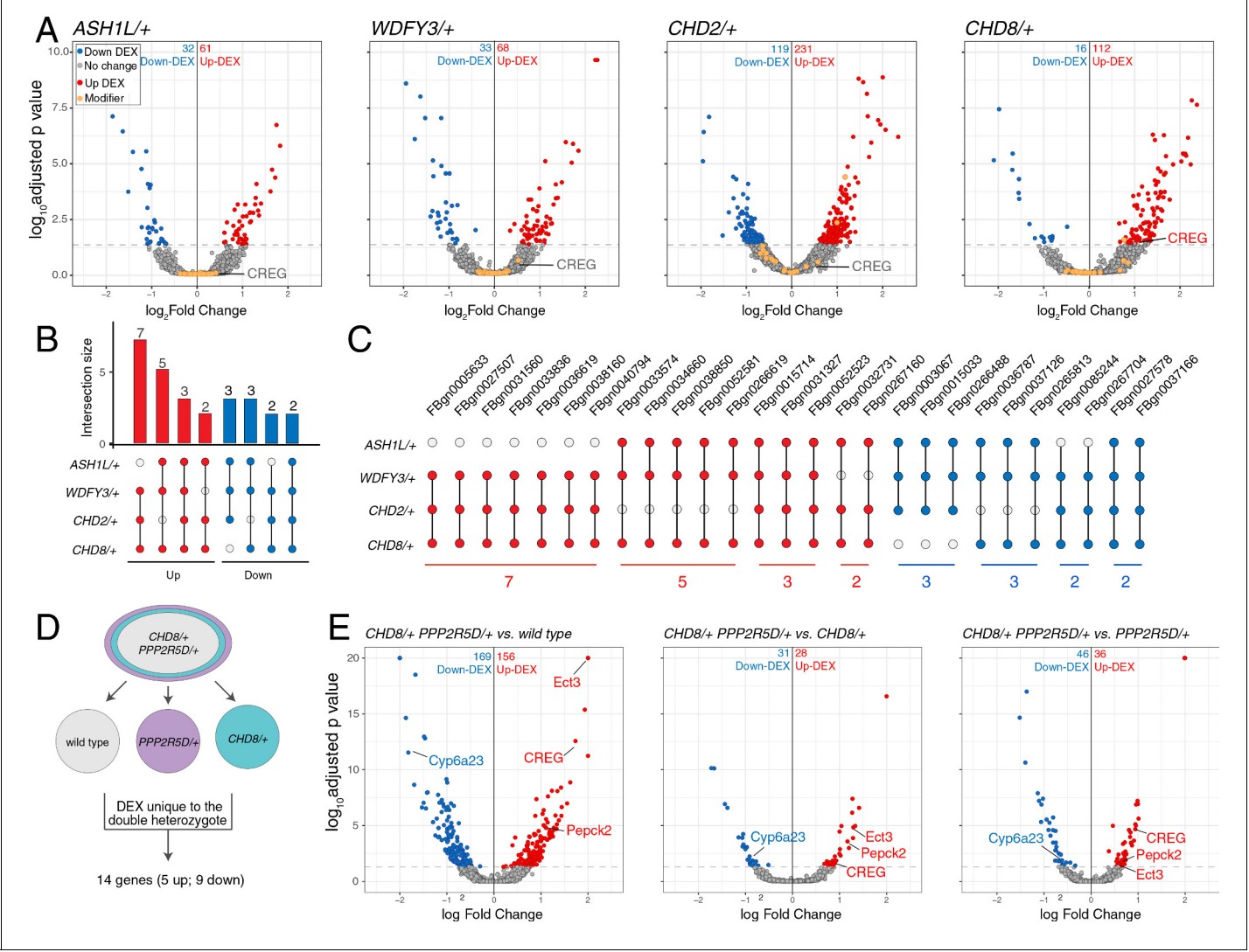

**Figure 7.** Differential gene expression analysis identifies *CREG*. (**A**) Volcano plot display of differentially expressed genes (DEX) for each heterozygous mutant versus wild type. Candidate ASD-gene modifiers are indicated (orange dots). Horizontal dashed line indicates cutoff of adjusted p-values (0.05). (**B**) Matrix shows all intersections of DEXs from the four indicated genotypes (see Database S1). Filled circles in the matrix indicate sets that are part of the intersection between genotypes. Bar graphs on the top show the total number of DEXs for each set, ordered by the size of intersection. (blue, up-regulated; red, down-regulated). (**C**) Individual genes are listed at the intersection of each genotypes. (**D**) Schematic showing the selection of 14 genes uniquely dysregulated in *CHD8/+; PPP2R5D/+* double heterozygous mutants. (**E**) Volcano plot display of DEX calculated as *CHD8/+; PPP2R5D/+* versus wild-type, *CHD8/+; PPP2R5D/+* versus *CHD8/+* and *CHD8/+; PPP2R5D/+* double heterozygotes versus *PPP2R5D/+* alone.

in the *CHD8/+* mutant (*Figure 7A*), and this was enhanced by the presence of the heterozygous *PPP2R5D/+* mutation (*Figure 7E*, left). Next, we confirmed the up-regulation of *CREG* in the third instar larval central nervous system by quantitative RT-PCR (*Figure 8—figure supplement 1B*). Finally, we took advantage of a previously published gene expression data set (*Parrish et al., 2014*) and document *CREG* expression in motoneurons throughout embryonic and larval development. *CREG* is strongly expressed in embryonic motoneurons (20–24 hr after egg laying – AEL), after which expression levels plummet (*Figure 8—figure supplement 1A*).

## *CREG* is a homeostatic repressor

*CREG* (Cellular Repressor of E1A-stimulated Genes) encodes an endosomal/lysosomal localized glycoprotein that is linked to stress responses in other systems as well as to the homeostatic

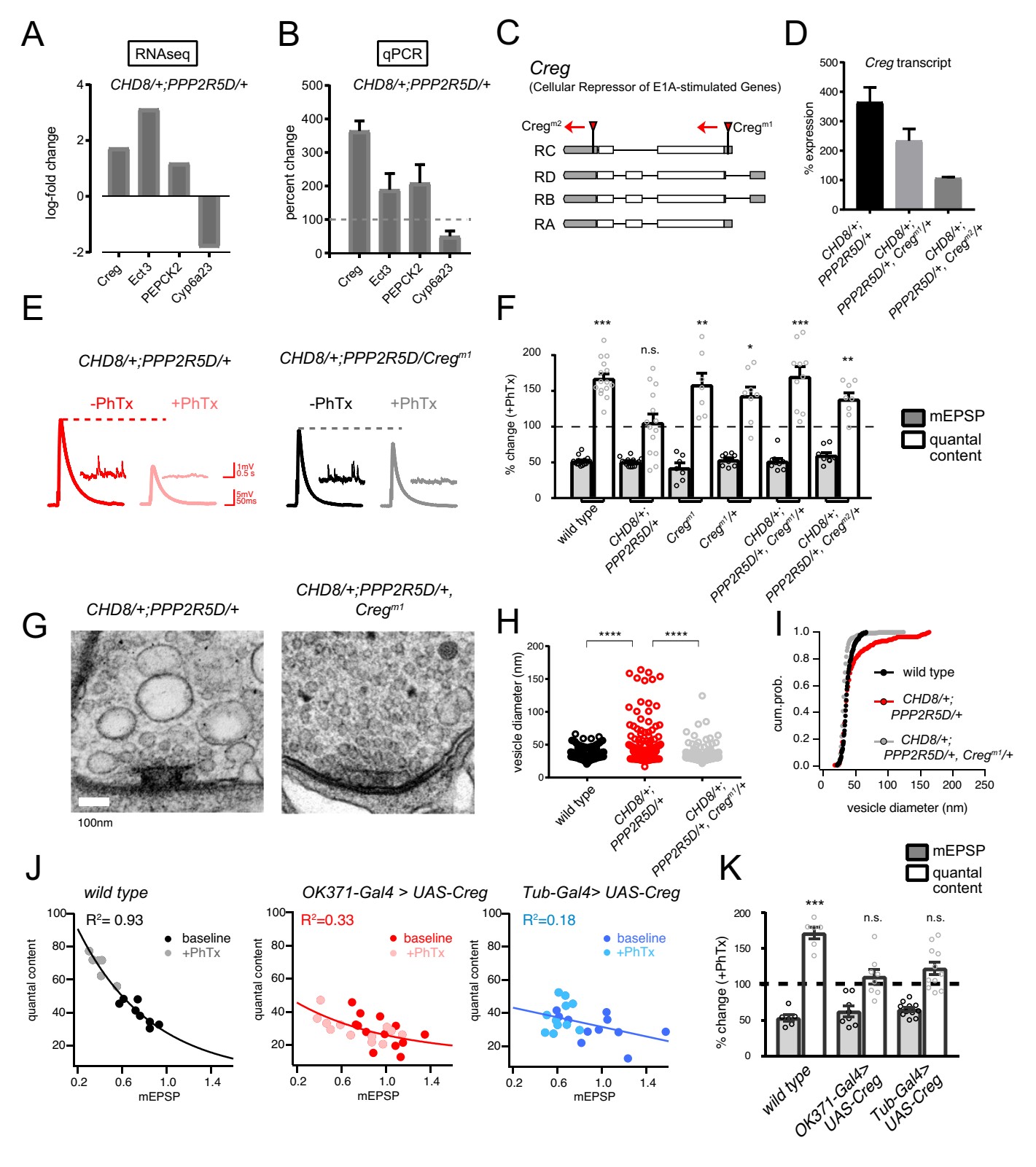

**Figure 8.** *CREG* is a homeostatic repressor that blocks PHP and regulates synapse ultrastructure. (**A**) Quantification of transcriptional changes calculated by RNAseq for four genes (*CREG, Ect3, PEPCK2* and *Cyp6a23*) in *CHD8/+; PPP2R5D/+* double heterozygous mutant versus wild-type. (**B**) Quantification of the transcriptional changes for the same genes in (**B**) by qPCR. (**C**) Schematic of the *Drosophila CREG* locus. The positions of two transposon insertion mutations are shown (red triangles). (**D**) Average *CREG* transcript levels calculated by qPCR are shown for the indicated genotypes

*Figure 8 continued on next page*

*Figure 8 continued*

E) Representative EPSP and mEPSP traces for indicated genotypes. (**F**), Bar graph (right) shows percent change in mEPSP (black filled) and quantal content (no fill) (+ / - PhTx). Sample size indicated as (–PhTx/+PhTx): *wild type* n = 17/15; *CHD8/+; PPP2R5D/+* n = 11/14; *Creg$^{M1}$* n = 8/7; *Creg$^{M1}$/+* n = 8/8; *CHD8/+; PPP2R5D/+; Creg$^{M1}$/+* n = 8/10; *CHD8/+; PPP2R5D/+; Creg$^{M2}$/+* n = 8/8. (**G**) Representative electron microscopy images of individual active zones from indicated genotypes (double heterozygous mutant at left, triple heterozygous mutant at right). Scale bar:100 nm. (**H**) Individual data points (vesicle size) shown for indicated genotypes. (**I**) Cumulative probability distribution of vesicle size for genotypes shown in (**H**). Sample sizes for (**H, I**): Animal number: *wild type* N = 2, *CHD8/+; PPP2R5D/+* N = 3; *CHD8/+; PPP2R5D/+; Creg$^{M1}$/+* N = 3. Active zone number: in same genotypic order n = 12, n = 14, n = 12; Vesicle number n = 97, n = 89, n = 112. (**J**) Scatter plots of quantal content (*y* axis) versus mEPSP amplitude (*x* axis) for *wild type* (left), *OK371-Gal4 > UAS* Creg (middle, red) and *Tub-Gal4 >UAS* Creg (right, blue). Fits as indicated. R$^2$ values as indicated (calculated based on linear fit). (**K**) Percent change in mEPSP (gray bars) and quantal content (red bars) in presence of PhTx compared to baseline. Sample sizes as in (**F**), *wild type* n = 8/6; *OK371-Gal4 > UAS*) Creg n = 14/12; *Tub-Gal4 >UAS*) Creg n = 11/11. n.s. p>0.05, **p<0.01, ***p<0.001, ****p<0.0001.

The online version of this article includes the following figure supplement(s) for figure 8:

**Figure supplement 1.** Expression levels of *CREG* during *Drosophila* larval development.
**Figure supplement 2.** CREG overexpression does not substantively alter NMJ anatomy.
**Figure supplement 3.** Ultrastructure analysis of the *CHD2/+; PPP2R5D/+* double heterozygous mutant.

maintenance of the vascular epithelium (*Ghobrial et al., 2018*; *Kowalewski-Nimmerfall et al., 2014*). Mammalian orthologs are expressed in the brain (*Yang et al., 2011*). However, *CREG* function has never been addressed in the nervous system of any organism. Given that we observe a strong synaptic internal membrane phenotype in the *CHD8+; PPP2R5D/+* double heterozygous mutant, and given that CREG localizes to the endo-lysosomal system, we chose to study *CREG* in greater detail, asking if it is causally involved in PHP.

Two independent transposon insertion mutations were identified, residing in the *Drosophila CREG* gene locus (*Figure 8C*). The *CREG$^{M1}$* transposon completely abolishes *CREG* expression and a heterozygous *CREG$^{M1}$/+* mutant reduces *CREG* expression (*CREG$^{M1}$* = zero expression compared to wild type, 3 biological and three technical replicates; *CREG$^{M1}$/+* = 51.5 ± 3.1% wild type expression, 3 biological and three technical replicates). Next, we generated a triple heterozygous mutant combination (*CHD8+; PPP2R5D/+, CREG$^{M1}$/+*) and find that the *CREG$^{M1}$/+* allele attenuates the up-regulation of *CREG* gene transcript in the triple heterozygous mutant background, a suppression effect of approximately 50%, as predicted (*Figure 8D*). Then, we repeated this analysis with the *CREG$^{M2}$* allele. This allele has a minor effect on baseline *CREG* expression (73.2 ± 2.7% wild type expression, 3 biological and three technical replicates). However, we discovered that this transposon insertion caused a complete block of *CREG* up-regulation in the triple heterozygous mutant combination, suggesting that this transposon insertion, residing in 3' UTR, may disrupt a transcription regulatory motif (*Figure 8D*).

Next, we asked whether the triple heterozygous mutant combinations, in which *CREG* up-regulation is either attenuated or abolished, would rescue the expression of homeostatic plasticity and synaptic ultrastructure. In both triple mutant combinations (*CHD8+; PPP2R5D/+, CREG$^{M1}$/+*) and (*CHD8+; PPP2R5D/+, CREG$^{M2}$/+*), the expression of PHP is fully rescued (*Figure 8E,F*). These data are consistent with the conclusion that the abnormally enhanced levels of *CREG* transcription are responsible for the block of homeostatic plasticity seen in the double heterozygous mutant combination. If true, then we might also see rescue of the ultrastructural phenotype in the *CHD8+; PPP2R5D/+, CREG$^{M1}$/+* triple mutant. Indeed, this is the case (*Figure 8G*). We observe full rescue of synaptic ultrastructure. Thus, preventing the dramatic up-regulation of *CREG*, without abolishing *CREG* expression, is sufficient to restore membrane trafficking and PHP to the presynaptic nerve terminal of the *CHD8+; PPP2R5D/+* double mutant combination.

It is possible that CREG is a novel suppressor of PHP. However, it is also possible that CREG mediates this effect only in the context of the other two heterozygous mutations. To address this possibility, we generated a *UAS-CREG* transgenic line, allowing cell-type specific overexpression of the *CREG* gene. Over-expression of *CREG* in a wild type background using either a ubiquitously expressed source of GAL4 (*tubulin-GAL4*), or a GAL4-line that is selective to motoneurons (*OK371-GAL4*), causes a complete block of PHP (*Figure 8J–K*). As a control for adverse developmental effects of *CREG* overexpression, we analyzed NMJ anatomy and find no substantive effects on NMJ growth or morphology that could account for the absence of PHP (*Figure 8—figure supplement 2*).

Finally, we assessed the consequences of the heterozygous and homozygous loss of function mutations on baseline neurotransmission and PHP. The $CREG^{M1}$ allele abolished expression (see above) and is the focus of these analyses. Neither the heterozygous nor homozygous animals affected expression of PHP (*Figure 8F*). The $CREG^{M1}$/+ heterozygous animals had no effect on baseline transmission compared to wild type (*wild-type* QC = 40.8 ± 2.2 n = 10; $CREG^{M1}$/+ QC = 38.6 (±3.0) n = 8; Student's t-test; p>0.5). The $CREG^{M1}$ homozygous allele decreased baseline transmission by ~18% ($CREG^{M1}$ QC = 33.5 ± 3.0 n = 8; p=0.02). Clearly, neither baseline release nor PHP are potentiated, demonstrating that the rescue of PHP in the triple heterozygous mutant condition cannot be considered an additive effect of the heterozygous $CREG^{M1}$ mutation. Taken together, our data are consistent with the conclusion that *CREG* is a homeostatic repressor, one of very few identified to date (*Spring et al., 2016*). This finding underscores the complexity of interpreting the double heterozygous mutant combinations that cause blockade of PHP.

## Assessing the generality of CREG as a mechanism for impaired PHP

In mammals, there are two *CREG* genes and *CREG2* is expressed in the brain (*Yang et al., 2011*). A recent study provides evidence that *CREG2* expression is enhanced in layer four excitatory neurons, isolated from human postmortem ASD patient brain tissue (*Velmeshev et al., 2019*), suggesting possible relevance. This fact prompted us to ask whether over-expression of *CREG* is the primary mechanism responsible for the disruption of PHP, or whether it is just one of many. Our existing gene expression analysis demonstrates that *CREG* is not up-regulated in the other heterozygous mutations (*ASH1L* or *CHD2* or *WDFY3*; *Figure 7*). This was extended to the *PDPK1/+* mutant and, again, *CREG* levels are not increased. Finally, we analyzed two additional double heterozygous mutant combinations (*PDPK1/+* with *CHD2/+* as well as *PPP2R5D/+* with *CHD2/+*). CREG was not up-regulated compared to single heterozygous controls. Finally, we repeated the ultrastructural analysis for a second genetic combination (*PPP2R5D/+* with *CHD2/+*). No phenotype of enlarged vesicles or endomembranes was observed (*Figure 8—figure supplement 3*). From these data, we conclude that the aberrant over-expression of *CREG* is not a universal cause of impaired PHP in the double heterozygous interactions. In the future, a systematic test of all genetic combinations identified in our screen may define whether CREG over-expression is unique to a single genetic interaction or whether it is reflected in a subset of gene interactions.

## Discussion

In this study, we make several fundamental advances. First, we provide evidence that mutations in multiple different ASD-associated genes sensitize homeostatic plasticity to fail (*Figure 9A,B*). Second, using genome-scale forward genetics and subsequent systems-genetic analyses, we identify the first phenotypic modifiers that *commonly enhance* five different ASD-associated gene mutations, causing a specific failure of PHP (*Figure 9A*). Third, we identify *PDPK1* and *PPP2R5D* as common phenotypic modifiers of multiple ASD-associated genes and, thereby, define a mechanistic link between synaptic transmission, PHP and chromatin remodeling complexes in the neuronal nucleus (*Figure 9A*). Finally, we define how PHP fails at the intersection of an ASD-associated gene mutation and phenotypic modifier. The mechanism is unexpected, involving the maladaptive up-regulation of a novel repressor of homeostatic plasticity (CREG) (*Figure 9C*). We demonstrate that up-regulation of *CREG* cannot explain other gene-gene interactions, underscoring the potential complexity of gene-gene interactions and the common failure of PHP. Regardless of potential mechanistic complexity, our data argue that impaired PHP may be a common pathophysiological effect downstream of LOF mutations in five different ASD-associated genes. If our data can be extended to additional ASD genes, and to other experimental systems including human neurons, then it may be possible to use this information to advance therapeutic approaches that modify ASD phenotypic severity regardless of the underlying genetic mutation(s) that confer risk for ASD.

The loss or impairment of PHP could contribute to the phenotypic penetrance of an ASD gene mutation in multiple ways. Impaired PHP is expected to render the nervous system less robust to perturbation including the effects of environmental stress, immunological stress, or genetic mutation (*Davis, 2013*; *Davis, 2006*). If an ASD-associated gene mutation leads to neural developmental defects, then loss of PHP would be expected to exacerbate the functional consequences. According to the same logic, loss of PHP might enhance the adverse effects of environmental or immunological

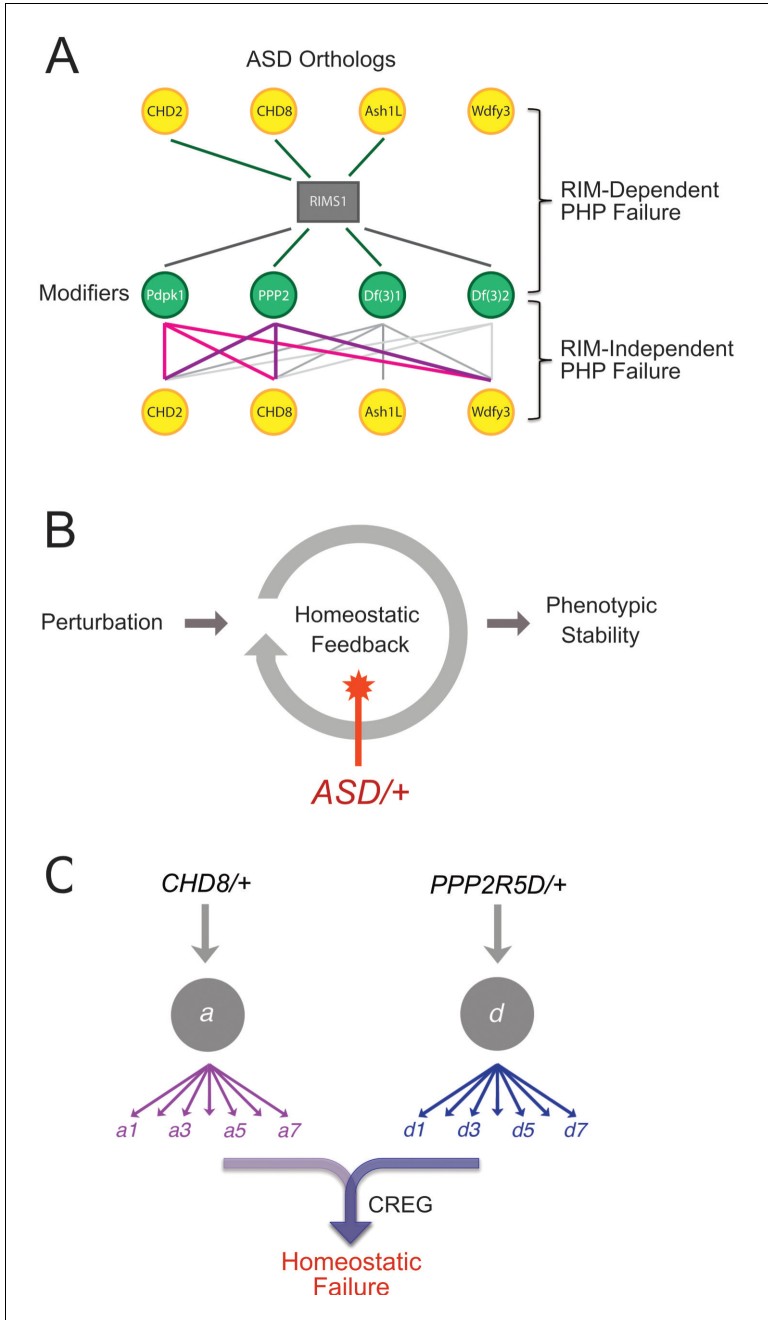

**Figure 9.** Summary and Model. (**A**) Summary of genetic interactions. *RIMS1* interacts with three of four ASD gene orthologues, impairing PHP. *RIMS1* interactions identified in a genetic screen as modifiers are shown below in green. Each modifier interacts with multiple ASD heterozygous mutations in a <u>*RIMS1*</u> independent manner, disrupting PHP. The data present a complex network of gene-gene interactions (yellow and green) that diminish the robustness of PHP. Df(3)1 refers to *Df(3)7562* and Df(3)2 refers to *Df(2)24953*. (**B**) Homeostatic signaling systems robustly ensure stable neural function. However, the homeostatic signaling system itself is sensitive to genetic perturbation. We demonstrate that PHP is sensitive to mutations in multiple genes that were identified as ASD risk factors. In at least one instance, this is due to the up-regulation of a PHP interfering factor (*CREG*) and the red star indicates this a possible mechanisms more generally. (**C**) Complexity of interpreting double heterozygous gene-gene interactions. Signaling systems are not blocked by heterozygous gene mutations, but are likely to be attenuated to some degree. The combined effect of two higher-order heterozygous gene mutations creates a downstream, intersectional effect that is very difficult to predict. In the case of this paper, we succeeded in identifying a novel intersection causing up-regulation of *CREG*, which disrupts the homeostatic signaling system.

stress, both of which are thought to contribute to ASD pathophysiology (*Beversdorf et al., 2018*; *Modabbernia et al., 2017*). Finally, loss of PHP could be relevant to the appearance or severity of ASD comorbidities, including epilepsy.

It should be emphasized that failed homeostatic plasticity cannot be determined by simply assessing the phenotype of a heterozygous ASD-associated gene mutation. The observation of a phenotype, such as altered E/I balance or impaired neurotransmission, could reflect failure of homeostatic plasticity, or it could reflect the outcome of successful homeostatic mechanisms that constrained a phenotype that might otherwise have been more severe (*Davis, 2013*; *Kulik et al., 2019*). Ultimately, the loss or impairment of homeostatic plasticity can only be determined by a direct test of homeostatic robustness; specifically referring the ability of a neuron, synapse or neural circuit to respond to a perturbation and sustain normal function in the continued presence of the perturbation (*Davis, 2013*; *Davis, 2006*). Thus, our data set the stage for similar analyses in other model organisms, potentially extending the connection between ASD-associated gene mutations and the robustness of PHP or other forms of homeostatic plasticity.

## The specificity of gene-gene interactions that cause PHP to fail

The genetic interactions that we document in our study appear to be highly specific. First, our genetic screen was based on the use of deficiency chromosomes that uncover 5–50 genes each, rendering those genes heterozygous. Thus, each deficiency can be considered to test pairwise gene-gene interactions among all the genes contained in the deficiency. According to this logic, we tested in excess of 50,000 double heterozygous gene combinations and discovered only 20 interactions that cause PHP to fail. Although it is unlikely that genes are completely randomly distributed throughout the genome, this calculation still has merit and emphasizes the rarity of gene-gene interactions that cause PHP to fail. In addition, we found no correlation between the number of genes deleted in a heterozygous deficiency and the impairment of PHP. Thus, the likelihood of a genetic interaction does not increase with the number of genes that are rendered heterozygous. Finally, it should be emphasized that PHP is a robust physiological process that is not unusually susceptible to the effects of genetic mutations. Previously, forward genetic have observed low rates of gene discovery. Two such screens tested transgenic RNAi against nearly every kinase and phosphatase encoded in the *Drosophila* genome, a gene set that includes prominent signaling proteins, the majority of which had no effect on PHP induction or expression (*Brusich et al., 2015*; *Hauswirth et al., 2018*). With this information as a background, the identification of genes that commonly enhance multiple ASD genes, causing PHP to fail, seems extraordinarily.

## The rapid induction versus long-term expression of PHP

There are two well-established methods to induce expression of PHP. Application of PhTx induces PHP within minutes, a process that can be maintained for hours (*Frank et al., 2006*). In addition, a mutation in the non-essential *GluRIIA* subunit of postsynaptic glutamate receptors drives persistent expression of PHP. Since the *GluRIIA* mutation is present throughout the life of the organism, it is inferred that this reflects the long-term maintenance of PHP. Although this distinction reflects only the duration of the perturbation (acute versus genetic), recent work does argue that the acute induction of PHP may transition to another long-term expression mechanism (*Harris et al., 2018*; *Harris et al., 2015*). Indeed, screens based on the acute versus long-term PHP have identified different candidate genes, even when screening a common transgenic RNAi collection (*Brusich et al., 2015*; *Hauswirth et al., 2018*).

It remains unknown whether one form of PHP is more relevant regarding the intersection of homeostatic plasticity with diseases or disorders of the nervous system. In the present study, the acute induction of PHP can be considered a type of 'stress test'. If the rapid induction of PHP fails, we can infer that the neurons are less robust to perturbation. In the future, it will be interesting to systematically determine whether the gene-gene interactions identified here also uniformly perturb PHP induced by the *GluRIIA* mutation. However, such an analysis is beyond the scope of the present study.

## Common phenotypic enhancers of multiple ASD gene orthologs

How can the existence of common phenotypic modifiers be explained? We began our study with the demonstration that heterozygous LOF mutations in four unrelated ASD-associated genes including *RIMS1* (presynaptic scaffolding protein), *CHD8* (chromatin helicase), *CHD2* (chromatin helicase) and *ASH1L* (transcriptional activator and histone methyltransferase), all sensitize the expression of PHP to fail (*Figure 1*, *Figure 1—figure supplement 2*, *Figure 9B*). One possibility, therefore, is that PHP is commonly sensitized to fail by heterozygous LOF mutations in each of the five ASD gene orthologs that we chose to study. If so, then a phenotypic modifier that interacts with one of these genes might also be expected to commonly interact with the other ASD genes. In other words, commonality arises because of the unexpected finding that each ASD gene ortholog has an activity that, when diminished, impairs the robustness of PHP. Our data generally support this model, given that three of four ASD genes interact with *RIMS1* to block PHP. According to this model, we provide the first evidence that sensitization of PHP is a common pathophysiological effect downstream of multiple ASD genes with, as yet, unrelated biological activities.

The finding that ASD gene mutations sensitize PHP to fail does not require that each ASD gene participate in the actual mechanisms of PHP. *RIMS1* is a core component that is required for PHP (*Müller et al., 2012*). However, a gene such as *CHD8* might compromise PHP indirectly by causing some form of cellular stress that interacts with the mechanisms of PHP (*Figure 8J,K*). Indeed, it was previously demonstrated that simultaneous induction of two different forms of homeostatic plasticity creates interference and homeostatic failure (*Bergquist et al., 2010*). The same argument can apply to the novel class of common phenotypic modifiers. Some modifiers may represent core components of PHP, including *PPP2R5D*, which seems to suppress PHP when knocked out (*Figure 5—figure supplement 2*). However, the *PDPK1* knockout has no effect on PHP and, therefore, may interact with the mechanisms of PHP indirectly. Thus, we cannot rule out the possibility that compounded cellular stressors occasionally intersect and cause PHP to fail.

## Novel mechanisms impair PHP; CREG-dependent suppression of PHP

We explored, in detail, how PHP fails at the intersection of *CHD8/+* and *PPP2R5D/+*. First, we discovered a profound effect on synaptic ultrastructure that was not observed in either single heterozygous mutation. This provided dramatic visual proof of a strong, genetic interaction between these two heterozygous gene mutations. Next, we demonstrate that this strong, genetic interaction is not a consequence of extensive transcriptional dysregulation. Indeed, when the effects of each heterozygous gene mutation are taken into account, only 14 genes show evidence of altered transcription. A single gene, *CREG*, was subsequently demonstrated to be the cause of impaired PHP and disrupted presynaptic membrane trafficking. Although *CREG2* is not upregulated in the heterozygous *CHD8/+* mouse, a recent study provides evidence that *CREG2* expression is enhanced in layer four excitatory neurons, isolated from human postmortem ASD patient brain tissue (*Gompers et al., 2017*; *Velmeshev et al., 2019*).

It remains to be determined how loss of *PPP2R5D* causes further dysregulation of *CREG* in the background of *CHD8/+*. One possibility is that *CREG* is a stress-response gene, and up-regulation occurs at the intersection of two cellular stresses. Other alternatives remain plausible, including a direct connection between *CHD8* and *CREG* that is modulated by PPP2R5D-mediated signaling. The biochemical and transcriptional relationships will be defined in subsequent work and are beyond the scope of our current study. The generality of this genetic interaction will also be explored. We note, for example, that *CREG* shows a mild increase only in the *CHD8/+* mutant, not in the other three ASD-associated genes (*Figure 7A*). This does not rule out *CREG* participating in genetic interactions involving other ASD-associated genes, but it might suggest additional mechanisms will be engaged.

*CREG* encodes a glycoprotein that localizes within the endo-lysosomal system and may also be secreted. In mammals, there are two *CREG* genes and *CREG2* is expressed in the brain (*Yang et al., 2011*). There is generally more information regarding the function of *CREG1*, which is an effector of tissue homeostasis in the vascular epithelium (*Ghobrial et al., 2018*). In this capacity, CREG seems to function as a stress response factor, influencing the activity of several potent signaling systems (*Ghobrial et al., 2018*). Our current phenotypic analyses suggest that increased levels of *CREG* may directly impact the integrity of synaptic vesicle membrane recycling and, either directly or indirectly, interface with the homeostatic potentiation of vesicle release. Thus, while a full dissection of *CREG*

activity remains for future studies, our data argue that *CREG* has an activity that could be directly coupled to vesicle release and recycling, an ideal situation to normally limit the homeostatic potentiation of vesicle fusion.

## Relevance and conclusions

It is well established that genetic context can profoundly influence the phenotypic severity of disease-causing gene mutations. For example, in mice, it has been shown that genetic context (strain background) influences phenotypic penetrance in an Alzheimer's disease model (*Neuner et al., 2019*). In humans, systematic screening of the phenotypically normal population has identified individuals that are resistant to the effects of well-established, debilitating disease causing mutations, an effect termed 'resilience' that is attributed to the effects of genetic context (*Chen et al., 2016*; *Friend and Schadt, 2014*). It seems plausible that the common phenotypic enhancers, identified in our genetic screen, could represent a mechanism by which genetic context influences the phenotypic penetrance of ASD-associated gene mutations. We recognize that *PDPK1* and *PPP2R5D* have fewer than expected LOF and missense mutations in humans (http://exac.broadinstitute.org). It remains to be determined if this will be the case with additional modifier genes. Furthermore, we note that PHP is completely blocked at the intersection of ASD gene mutations and the common modifiers we identify. Therefore, subtle changes in the expression or function of common phenotypic modifiers, perhaps caused by mutations in enhancer/promoter regions, could impact expression or robustness of PHP with cascading negative phenotypic consequences. If our findings can be extended to other systems, including humans, it is conceivable that our emerging mechanistic understanding could be used to restore the beneficial effects of homeostatic plasticity and alleviate aspects of ASD phenotype, irrespective of individual genetic makeup.

## Materials and methods

**Key resources table**

| Reagent type (species) or resource | Designation | Source or reference | Identifiers | Additional information |
|---|---|---|---|---|
| Genetic reagent (*D. melanogaster*) | w1118 | Bloomington *Drosophila* Stock Center | BDSC:3605 | |
| Genetic reagent (*D. melanogaster*) | rim103 | PMID:23175813 | | |
| Genetic reagent (*D. melanogaster*) | kis1 | Bloomington *Drosophila* Stock Center | BDSC:431 | |
| Genetic reagent (*D. melanogaster*) | chd1[1] and chd1 [1], chd1[wt] | PMID:21177652 | | |
| Genetic reagent (*D. melanogaster*) | ash1-mimic | Bloomington *Drosophila* Stock Center | BDSC:23524 | |
| Genetic reagent (*D. melanogaster*) | bchs58 | Bloomington *Drosophila* Stock Center | BDSC:9887 | |
| Genetic reagent (*D. melanogaster*) | UAS-CREG | this paper | | Fly carrying UAS-Creg transgene |
| Genetic reagent (*D. melanogaster*) | wrd104 | PMID:16957085 | | |
| Genetic reagent (*D. melanogaster*) | pdk1 | PMID:21930778 | | |
| Genetic reagent (*D. melanogaster*) | ok371-gal4 | PMID:16378756 | | |
| Genetic reagent (*D. melanogaster*) | tubulin-gal4 | PMID:21930778 | | |

*Continued on next page*

*Continued*

| Reagent type (species) or resource | Designation | Source or reference | Identifiers | Additional information |
|---|---|---|---|---|
| Genetic reagent (*D. melanogaster*) | 3rd chromosome deficiency collection | Bloomington *Drosophila* Stock Center | | |
| Genetic reagent (*D. melanogaster*) | Creg-m1 | Bloomington *Drosophila* Stock Center | BDSC:42140 | |
| Genetic reagent (*D. melanogaster*) | Creg-m2 | Bloomington *Drosophila* Stock Center | BDSC:22800 | |
| Genetic reagent (*D. melanogaster*) | MN1b-gal4 | Bloomington *Drosophila* Stock Center | BDSC:40701 | |
| Genetic reagent (*D. melanogaster*) | MN1s-gal4 | Bloomington *Drosophila* Stock Center | BDSC:49227 | |
| Genetic reagent (*D. melanogaster*) | uas-cd8:gfp | PMID:10197526 | | |
| Chemical compound, drug | Philanthotoxin-433 | Santa Cruz Biotechnology | GH28782 | |
| Antibody | Anti-brp (Mouse monoclonal) | Developmental Studies Hybridoma Bank | RRID:AB_2314866 | IF (1:100) |
| Antibody | Anti-dlg (Rabbit monoclonal) | PMID:29303480 | | IF (1:1000) |
| Antibody | Cy3 anti-rabbit | Jackson Immuno-research Laboratories | RRID:AB_2338000 | IF (1:500) |
| Antibody | Alexa488 anti-mouse | Jackson Immuno-research Laboratories | RRID:AB_2338840 | IF (1:500) |
| Software, algorithm | Igor Pro 8.03 | Wavemetrics | RRID:SCR_000325 | |
| Software, algorithm | Graphpad PRISM 7.04 | Graphpad | RRID:SCR_002798 | |
| Software, algorithm | Adobe Illustrator CC 2018 | Adobe | RRID:SCR_010279 | |
| Software, algorithm | MiniAnalysis 6.0.7 | SynaptoSoft | RRID:SCR_002184 | |
| Software, algorithm | SlideBook 6 | Intelligent Imaging | RRID:SCR_014300 | |
| Sequence-based reagent | CREG primer | Applied Biosystems | Dm02135967_g1 | |
| Sequence-based reagent | Ect3 primer | Applied Biosystems | Dm02139373_g1 | |
| Sequence-based reagent | Pepck2 | Applied Biosystems | Dm02366462_s1 | |
| Sequence-based reagent | Cyp6a23 | Applied Biosystems | Dm01824231_g1 | |
| Sequence-based reagent | rpl32 | Applied Biosystems | Dm02151827_g1 | |
| Commercial assay or kit | RNeasy Plus Mini kit | QIAGEN | ID:74134 | |
| Commercial assay or kit | SuperScript III First-Strand synthesis system | Invitrogen | Cat# 18080051 | |

*Continued on next page*

*Continued*

| Reagent type (species) or resource | Designation | Source or reference | Identifiers | Additional information |
|---|---|---|---|---|
| Commercial assay or kit | TURBO DNA-free kit | ThermoFisher | Cat# AM1907 | |
| Commercial assay or kit | TaqMan Fast Universal PCR Master Mix | Applied Biosystem | Cat# 4352042 | |
| Commercial assay or kit | Lexogen's Split RNA Extraction Kit | Lexogen | Cat# 008 | |
| Commercial assay or kit | 3'mRNA-Seq Library Prep Kit | Lexogen | Cat# 015 | |
| Commercial assay or kit | Single Cell/Low Input RNA Library Prep Kit | New England Biolabs | Cat# E6420S | |
| Recombinant DNA reagent | pTW (Gateway vector) | DGRC | Cat# 1129 | |
| Recombinant DNA reagent | pENTR-dTOPO Cloning Kit | Invitrogen | Cat# K240020 | |
| Recombinant DNA reagent | Creg cDNA | Drosophila Genomics Resource Center | GH28782 | |

## Fly stocks

All *Drosophila* stocks were kept and raised on standard food at 25˚C. *RIMS1* was previously described (*Müller et al., 2012*). *PPP2R5D* was a gift from Dr. Aaron Diantonio. *Chd2[1]* and *Chd2[1], Chd2^{wt}* flies were gifts from Dr. Alexandra Lusser and Dr. Dmitry Fyodorov. All other *Drosophila* stocks were obtained from the Bloomington *Drosophila* Stock Center unless otherwise noted. *W1118* is used as wild-type controls. 3$^{rd}$ chromosome deficiency fly stocks are balanced over *TM6b* and all stock are on the *W1118* background.

## Molecular biology

*Drosophila* CREG cDNA was obtained by amplifying the single open reading frame from genomic DNA by PCR and cloning directly in to the pENTR vector (Gateway Technology; Invitrogen). We engineered a CACC site in the forward primer for the subsequent Gateway reaction: forward primer for pUASt-creg: 5' CACCATGGATTCGGACAGCACC 3'; reverse primer for pUASt-creg with a stop codon, 5' TCA ATT CGA AAC AGC GTA ATA 3'. The final construct were sequenced to ensure there were no mutations. The creg cDNA was then cloned into proper destination vector obtained from the *Drosophila* Gateway Vector Collection (Carnegie Institution, DGRC barcode #1129). Transgenic lines were generated and mapped using standard methods.

## Electrophysiology

All current clamp recordings were performed from muscle six, at the second and third segment of the third-instar *Drosophila* with an Axoclamp 900 amplifier (Molecular Devices). The composition of the extracellular solution (HL3) is (in mM) 70 NaCl, 5 KCl, 10 MgCl$_2$, 10 NaHCO$_3$, 115 sucrose, 4.2 trehalose, 5 HEPES. Ca$^{2+}$ concentration in the extracellular solution is 0.35 mM unless otherwise noted. Homeostatic plasticity was induced by incubating the larvae with Philantotoxin-433 (PhTx, 15–20 μM, Sigma) for 10 min as previously described (*Frank et al., 2006*; *Genç et al., 2017*). Quantal content calculation is made by dividing average EPSP to mEPSP. mEPSPs were analyzed with MiniAnalysis program (Synaptosoft). All other physiology data were analyzed with custom written functions in Igor 6 (Wavemetrics Inc). Data collected from a minimum of two animals from two independent crosses.

## Immunohistochemistry

Third-instar larvae were dissected, fixed in Bouin's fixative or 4% PFA in PBS, and immunostained with previously described methods (*Eaton et al., 2002*; *Harris et al., 2015*). Third instar larvae were dissected with cold HL3 and immediately fixed with PFA (4%) and incubated overnight at 4 C with primary antibodies (rabbit anti-Dlg, 1:1000; anti-Brp 1:100, Life Technologies). Alexa-conjugated

secondary antibodies were used for secondary staining (Jackson Laboratories 1:500). An inverted epifluorescence deconvolution confocal microscope (Axiovert 200, Zeiss) equipped with a 100X objective (N.A. 1.4), cooled CCD camera (CoolSnap HQ, Roper Scientific) was used to acquire images. All acquisition, deconvolution and analysis were done by Slidebook 5.0 software (3I, Intelligent Imaging). Structured illumination microscopy (Nikon LSM 710 equipped with 63X objective and Andor Ixon EMCCD camera) was used to perform Brp puncta and Dlg labeling experiments. Bouton numbers were quantified as described previously (*Harris et al., 2015*).

## RNA extraction and library preparation for RNA sequencing

RNA was extracted from the adult heads (5–7 days post-pupation) of heterozygous mutants of four genotypes (*ASH1L*/+ and *WDFY3*/+ and *CHD2*/+ and *CHD8*/+) and wild types with four biological replicates per group by using Lexogen's RNA Extraction Kit (Lexogen). RNA quality was checked with Bioanalyzer (Agilent Technologies Inc) prior to library amplification. 3'mRNA-Seq Library Prep Kit for Illumina (FWD) from Lexogen was used for first strand cDNA, second strand synthesis, dsDNA purification, i7 single indexing, library amplification and final library purification. To estimate the PCR cycle numbers for library amplification, qPCR was done by using PCR Add-On Kit (Lexogen). Purified final libraries were quality tested by using Agilent Bioanalyzer 2100 with High Sensitivity DNA chips (Agilent Technologies Inc). Qubit fluorometer (ThermoFisher Scientific) was used to quantify the concentration of the final library. Barcoded libraries are then sequenced using an Illumina HiSeq 4000 at 50 bp single-end reads in the CAT genomic facility at UCSF. There was a median of 14.3 million (M) mapped reads per sample (interquartile range, IQR: 8.0 M-20.6M).

## Read mapping and quantification of gene expression

Read count and transcript per million reads mapped (TPM) were determined using Salmon software version 0.12.0. A reference genome index for Salmon was created according to developer's instructions for the *Drosophila melanogaster* genome BDGP6 (Ensembl v92). Reads mapping and quantitation was simultaneously performed to individual transcripts.

## Differential expression across heterozygous mutant flies

Differential expression of heterozygous mutant flies was determined by pooling samples from the same genotype. Gene expression profiles between mutant and wild type were collated using the R package tximport (version 1.6.0). The R package DESeq2 (version 1.18.1) used raw gene counts to determine differentially expressed genes (DEGs) by genotype with the linear model [Gene counts ~Batch + Genotypes]. Protein coding and lincRNA genes defined by the BDGP6 were included in differential expression. Expression was adjusted for batch to account for difference between fly lines, tissue source, and library preparation. The p-values were adjusted for Benjamini-Hochberg Procedure through DESeq2 with a target alpha = 0.1, and genes were considered DEGs at FDR < 0.05 and±50% expression changes.

## Single cell Patch-seq

To obtain the cytoplasmic content of the cell, we performed whole-cell tight-seal patch clamp in motoneurons expressing GFP (*Kulik et al., 2019*). We established whole-cell configuration with leak currents less than 100 pA. We gently sucked the cytoplasmic content of the cell by applying a negative pressure to the patch-pipette. Then, we pulled the individual motoneuron from the tissue while visually confirming the GFP fluorescent signal at the tip of the pipette. Immediately after, we immersed the pipette tip in a test tube containing the Cell Lysis Buffer and RNAse inhibitor medium and broke the pipette tip by gently touching to the tube wall. The content of the pipette tip was ejected by applying positive pressure. We pooled 4–8 motoneurons for one reaction.

We used the Low Input RNA: cDNA Synthesis, Amplification and Library Generation kit from NEB (New England Biolabs Inc) to isolate, reverse-transcribe the RNA and prepare the libraries for sequencing. Following the reverse transcription and template switching, we amplified the cDNA by PCR. Amplified cDNA was cleaned up by using SPRI beads. The quality and quantity of the amplified cDNA was assessed by Bioanalyzer (Agilent Technologies Inc). After fragmentation and adaptor ligation, adaptor-ligated DNA were enriched with i7 primer and universal primer by PCR-amplification. Amplified libraries were quality checked by Biolanalyzer with High Sensitivity DNA chips (Agilent

Technologies Inc) and the quantity was measured by Qubit fluorometer (ThermoFisher Scientific). Barcoded libraries were sequenced using an Illumina HiSeq 4000 at 100 bp paired-end reads in the CAT genomic facility at UCSF.

## Patch-seq transcriptional analysis

(Raw reads were first processed with flexbar version 3.5.0 (https://github.com/seqan/flexbar; *Roehr, 2019*) to remove adapters specific to the NEBNext library prep, using parameters as described in https://github.com/nebiolabs/nebnext-single-cell-rna-seq (*Shtatland and Langhorst, 2018*). The reads were then processed with HTStream v.1.1.0 (https://github.com/s4hts/HTStream; *Hunter, 2019*) to perform data QA/QC, remove Illumina adapter contamination, PCR duplicates, and low-quality bases/sequences.

The trimmed reads were aligned to the *Drosophila melonogaster* genome v.BDGP6.22 (http://ensembl.org/Drosophila_melanogaster/Info/Annotation) with annotation release version 98 using the aligner STAR v. 2.7.0e (*Dobin et al., 2013*) to generate raw counts per gene. On average, 93.7% of the trimmed reads aligned to the *Drosophila* genome, and 80% of the trimmed reads uniquely aligned to an annotated *Drosophila* gene.

Differential expression analyses were conducted using limma-voom in R (limma version 3.40.6, edgeR version 3.26.7, R 3.6.1). Prior to analysis, genes with fewer than five counts per million reads in all samples were filtered, leaving 8598 genes. The differential expression analysis was conducted independently for the two experiments represented in the samples.

## qPCR

RNA was extracted from third-instar larval CNS or adult heads (5-7 days post-pupation) with RNeasy Plus Micro kit (Qiagen). RNA isolation was followed with DNase digestion with Turbo DNA-free (Ambion). For the first strand synthesis Super Script II RT was used (Invitrogen). Taqman Fast Universal PCR solution was mixed with TaqMan probe with an Applied Biosystems FAM dye. RPL32 was amplified as an internal control. Expression fold-changes are quantified by ddCT method. Data represent three biological and three technical replicates.

## Electron microscopy

Electron microscopy experiments were performed as previously described (*Harris et al., 2015*). For high-frequency stimulation experiments, larval fillet preparations were fixed immediately (1–5 s) following stimulation. Data are acquired from at least two animals.

## Statistical analysis of physiology and morphology data

Average values are presented as mean ± standard error of mean. All statistical tests are indicated in the figure legends, referring to individual panels within the figure. For multiple comparisons, we used one-way ANOVA, followed by Dunnett's or Tukey multiple comparisons. To test the difference between two groups, we used unpaired two-tailed Student's t-test. Pearson correlation coefficients were calculated following a linear-fit of the X-Y (quantal size vs. quantal content) data, although supra-linear best-fits are sometimes displayed, purely for the purpose of display.

## Acknowledgements

Supported by NINDS Grant (R35-NS097212) and Simons Foundation (SFARI #401636) to GWD, Simons Foundation (SFARI #402281) and NIMH (R01 MH110928) to SJS, and NRF-2017M3C7A1026959 to J-YA. We thank Matt State for comments and support and members of the Davis, State and Sanders labs for critical evaluation of the manuscript.

## Additional information

### Competing interests

Graeme W Davis: Reviewing editor, *eLife*. The other authors declare that no competing interests exist.

## Funding

| Funder | Grant reference number | Author |
| --- | --- | --- |
| National Institute of Neurological Disorders and Stroke | R35-NS097212 | Graeme W Davis |
| Simons Foundation | SFARI #401636 | Graeme W Davis |
| Simons Foundation | SFARI #402281 | Stephan J Sanders |
| National Institute of Mental Health | R01 MH110928 | Stephan J Sanders |
| Neurosciences Research Foundation | 2017M3C7A1026959 | Joon-Yong An |

The funders had no role in study design, data collection and interpretation, or the decision to submit the work for publication.

## Author contributions

Özgür Genç, Data curation, Formal analysis, Validation, Investigation, Visualization, Writing - review and editing; Joon-Yong An, Data curation, Formal analysis, Writing - review and editing; Richard D Fetter, Data curation, Formal analysis, Methodology, Writing - review and editing; Yelena Kulik, Giulia Zunino, Data curation, Formal analysis; Stephan J Sanders, Methodology, Writing - review and editing; Graeme W Davis, Conceptualization, Resources, Supervision, Funding acquisition, Visualization, Methodology, Writing - original draft, Project administration, Writing - review and editing

## Author ORCIDs

Özgür Genç https://orcid.org/0000-0003-0635-3192
Graeme W Davis https://orcid.org/0000-0003-1355-8401

## Decision letter and Author response

Decision letter https://doi.org/10.7554/eLife.55775.sa1
Author response https://doi.org/10.7554/eLife.55775.sa2

# Additional files

## Supplementary files

• Supplementary file 1. Supplementary Tables 1-7 are presented. Each table is referred to independently in the text.

• Transparent reporting form

## Data availability

Sequencing data have been deposited in GEO under accession code GSE153225. Analysis code is available via Github https://github.com/joonan30/Genc2020_RNAseq (copy archived at https://github.com/elifesciences-publications/Genc2020_RNAseq).

The following dataset was generated:

| Author(s) | Year | Dataset title | Dataset URL | Database and Identifier |
| --- | --- | --- | --- | --- |
| Genç O, An J-Y, Fetter RD, Kulik Y, Zunino G, Sanders SJ, Davis GW | 2020 | Transcriptomics analysis of heterozygous mutant and wild-type flies for presynaptic homeostatic plasticity | http://www.ncbi.nlm.nih.gov/geo/query/acc.cgi?acc=GSE153225 | NCBI Gene Expression Omnibus, GSE153225 |

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
