## [Decision Letter]

**Acceptance summary:**

The concept that *Drosophila* can contribute to the molecular underpinnings of some aspects of autism is exciting. The data that you present in the manuscript will therefore be of wide interest to researchers who study neurodevelopmental disorders like ASD as well as neuroscientists who use animal models to study synapse function.

**Decision letter after peer review:**

[Editors’ note: the authors submitted for reconsideration following the decision after peer review. What follows is the decision letter after the first round of review.]

Thank you for choosing to send your work, "Homeostatic plasticity commonly fails at the intersection of autism gene mutations and novel phenotypic modifiers", for consideration at *eLife*. Your submission has been assessed by a Senior Editor in consultation with a member of the Board of Reviewing Editors. Although the work is of significant interest and the manuscript is very ambitious, the reviewers felt that there were missing pieces of information and experiments that preclude acceptance at this stage. We know that you appreciate *eLife*'s policy to not ask for revisions that the reviewers feel would require more than two months of work. Consequently, we are declining this version at this time. Nonetheless, we would be willing to consider a resubmission at some later date if you feel that you can deal with these critiques.

Three very knowledgeable reviewers read your work and there is a clear consensus. They all think the manuscript has the backbone of a story that could be developed into an interesting *eLife* paper. but they all require a substantial number of experiments. The reviewers discussed your work and one reviewer nicely summarized it all as follows.

"Regarding additional experiments, by my reading, the requested additions would be:

1) Evaluations of the chronic maintenance of homeostatic plasticity (for several key genotypes). This was requested by two separate reviews. If the consensus ends up being to move forward with a revision request, we would need to have a discussion about which genotypes are important for the authors to examine.

2) The authors need to provide a much better evaluation of CREG gene function and its role in PHP as a potential homeostatic repressor. Two separate reviews requested overexpression of CREG to demonstrate its activity as a repressor. Two reviews requested more information of its loss-of-function phenotypes. Two reviews requested better information about its expression (especially at the larval stage where the homeostatic phenotypes are measured).

3) There also seems to be a consensus that the authors would need to either: 1) redo underpowered experiments that are important for main conclusions; 2) and/or conduct pairwise combinations that were excluded without explanation (Figure 1); 3) and/or providing an affirmative rationale for choosing to follow some pairwise combinations but not others.

The same reviewer also asked "tissue-specific expression information should be shared for new genes whose roles in PHP induction are delineated (*CHD8, ASH1L, CHD2, WDFY3, PPP2R5D, PDPK1, CREG*). The request in this case is informational (e.g. existing tissue-specific RNAseq data or in situ data), not a request to generate de novo transgenic lines for all of these genes. I did a brief search and found the expression pattern of nearly all genes : ASHIL homologue has a Pacman clone that is tagged with GFP (Venken et al., 2006); *CHD2* homologue has T2A-GAL4 (Lee et al., 2018; my web site); WDEY3 hom. has a T2A-GAL4 (same); *PPP2R5D* is tagged with GFP (Nagarkar-Jaiwal et al., 2015; my web site); *PDPK1* is tagged with T2A_GAL4 and GFP (same); CREG has a MiMIC (Venken et al., 2011). This means that there is a plethora of reagents that are easily accessible to determine expression and in some case protein localization. Many of these are also mutants and allow many elegant experiments that would answer the reviewer’s concerns.

Reviewer #1:

The manuscript by Genç et al. details a series of genetic interactions that affect homeostatic synaptic plasticity at the *Drosophila* NMJ. The first main finding is that a rim/+ genetic background (*RIMS1/+*) sensitizes the NMJ to heterozygous loss of autism-associated gene homologs in *Drosophila*. The authors detail how several double heterozygous genetic conditions abrogate an acute expression of presynaptic homeostatic plasticity (PHP).

The authors searched for additional interactors, screening *Drosophila* Chromosome III using Deficiency (Df) lines – all in a rim/+ genetic background. The second main finding is that several Df/+ conditions synergistically impair PHP with rim/+. The Df/+ conditions also synergistically impair PHP with heterozygous loss of autism-associated gene homologs. For two cases, the authors identified single genes (*PPP2R5D* and *PDPK1*) uncovered by Df chromosomes that phenocopied the Df/+ enhancement phenotype when mutated. For a *CHD8/+ PPP2R5D/+* double heterozygous condition, synaptic membranes appeared highly disorganized, and there are large synaptic vesicles.

Finally, the authors tested a hypothesis that some of these synergistic conditions could cause adaptive (or maladaptive) changes in gene transcription. Using RNAseq they found that CREG gene expression increased for several conditions, namely the *CHD8/+PPP2R5D/+* condition. When tested as a potential regulator of PHP, the authors found that a *CREG/+* heterozygous condition restored PHP to the impaired *CHD8/+PPP2R5D/+* genetic background. They concluded that CREG was a homeostatic repressor.

Main points:

1) PhTx (tested) vs. chronic glutamate receptor loss (untested): Recent papers in the field suggest that the acute expression of PHP and the chronic maintenance of PHP have important similarities and differences. This paper only tests acute expression of PHP, using PhTx as a homeostatic challenge. Yet there is no information about how NMJs of these genotypic backgrounds respond to chronic glutamate receptor subunit loss or knockdown.

There are many unknowns about how these genes might influence synaptic function. The authors correctly state that the effects on PHP could be due to indirect cellular stress or other mechanisms. In terms of organizing this new information and drawing more informed conclusions, tests of PHP maintenance for a subset of the genotypes would be helpful - especially since maintenance might reflect some long-term transcriptional changes. Some possible genotypes to test (given their importance in the paper):

CHD8/+; RIMS1/+

PPP2R5D/+; RIMS1/+

PPP2R5D/CHD8

CHD8/+; PPP2R5D/CREG

2) Expression. Several new genes are described that are relevant to presynaptic biology and PHP. It is not at all clear where or when these genes might be expressed – and how many of the PHP-blocking conditions might affect synaptic ultrastructure like the *CHD8/+ PPP2R5D/+.*

To the extent that it exists, tissue-specific expression information should be shared for new genes whose roles in PHP induction are delineated (*CHD8*, *ASH1L, CHD2, WDFY3, PPP2R5D, PDPK1, CREG*). The request in this case is informational (e.g. existing tissue-specific RNAseq data or in situ data), not a request to generate de novo transgenic lines for all of these genes.

Related: The authors seem to be arguing that CREG upregulation is a special maladaptive case that impairs PHP, at least in the case of the *CHD8/+, PPP2R5D/+* genetic background. There is a TRiP-overexpression (TOE) construct for CREG at Bloomington. One clear prediction from the paper is that if this TOE line were used to upregulate CREG (in the correct tissue) that it would block PHP – and maybe it would affect ultrastructure too?

3) Ultrastructure (Figure 6). These images are beautiful, but it is not clear what these phenotypes mean regarding PHP – or if these phenotypes might be generalizable to the other autism gene interactions. The large membrane structures are reminiscent of the large structures seen in endocytosis mutants. Endocytic mutants at the NMJ often have an electrophysiological signature (e.g. large minis). Were any recordings conducted post-stimulation? The baseline physiology for the *PPP2R5D/+; CHD8/+* condition (Supplementary file 1 table 7) does not look like an endocytic mutant, but maybe post-tetanus it would.

4) This reviewer found some claims that pieces of data provided the first evidence of a phenomenon, but the claims were not correct, even within the realm of the *Drosophila* NMJ itself. Specific Examples:

The idea that CHD8 had never before been connected to presynaptic biology is not right. At the *Drosophila* NMJ, there are well-documented defects for kismet mutants in both development and presynaptic neurotransmitter release.

The idea that CREG is the first-ever documented repressor of PHP in any system is not correct. Loss-of-function conditions at the NMJ have been documented that relieve blocks of PHP maintenance (two examples are loss of FasII and loss of 4E-BP). By the criteria set out by the current data set, these earlier examples would also constitute homeostatic repressors.

Reviewer #2:

The manuscript by Genc and colleagues examines the loss of presynaptic homeostatic plasticity (PHP) in heterozygous combinations as a model for polygenic disorders. Using the loss of PHP in trans-heterozygous combination as readout, the authors uncovered genetic interactions among unrelated autism genes (*RIMS1*, *CHD8, CHD2, WDFY3* and *ASH1L*) with fly orthologs. They expanded this strategy to screen for novel modifiers of PHP using the RIMS1/+ background and a series of overlapping deficiencies on 3rd chromosome. The screen identified modifiers in 20 lines (about 10% of the Df screened). Interestingly, some of these *RIMS1*- interacting Dfs also caused PHP defects when screened against the other autism genes (*CHD8, CHD2, WDFY3* and *ASH1L*). The authors mapped two of the interacting loci to a kinase and phosphatase, *PDPK1* and *PPP2R5D*, and confirmed that they interact genetically with most of the autism genes in this study. However, neither enzyme is required for PHP; also, the molecular basis for their observed interaction with *RIMS1* and other loci remains unknown. Finally, the authors searched for differentially expressed transcripts (using RNAseq from adult heads) and reported that CREG was elevated in *CHD8/+; PPPR25D/+* trans-heterozygotes. Importantly, lowering the dose of CREG rescued the PHP defects of *CHD8/+; PPPR25D/+*. How CREG restores the PHP in this trans-heterozygote and whether CREG is elevated in any of the other synthetic phenotypes remain open questions.

The study aims to model the genetic heterogenicity of autistic patients. The approach is sound, and the amount of data is substantial and of high quality, but the manuscript only tackles genetic interactions that the authors cannot explain nor connect to any previous PHP pathways. These numerous synthetic phenotypes seem merely "synthetic" without some anchoring into known phenomena; the same is true for the accompanying EM micrographs.

1) The description of genetic interaction concepts is unnecessarily long and convoluted and must be consolidated.

2) The screen itself is poorly described. Did the authors use the Bloomington Df kit for their screen? If not, specify. A table with the reagents used here should be provided.

3) The authors chose to compare a small and a large Df in lieu of examining additive effects for gene heterozygosity (Figure 3). These arguments are weak. Instead, a more detailed description of the screen results would be more meaningful and should be included here.

For example, how many of the Dfs tested contain known PHP modulators? What was the outcome of the screen for those Dfs? How does that outcome compare with prior single gene analyses?

4) When a Df covering a known PHP player did not show the expected interaction, what does this mean for the results of the screen?

5) It is somewhat anticlimactic that the *PDPK1* and *PPP2R5D* are not required for acute PHP. However, the authors hint to a possible role in chronic homeostasis, more similar to the mTOR signaling components. This probably has already been examined and should be included here.

6) The RNAseq analysis was performed with adult heads due to understandable limitations. But the qPCR validation of CREG up-regulation must be done in third instar larvae, at the stage when PHP is recorded.

7) The authors propose that CREG is a homeostatic repressor. This predicts that overexpression of CREG will block PHP. The authors should directly test this prediction which will strengthen their conclusions.

8) Reducing the dose of CREG would be expected to restore PHP in other trans-heterozygote combinations. This should be examined here since a positive result could expand the relevance of this finding beyond an isolated trans-heterozygous case presented here.

9) Another elegant solution to link the disconnected parts of this study would be to overexpress CREG in some of the single heterozygotes (*RIMS1*, *CHD8, CHD2, WDFY3*, *ASH1L* – or *PDPK1, PPP2R5D*) and block PHP.

10) Using qPCR to search for increased levels of CREG in (third instar) mutants lacking PHP would be an easy, fast way to implicate CREG in other known PHP pathways and will tremendously benefit the author's arguments.

Reviewer #3:

Genc et al. present in this study findings arguing that five seemingly unrelated autism associated genes are linked to presynaptic homeostatic plasticity (PHP) and synaptic transmission. They propose through genetic interaction studies in *Drosophila* that phenotypic enhancers could influence the robustness and phenotypic penetrance of these autism associated gene variants. Although several aspects of this study appear to support their conclusion, the reliability and reproducibility of their findings is hampered by conceptual and technical concerns. Addressing these concerns would help to strengthen the study.

1) The authors propose that five selected autism genes converge on PHP, but they only examine the genetic interaction between two pairs (*RIMS1* and *CHD8*) and (*RIMS1* and *ASH1L*). It is unclear why the authors do not complete the pairwise interaction studies for the remaining two genes. No rationale is provided to explain this incomplete analysis.

2) Due to the possibility of second site mutations contributing to the genetic interaction findings, it is important to assess other loss of function alleles of *RIMS1*, *CHD8*, and *ASH1L* in the genetic interaction studies. As well as for the remaining two selected autism genes. Furthermore, can these findings be ameliorated with genomic rescue of *RIMS1*, *CHD8*, *ASH1L*, etc in the pairwise genetic interaction studies?

3) What is the rationale for choosing only five of the 20 deficiencies that showed interactions with *RIMS1* to test their interactions with other gene? The authors need to provide a rationale for this selection. Otherwise it raises the question as to whether the remaining deficiencies were tested but did not show as strong of an interaction effect.

4) Critical resource and technical information is lacking, which make it difficult to interpret or reproduce the study in the future. In particular, fly stocks (origin, type of mutation, specific breakpoints, stock center, etc) was not reported. What were the specific deficiency lines – including the "20 small deficiencies that cause PHP to fail"?

5) It is often unclear in the manuscript and figures if the sample size "n" is referring to animals, neurons, synapses, vesicles, etc. The sample size is also very low (n of only three) for some of the studies, raising the concern as to whether the studies were adequately powered to detect a statistical difference. For example, "we performed multiple (n=3-15) intracellular recordings".

6) The authors argue that CREG is a homeostatic repressor that blocks PHP and regulates synaptic ultrastructure (Figure 8). It would be important to show the phenotype of the CREG single mutant. What is the ultrastructural phenotype of the CREG single mutant?

7) EM studies are performed in tissue fixed 1-5 seconds after high frequency stimulation with a sample size as low as only 2 animals. How does the difference between 1 to 5 seconds post-high frequency stimulation affect the findings?

8) The rationale for selecting *RIMS1*, *CHD8*, *CHD2*, *WDFY3*, and *ASH1L* is still unclear. The authors state they were selected as they were "category 1 or 2 ASD-associated genes based on SFARI Gene". However these two categories encompass many more than these five genes. Were these the only genes with fly orthologs and available resources? Were there other reasons for choosing these genes?

9) The authors need to be careful not to overstate or misstate their findings. Particularly as the premise of the paper hinges upon 5 selected genes associated with autism, and far more than 5 genes are implicated in autism. Providing a better rationale for selecting these 5 genes would help frame their findings better.

[Editors’ note: further revisions were suggested prior to acceptance, as described below.]

Thank you for submitting your article "Homeostatic plasticity fails at the intersection of autism-gene mutations and a novel class of common genetic modifiers" for consideration by *eLife*. Your article has been reviewed by three peer reviewers, and the evaluation has been overseen by Hugo Bellen as the Reviewing Editor and Ronald Calabrese as the Senior Editor. The following individual involved in review of your submission has agreed to reveal their identity: C. Andrew Frank (Reviewer #1).

The reviewers have discussed the reviews with one another and the Reviewing Editor has drafted this decision to help you prepare a revised submission.

Another issue that was raised and all reviewers agreed upon is that one cannot tell which 18-20 interacting Dfs fall into the gray shaded area of Figure 2B. This would be a matter of a text revision or a table that indicates something about the physiological data of those Dfs identified in the screen. Specifically, save a few labeled examples, the data points in the figures do not actually match up with specific Dfs, and therefore, it is difficult to interpret the findings in the context of what was expected/unexpected a priori. To address this, the authors need to do a couple of things. First, they would need to publish a long supplemental table or Excel file for all 168 screened genotypes. The type of data expected are: genotype, average mini amplitude, average evoked amplitude, calculated quantal content, and n. This should not be too onerous to do (depending on how the data are currently organized).

The reviewers agreed that the main message, combinatorial loss of autism-associated genes can result in profound synaptic phenotypes, including blocked PHP, is valuable. This is an interesting and novel idea in itself, and worthy of publication in *eLife*.

Below are issues that all can be addressed in a timely fashion.

Summary and General Evaluation

Complex neurodevelopmental disorders like Autism Spectrum Disorder (ASD) are not well understood, either on genetic and neurophysiological levels. The study by Genç et al. uses a genetic model to tackle this knowledge gap. The study uses the fruit fly *Drosophila melanogaster* and its model synapse, the neuromuscular junction (NMJ). The core finding is clear: in *Drosophila*, double heterozygous combinations of ASD-related genes can manifest in striking synaptic phenotypes. Moving forward, it stands to reason that genetic combinations in *Drosophila* system (and other models) could be exploited, potentially to understand biology underlying ASD and other neurodevelopmental disorders.

The work reports novel scientific findings. First, double heterozygous losses of *Drosophila* homologs of high confidence ASD genes impair a form of homeostatic synaptic plasticity, called presynaptic homeostatic potentiation (PHP). Second, heterozygous Deficiency (Df) lines synergistically cause PHP to fail when combined with a rim/+ (*RIMS1/+*) genetic background; a subset of those Df lines synergistically interact with other heterozygous ASD gene combinations to impair PHP expression. Third, the authors identify two genes – *PPP2R5D* and *PDPK1* – uncovered by two of the Df chromosomes that interacted with *RIMS1/+*. Heterozygous loss of *PPP2R5D* and *PDPK1* phenocopy the corresponding Df/+ condition. Moreover, heterozygous loss-of-function mutations in *PPP2R5D* and *PDPK1* interact with other heterozygous ASD gene homologs, again resulting in instances of PHP failure.

The authors follow one genetic combination as an example. By doing this, they looked deeper, on a level of cell biology (by EM) and on a level of gene transcription (by RNAseq), and they complemented that work with electrophysiological analyses. For the *CHD8/+ PPP2R5D/+* double heterozygous combination, EM shows that synaptic membranes at the NMJ are highly disorganized, there are large synaptic vesicles, and there is clear evidence of endocytic defects both visually and by electrophysiology. Using RNAseq, the authors find that CREG is upregulated in the *CHD8/+ PPP2R5D/+* condition. They test the idea that CREG upregulation is maladaptive for the synapse. Partial loss of CREG reverses the electrophysiology and EM phenotypes of the *CHD8/+ PPP2R5D/+* genetic combination. Consistently, transgenic overexpression of CREG blocks PHP, and the authors conclude that CREG is a novel suppressor of homeostatic plasticity.

A prior review requested testing the chronic maintenance of PHP vs. the acute induction of PHP.

The paper examines the acute induction of PHP. For the prior submission, this reviewer offered that examining the chronic maintenance of PHP would be helpful – especially since ASD-related mutations could represent modeling a chronic condition. The authors made a counterargument that the PHP induction paradigm is a stress test of sorts – and their core findings remain robust with this test, even in the absence of testing long-term maintenance. This is well reasoned. The authors correctly state that testing the maintenance of PHP could be a logical extension of the work, but the subject of a different study. This is a point added to the text of the article.

The prior review requested more evaluation of CREG gene function as a novel homeostatic repressor.

The set of experiments pursued by the authors is very nicely done and firmly establish CREG as a homeostatic repressor at the NMJ. They also implicate CREG upregulation in the EM endocytic phenotypes.

The prior review requested more complete analyses of the possible ASD-related gene combinations.

The authors have responded to this request experimentally. The main new work is summarized in Figure 1—figure supplement 2, which summarizes all of the pairwise one-gene combinations of ASD-related heterozygous conditions.

The prior review requested expression data for new genes characterized in this study.

The authors responded to this request in a creative way, by performing a PatchSeq analysis and examining gene expression in motor neurons that innervate larval body wall muscles. The results are consistent with the electrophysiology: both types of motor neurons tested express the genes tested in the study, and at similar levels. Those data are summarized in Figure 1—figure supplement 1. Future studies can use this as a starting point to interrogate the roles that those genes and their gene products might play in motor neurons. Roles in other tissue types are also possible.

Revisions required for this paper:

1) Clarify or modify broad statements about lack of information or claims of "novelty", that falters in the context of existing information in the literature. The study will be strengthened by building connections with existing knowledge, which will minimize the impression that these findings are "synthetic" without anchoring in known phenomena.

a) In discussing the random selection of *RIMS1, CHD8, CHD2, WDFY3*, and *ASH1L* the authors state that there are no known biochemical and genetic interactions among these five genes in *Drosophila*, and no known direct biochemical interactions in humans. However, it is curious that the authors fail to also state the potential for interactions between these genes by incorporating many findings in mouse models indicating a potential convergence between these genes on neurodevelopment and function. Presenting this at the outset will further strengthen the study by utilizing prior literature to support the probability of identifying disease relevant genetic interactions.

b) By extension, the known disease associations for these genes should be presented and referenced, as well as if they are all impacting similar neurologic functions in humans. All five genes have known disease associations: OMIM: *RIMS1* (#603649, Cone-rod dystrophy type 7), *CHD8* (#615032, Susceptibility to autism type 18), *CHD2* (#615369, Epileptic encephalopathy childhood onset), *WDFY3* (#617520, Primary microcephaly type 18), and *ASH1L* (#617796, autosomal dominant mental retardation type 52).

c) Heterozygous loss of function mutations in *CHD8, CHD2, WDFY3*, and *ASH1L* all cause neurodevelopmental disorders, including autism, epilepsy, and microcephaly. In the context of the human data, the authors' finding that PHP in the single heterozygous flies is normal suggests that PHP may not be the most important mechanism driving autism phenotypes in neurodevelopmental disorders. In contrast, although pathogenic variants in *RIMS1* have only been reported with a photoreceptor phenotype in humans, there is a recent publication by Peter et al., 2019 (PMID 30949922) presenting evidence in humans for pleiotropic and complex effects involving *RIMS1* in sporadic autism spectrum disorder. Therefore, the authors can consider strengthening the conceptual impact of their study from the perspective that variants in *RIMS1* may increase the susceptibility to autism spectrum disorder but require other polygenic factors to manifest the disorder.

d) The authors state that the "five ASD gene orthologs do not share any known common biological activity. Therefore, the identification of common genetic modifiers is completely unexpected." This is a confusing statement to make given that CHD8 and CHD2 are in the same subfamily of chromodomain helicase DNA-binding proteins, indicating a common biological activity, and ASH1L localizes to the chromatin. Based on these common functions and subcellular localization it is not entirely unexpected to find common genetic modifiers. These statements should be revised. Noting the potential for common biological activity will not detract from the potential impact of the authors' findings of common genetic modifiers through their screening approach.

e) The authors state that "there is no clear connection, biochemically or genetically, to the role of *PDPK1* or *PPP2R5D* in the nervous system." The authors need to clarify this statement as there are rodent models demonstrating a genetic role for *PDPK1* and PPP2RD5 in the nervous system. There is altered brain development in *PDPK1* KO mice (Lawlor et al., EMBO J, PMID 12110585) and abnormal tau pathology in the brains of PPP2RD5 KO mice (Louis et al., PNAS, PMID 21482799). Based on the literature there is clear connection for a role of *PDPK1* and *PPP2R5D* in the nervous system.

f) The authors propose that upregulation of CREG in *PPP2R5D/+;CHD8/+* underlies the failure of PHP, which is further supported by the complete block of PHP when CREG is overexpressed ubiquitously or in motor neurons. They identified two CREG mutant alleles with differential effects on normalizing CREG levels in Figure 8, panel D (m2 restored the double mutant to wildtype, but not m1) suggesting that m2 is possibly a stronger loss of function allele. But surprisingly, m2 seems to have a reduced rescue effect compared to m1 (Figure 8, panel F). Is there a difference in CREG expression level between these two alleles? The authors only report the m1 transposon completely abolishes CREG expression and do not show the data. They do not report the effect of the m2 transposon. The effect on CREG expression from both alleles should be examined and shown as this may indicate a dosage sensitivity for CREG.

g) A rationale for selecting only 5 of the 20 deficiencies showing interactions with *RIMS1* is still lacking. Were they selected based on the type of genes in the deficiencies (brain specific in mammals), known or unknown function of the genes, prior implications in homeostatic plasticity, availability of reagents, strength of the interaction effect, etc?

2) Technical concerns

a) There appears to be incomplete presentation of some datasets. The authors report that "altered NMJ growth was not commonly observed in the majority of genetic interactions tested (Figure 5—figure supplement 1)", implying that multiple pairwise interactions were examined. In fact, this is not the case. Figure 5—figure supplement 1 shows NMJ morphology data for wildtype, single heterozygotes for *PDPK, PPP2R5D, CHD8, CHD2, ASH1, WDFY3*. The only genetic interactions shown were for *CHD8* with *PDPK1* and *CHD8* with *PPP2R5D*. The data actual shows an increase in bouton number for the *CHD8/+;PPP2R5D/+* genotype compared to wildtype and *PPP2R5D/+* alone, it did not reach statistical significance compared to *CHD8/+* alone but perhaps this is due to the sample size? The data for the other pairwise genetic interactions should also be included in order for the authors to make the statement that altered NMJ growth was not commonly observed.

b) Sample size still remains unclear for Figures 5—figure supplement 2, Figure 5—figure supplement 3, Figure 8F, Figure 8K. Please clarify the sample size so that the interpretation and reproducibility of the data is improved.

c) In Figure 8, what does the sample size "n" in panels H and I represent? The authors note in the figure legend that "individual data points shown for indicated genotypes". Clarify if the individual data points represent individual vesicles, average vesicle size per synapse, or average vesicle size per animal. How many synapses were quantified per animal? How many animals per genotype?

d) Similarly, in Figure 6, what is the sample size representing? Individual vesicles, average size of vesicles per boutons, per animal? How many animals and synapses per animal were assessed?

e) It is important to provide all the necessary information about reagents and resources used for rigor and reproducibility. The authors now include a table S1 showing all of the deficiency stocks that were used in the study. But it remains unclear which of the listed 168 deficiency stocks were the "20 small deficiencies that cause PHP to fail when combined with *RIMS1/+*". The authors provide a schematic showing the distribution of these deficiencies in Figure 2C, but it is important to also denote in Table S1 the stock ID number corresponding to these 20 small deficiencies. This should be very straightforward to indicate in table S1.

f) The authors assessed the consequences of CREG heterozygous and homozygous loss of function mutations on baseline neurotransmission and PHP and describe the findings in the text. This is an important control data. For comparison to the other genotypes it would be helpful to also include the data graphically in Figure 8, panel F.

g) You stated: "First, we present every single deficiency in the graph (Figure 3). We provide representative examples of one large and one small deficiency. The reviewer seems to have misread the figure and our text in this regard." The reviewers did not find the data, legend or text presenting the interacting deficiencies. Figure 3 has no such graph. They are wondering whether you refer to the drawing in Figure 2C. Do the authors imply that this is the summary result that presents every single deficiency? Are we missing something? Moreover, the stock numbers for the Bloomington deficiencies provided in the table do not match the ones described in the text.

Information is needed for why data sets are not shown (ie CREG expression level in m1 allele), clarity on what the sample size represents in multiple datasets, details as to how many synapses, boutons, animals are included in the NMJ and EM datasets.

---

## [Author Response]

[Editors’ note: the authors resubmitted a revised version of the paper for consideration. What follows is the authors’ response to the first round of review.]

Three very knowledgeable reviewers read your work and there is a clear consensus. They all think the manuscript has the backbone of a story that could be developed into an interesting eLife paper. but they all require a substantial number of experiments. The reviewers discussed your work and one reviewer nicely summarized it all as follow."Regarding additional experiments, by my reading, the requested additions would be:1) Evaluations of the chronic maintenance of homeostatic plasticity (for several key genotypes). This was requested by two separate reviews. If the consensus ends up being to move forward with a revision request, we would need to have a discussion about which genotypes are important for the authors to examine.

We respectfully disagree. We pioneered the field of presynaptic homeostatic plasticity and are well aware of the emerging distinctions between the acute induction of PHP and the effects induced by a genetic mutation in the glutamate receptor (Frank et al., 2006; Frank et al., 2009). Indeed, in a recent publication, we delineated a bifurcating signaling cascade that controls both the rapid induction and long-term expression of homeostatic plasticity (Harris et al., 2017). However, we would like to point out that the injection of PhTx into *Drosophila* larvae was originally performed and demonstrated to induce a lasting homeostatic effect, persisting for many hours (Frank et al., 2006). It remains unclear how PhTx-induced homeostatic plasticity relates to *GluRIIA* induced homeostatic plasticity. One might be an extension of the other, or they might be independent effects with over-lapping mechanisms.

In justifying a request for substantial addition of new data (analysis of new triple and quadruple mutants and analyses thereof) the reviewers argue that GluRIIA-induced PHP is most relevant to autism. We disagree for the following reasons:

a) Currently, there has yet to be a clear experimental link between PHP and the mechanisms of autism. Thus, we have already made an advance and it was achieved systematically across many genotypes.

b) We use the acute induction of PHP to probe the capacity of the nervous system to respond to a homeostatic challenge. This is the sole purpose of our study. Consider the following analogy: It is common for physicians to diagnose pre-diabetes by challenging a patient with a glucose tolerance test. This is an acute perturbation, and the physician determines whether the individual is capable of robust, glucose homeostasis. This is precisely the logic that we use. Can the nervous system respond to an acute challenge with a robust homeostatic response? This could reasonably be directly relevant to ASD.

c) Is the *GluRIIA* mutation more directly relevant to ASD, as argued by the reviewers? To our knowledge, there is no evidence for a systematic change in AMPA receptor abundance in the ASD brain (changes do occur in fragile-X animals, but not generally across many different models). So, the argument that there is some unique importance of using the *GluRIIA* mutation is not well substantiated. Indeed, a counter argument is possible. Because the *GluRIIA* mutant is present throughout neural development, there is the very real possibility of creating effects in the GluRIIA-ASD-modifier triple mutants that are unique and not relevant to ASD. By using an acute perturbation, any developmental effects are restricted to the ASD orthologs and newly identified modifiers that we are studying.

d) We generally agree that the *GluRIIA*-dependent mechanisms of PHP could be a logical, future extension of our work. But, any such analysis should be performed in the same systematic manner as performed here – a thorough interrogation of many dozens of different genotypes, clearly beyond the scope of our current paper.

e) Finally, we have added two paragraphs to our Discussion regarding the difference between the acute induction and long-term expression of homeostatic plasticity. The paragraphs are copied here:

“The rapid induction versus long-term expression of PHP

There are two well-established methods to induce expression of PHP. Application of PhTx induces PHP within minutes, a process that can be maintained for hours (Frank et al., 2006). […] In the future, it will be interesting to systematically determine whether the gene-gene interactions identified here also uniformly perturb PHP induced by the *GluRIIA* mutation. However, such an analysis is beyond the scope of the present study.”

2) The authors need to provide a much better evaluation of CREG gene function and its role in PHP as a potential homeostatic repressor. Two separate reviews requested overexpression of CREG to demonstrate its activity as a repressor. Two reviews requested more information of its loss-of-function phenotypes. Two reviews requested better information about its expression (especially at the larval stage where the homeostatic phenotypes are measured).

We thank the reviewers for prompting us to do these experiments. We have addressed all three requests for additional information.

a) We have generated the necessary transgenic animals and performed the requested analyses. Overexpression of CREG, either specifically within motoneurons or ubiquitously, blocks homeostatic plasticity without affecting baseline transmission. The new data are added to Figure 8.

b) We have added an analysis of the loss of function mutation, focusing on the mutation that abolishes CREG expression. There is no effect on baseline transmission or PHP. The new data are added to the text.

c) We have added an extensive amount of information regarding CREG expression including assessing CREG expression in motoneurons at four stages of neural development and assessing the up-regulation of CREG in larval motoneurons. These data have been added as a new supplemental figure (new Figure 8—figure supplement 1).

3) There also seems to be a consensus that the authors would need to either: 1) redo underpowered experiments that are important for main conclusions; 2) and/or conduct pairwise combinations that were excluded without explanation (Figure 1); 3) and/or providing an affirmative rationale for choosing to follow some pairwise combinations but not others.

We have performed additional experiments of pairwise gene-gene combinations as requested regarding the data originally presented in Figure 1. We now demonstrate that *RIMS1/+* interacts with *CHD2/+* and *ASH1L/+* but not with *WDFY3/+*. These data support our existing models and greatly strengthen the paper. We thank the reviewers for prompting us to do this work. The data are now presented as a new supplemental figure (new Figure 1—figure supplement 2).

Comment 3, continued: “redo underpowered experiments”

The comment regarding sample size is unfair. The reviewer quotes a statement that specifically refers to our screen (for which no statistical comparisons are made between genotypes) – yet the reviewer implies that this statement reflects upon other data in our paper (which is factually incorrect). We stated in our original paper:

“The screen that we performed is diagrammed in Figure 2A. We took advantage of a collection of small chromosomal deficiencies (5-50 genes per deficiency, each with known chromosomal breakpoints; listed in Supplemental Table 1) that tile the 3rd chromosome, uncovering approximately 6000 genes in total. For every double heterozygous combination of *RIMS1/+* with a heterozygous deficiency, we performed multiple (n=3-15) intracellular recordings, quantifying mEPSP amplitude, EPSP amplitude, quantal content (EPSP/mEPSP), resting membrane potential and input resistance.”

A genetic screen, such as ours, is designed to identify “hits” as efficiently as possible. In our genetic screen, as is common practice, we do not make any claims regarding the statistical validity of any individual observation. The only purpose of a screen is to identify “hits”. We establish well-justified parameters to select “hits”. This is done in a conservative fashion in order to protect the experimenter from performing un-necessary follow-up work. Importantly, every “hit” is re-validated. We are now more explicit, stating, “To achieve a final list of 20 hits, each potential hit was re-validated in a

second set of experiments, increasing sample sizes (generally 7-16 NMJ). During the process of revalidation, we rule out approximately one third of the potential hits selected from the screen.”

Furthermore, the reviewer is referred to the fact that multiple “hits” from our screen were re-validated, multiple times throughout our manuscript. First, the hits were identified. Second, they were revalidated as a hit. Third, five of the deficiencies were incorporated into a matrix of multiple gene-gene interactions, a further re-validation. Finally, two deficiencies were resolved to individual genes, requiring multiple rounds of re-validation. And, the single genes precisely recapitulate the results of the deficiencies in which they originally resided. Surely, at this point, the extensive nature of revalidation, with statistically relevant sample sizes at every stage, should inspire statistical confidence.

How did we determine adequate sample sizes? We performed a power analysis to estimate necessary sample size. We have an effect size of 150-200% with a low standard deviation. Estimation of sample size suggests a minimal sample size of three to four, which is obviously very small. In our primary screen, when efficiency is emphasized, we use a sample size of 3-15, as reported. None of the data from the screen are compared statistically – the use of sample sizes here is simply for transparent reporting. In our study, when statistical comparisons are made, sample sizes were never less than n=7. In instances where effect size was on the smaller end of the range, sample sizes are generally larger to ensure confidence.

In our study, we analyze more than 40 genotypes electrophysiologically, under two conditions (with and without PhTX), representing more than 80 experimental conditions – not including our screen. In only two experiments out of more than 80 experimental conditions are sample sizes equal to 4 (in the original submission, there were four instances, and new data have been incorporated). These instances were controls that adhered to high effect size and low variance. In every other experiment (>80 conditions) sample size was never less than 7, generally in excess of 10 and often greatly exceeding these sizes. All the data are transparently presented in figures with individual data points and in tables with sample sizes. In every instance, sample sizes were sufficiently powered and all statistical tests and values reported.

Comment 3, “provide affirmative rationale for choosing to follow some pairwise combinations but not others”

If we understand this criticism correctly, the reviewer seems to suggest that we follow every single double heterozygous combination (36 were systematically tested), a request that seems out of line with the existing literature and reviewing standards.

It is possible that the reviewer simply wants additional justification for why we chose to follow the *CHD8/+; PPP2R5D/+* double heterozygous mutant. It was a highly penetrant double mutant and CHD8 is one of the most well-established risk factors for ASD. As we now state in our updated text:

“It is rare for a genetic study to define, precisely, how a double heterozygous interaction creates a synthetic phenotype if the two genes do not encode proteins that biochemically interact. Simply put, there are a vast number of possible mechanisms by which SSNC could occur (Yook et al., 2001). None-the-less, we attempted to do so for at least one double heterozygous combination. Although this represents only a single mechanism of SSNC, it could provide proof of principle for how PHP is affected in other ASD gene interactions. We chose the genetic interaction of *PPP2R5D/+ with CHD8/+*. This combination was chosen because *CHD8* is among the most common ASD de novo gene mutations. Furthermore, the genetic interaction is highly penetrant.”

Reviewer #1:The manuscript by Genç et al. details a series of genetic interactions that affect homeostatic synaptic plasticity at the *Drosophila* NMJ. The first main finding is that a rim/+ genetic background (RIMS1/+) sensitizes the NMJ to heterozygous loss of autism-associated gene homologs in *Drosophila*. The authors detail how several double heterozygous genetic conditions abrogate an acute expression of presynaptic homeostatic plasticity (PHP).The authors searched for additional interactors, screening *Drosophila* Chromosome III using Deficiency (Df) lines – all in a rim/+ genetic background. The second main finding is that several Df/+ conditions synergistically impair PHP with rim/+. The Df/+ conditions also synergistically impair PHP with heterozygous loss of autism-associated gene homologs. For two cases, the authors identified single genes (PPP2R5D and PDPK1) uncovered by Df chromosomes that phenocopied the Df/+ enhancement phenotype when mutated. For a CHD8/+ PPP2R5D/+ double heterozygous condition, synaptic membranes appeared highly disorganized, and there are large synaptic vesicles.Finally, the authors tested a hypothesis that some of these synergistic conditions could cause adaptive (or maladaptive) changes in gene transcription. Using RNAseq they found that CREG gene expression increased for several conditions, namely the CHD8/+ PPP2R5D/+ condition. When tested as a potential regulator of PHP, the authors found that a CREG/+ heterozygous condition restored PHP to the impaired CHD8/+ PPP2R5D/+ genetic background. They concluded that CREG was a homeostatic repressor.Main points:1) PhTx (tested) vs. chronic glutamate receptor loss (untested): Recent papers in the field suggest that the acute expression of PHP and the chronic maintenance of PHP have important similarities and differences. This paper only tests acute expression of PHP, using PhTx as a homeostatic challenge. Yet there is no information about how NMJs of these genotypic backgrounds respond to chronic glutamate receptor subunit loss or knockdown.There are many unknowns about how these genes might influence synaptic function. The authors correctly state that the effects on PHP could be due to indirect cellular stress or other mechanisms. In terms of organizing this new information and drawing more informed conclusions, tests of PHP maintenance for a subset of the genotypes would be helpful -especially since maintenance might reflect some long-term transcriptional changes. Some possible genotypes to test (given their importance in the paper):CHD8/+; RIMS1/+PPP2R5D/+; RIMS1/+PPP2R5D/CHD8CHD8/+; PPP2R5D/CREG

We respectfully disagree with this reviewer see above for our response to major comment 1.

2) Expression. Several new genes are described that are relevant to presynaptic biology and PHP. It is not at all clear where or when these genes might be expressed – and how many of the PHP-blocking conditions might affect synaptic ultrastructure like the CHD8/+ PPP2R5D/+.To the extent that it exists, tissue-specific expression information should be shared for new genes whose roles in PHP induction are delineated (CHD8, ASH1L, CHD2, WDFY3, PPP2R5D, PDPK1, CREG). The request in this case is informational (e.g. existing tissue-specific RNAseq data or in situ data), not a request to generate de novo transgenic lines for all of these genes.

The reviewers asked that we determine the expression of each of the genes that we analyze in our manuscript, inclusive of seven genes. The reviewers suggested a number of resources including transposable element-mediated gene tags, some of which remain un-validated. We reasoned, however, that the only way that we could quickly and definitively address this comment was to pursue a Patch-Seq gene-expression profiling experiment, assaying gene expression selectively in the two cells that we analyze electrophysiologically.

We performed Patch-Seq experiments analyzing gene expression in type 1b and type 1s motoneurons. We demonstrate that all of the genes studied in our manuscript are expressed in both of these motoneuron types (at very similar levels). These data are now included as a new supplemental figure (Figure 1—figure supplement 1).

There is added significance to the addition of these new data. We recently demonstrated that homeostatic plasticity is globally induced at all connections to muscle 6, but the expression of homeostatic plasticity can vary depending upon the experimental conditions. Some conditions favor expression by type 1b motoneurons and some by type 1s motoneurons (Genc et al., 2019). By demonstrating that all ASD genes queried in our study are expressed by both motoneurons, it seems likely that genetic mutations will perturb both cells. Indeed, the expression levels are remarkably similar comparing MN1s and MN1b.

(continued)…how many of the PHP-blocking conditions might affect synaptic ultrastructure like the CHD8/+ PPP2R5D/+.

This reviewer seems unaware of the low-throughput nature of electron microscopy. We originally provided data for two controls, two heterozygous mutants, a double mutant, and the triple heterozygous mutant combination (an experiment that included its own independent control).

Furthermore, we now have added three more genotypes to our EM analysis based on this request. Finally, the reviewer should also note that we now include an electrophysiological analysis to support our conclusions made based on electron microscopy (see below).

Electron microscopy is not a screening tool. The HHMI funded laboratory of Dr. Bellen has been able to fully support such an effort in the past, but that is both remarkable and almost completely unique. Please note that the majority of papers examining homeostatic plasticity at the NMJ do not include quantitative EM, inclusive of papers recently published in *eLife* by other laboratories.

Related: The authors seem to be arguing that CREG upregulation is a special maladaptive case that impairs PHP, at least in the case of the CHD8/+, PPP2R5D/+ genetic background. There is a TRiP-overexpression (TOE) construct for CREG at Bloomington. One clear prediction from the paper is that if this TOE line were used to upregulate CREG (in the correct tissue) that it would block PHP – and maybe it would affect ultrastructure too?

This is a reasonable prediction and something that we have been able to test directly. We have added two types of data to our manuscript that address this issue:

a) We now present the developmental expression profile for *CREG* in motoneurons at four time points including the embryo, 1^st^ instar, 2^nd^ instar, early third instar and late third instar. The data demonstrate that *CREG* is expressed highly in the embryo and then is turned off. However, in the double heterozygous mutant combination, *CREG* is induced at the late third instar stage (4-fold induction). These new data clearly validate our assertion that *CREG* is up-regulated in a specific manner in the double mutant.

b) We generated a *UAS-CREG* transgenic line to drive CREG expression in a tissue-specific manner. Over-expression of *CREG* either ubiquitously (*Tubulin-GAL4*) or specifically in motoneurons (*OK371-GAL4*) is sufficient to completely block homeostatic plasticity. We did not perform additional ultrastructure.

3) Ultrastructure (Figure 6). These images are beautiful, but it is not clear what these phenotypes mean regarding PHP – or if these phenotypes might be generalizable to the other autism gene interactions. The large membrane structures are reminiscent of the large structures seen in endocytosis mutants. Endocytic mutants at the NMJ often have an electrophysiological signature (e.g. large minis). Were any recordings conducted post-stimulation? The baseline physiology for the PPP2R5D/+; CHD8/+ condition (Supplementary file 1 table 7) does not look like an endocytic mutant, but maybe post-tetanus it would.

There are two points that we would like to make.

a) We now include an entire figure in which we analyze synaptic rundown, a parallel test of a defect in vesicle endocytosis. These data are presented in new Figure 6—figure supplement 1. The data argue in favor of an endocytic defect, as reported by EM.

b) We agree that the EM effect may or may not be directly related to homeostatic plasticity. Nonetheless, the EM provides a striking visual demonstration of the effects of the double mutant combination. And, it is clearly a consequence of increased CREG in the double mutant context based on the rescue of the EM in the triple mutant. We believe that these data will be important for the field, as it might encourage those laboratories studying synapses in the rodent central nervous system to examine synaptic ultrastructure. We state in our text:

“These data provide a striking visual confirmation of the genetic interaction between *PPP2R5D/+* and *CHD8/+*. And, this is further evidence linking the action of a chromatinremodeling factor (*CHD8*) to the stability of synaptic transmission.”

4) This reviewer found some claims that pieces of data provided the first evidence of a phenomenon, but the claims were not correct, even within the realm of the *Drosophila* NMJ itself. Specific Examples:The idea that CHD8 had never before been connected to presynaptic biology is not right. At the *Drosophila* NMJ, there are well-documented defects for kismet mutants in both development and presynaptic neurotransmitter release.

*Kismet* (*Drosophila* ortholog of CHD8) has been studied previously by gene knockdown, reducing expression levels below 50%. No phenotype has been reported in the heterozygous mutation to our knowledge. Regardless, we have removed the offending text as it was un-necessary.

The idea that CREG is the first-ever documented repressor of PHP in any system is not correct. Loss-of-function conditions at the NMJ have been documented that relieve blocks of PHP maintenance (two examples are loss of FasII and loss of 4E-BP). By the criteria set out by the current data set, these earlier examples would also constitute homeostatic repressors.

The reviewer is correct that over-expression of a specific isoform of Fas2 has been demonstrated to disrupt PHP. We now cite this paper.

Reviewer #2:The manuscript by Genc and colleagues examines the loss of presynaptic homeostatic plasticity (PHP) in heterozygous combinations as a model for polygenic disorders. Using the loss of PHP in trans-heterozygous combination as readout, the authors uncovered genetic interactions among unrelated autism genes (RIMS1, CHD8, CHD2, WDFY3 and ASH1L) with fly orthologs. They expanded this strategy to screen for novel modifiers of PHP using the RIMS1/+ background and a series of overlapping deficiencies on 3rd chromosome. The screen identified modifiers in 20 lines (about 10% of the Df screened). Interestingly, some of these RIMS1- interacting Dfs also caused PHP defects when screened against the other autism genes (CHD8, CHD2, WDFY3 and ASH1L). The authors mapped two of the interacting loci to a kinase and phosphatase, PDPK1 and PPP2R5D, and confirmed that they interact genetically with most of the autism genes in this study. However, neither enzyme is required for PHP; also, the molecular basis for their observed interaction with RIMS1 and other loci remains unknown. Finally, the authors searched for differentially expressed transcripts (using RNAseq from adult heads) and reported that CREG was elevated in CHD8/+; PPPR25D/+ trans-heterozygotes. Importantly, lowering the dose of CREG rescued the PHP defects of CHD8/+; PPPR25D/+. How CREG restores the PHP in this trans-heterozygote and whether CREG is elevated in any of the other synthetic phenotypes remain open questions.The study aims to model the genetic heterogenicity of autistic patients. The approach is sound, and the amount of data is substantial and of high quality, but the manuscript only tackles genetic interactions that the authors cannot explain nor connect to any previous PHP pathways. These numerous synthetic phenotypes seem merely "synthetic" without some anchoring into known phenomena; the same is true for the accompanying EM micrographs.

First, we thank the reviewer for commenting that our approach is sound. However, we find the subsequent criticism to be quite strange. We have pioneered something new in several respects. 1. We have determined precisely how combinations of heterozygous mutations can cause a synthetic disruption of PHP. This demonstration is proof of principle that double heterozygous gene-gene interactions could have implications for the etiology of ASD (or other genetic disorders of the nervous system). 2. We present evidence that homeostatic plasticity can fail at the intersection of two heterozygous genetic mutations without either gene being a core component of the homeostatic machinery. The data support the conclusion that the ASD orthologue mutation sensitizes the homeostatic signaling system to fail, and the modifier mutation exacerbates the effect, causing PHP to fail. A normally robust process, PHP, is rendered fragile and then fails. This is a fundamentally different concept compared to the standard search for core molecular mechanisms.

This is new territory and we suggest the following parallel. Anyone working in a rodent model system knows that a phenotype can be dramatically enhanced or suppressed by moving a mutation onto a new genetic background. It is very rare, but not unprecedented, to determine the mechanism by which a genetic background influences phenotypic penetrance. When successful, the discoveries are quite unexpected. This is effectively what we are doing by systematically testing double heterozygous mutant combinations. It stands to reason that the mechanisms we identify will be new. This possibility is further underscored by two studies in yeast, where systematic double heterozygous mutant combinations were tested (see our Introduction). Gene-gene interactions were identified that were entirely unpredicted. Given that ASD gene mutations are often heterozygous loss of function mutations, they will likely interact with standing (heterozygous) genetic variation in the human genome, unique to each individual. The reviewer is referred to the following papers:

Regarding the concept of resilience to disease causing mutations:

S. M. Neuner, S. E. Heuer, M. J. Huentelman, K. M. S. O’Connell, C. C. Kaczorowski, Harnessing Genetic Complexity to Enhance Translatability of Alzheimer’s Disease Mouse Models: A Path toward Precision Medicine. *Neuron*. **101**, 399-411.e5 (2019).

R. Chen et al., Analysis of 589,306 genomes identifies individuals resilient to severe Mendelian childhood diseases. *Nat. Biotechnol.***34**, 531–538 (2016).

S. H. Friend, E. E. Schadt, Clues from the resilient. *Science***344**, 970–972 (2014).

Regarding double heterozygous gene interactions:

Ashworth A, Lord CJ, Reis-Filho JS. 2011. Genetic interactions in cancer progression and treatment. *Cell***145**:30–8. doi:10.1016/j.cell.2011.03.020.

Bharucha, N. et al. A Large-Scale Complex Haploinsufficiency-Based Genetic Interaction Screen in *Candida albicans*: Analysis of the RAM Network during Morphogenesis. *PLoS Genet.***7,** e1002058 (2011).

Glazier, V. E. et al. Genetic analysis of the *Candida albicans* biofilm transcription factor network using simple and complex haploinsufficiency. *PLOS Genet.***13,** e1006948 (2017).

Chan DA, Giaccia AJ. 2011. Harnessing synthetic lethal interactions in anticancer drug discovery. *Nat Rev Drug Discov***10**:351–64. doi:10.1038/nrd3374

Sardi M, Gasch AP. 2018. Genetic background effects in quantitative genetics: gene-by-system interactions. *Curr Genet***64**:1173–1176. doi:10.1007/s00294-018-0835-7

Finally, regarding our electron microscopy. We agree that causality is never established. Nonetheless, the EM provides clear, visual demonstration of the synthetic nature of the gene-gene interaction that is undeniable, even for those readers who are not well versed in electrophysiology. The fact that this phenotype is rescued by the heterozygous mutation in CREG (the triple heterozygous mutant combination) is clear evidence of a link to CREG. We clearly state:

“Regardless of the underlying molecular mechanism leading to this EM phenotype and associated physiological deficits (a topic for future study), these data present a striking, visual confirmation of a strong synthetic genetic interaction between *PPP2R5D/+* and the *CHD8/+* heterozygous mutations. Furthermore, these data link the activity of a chromatin remodeling factor, present in the nucleus (*CHD8*), to a profound synaptic defect.”

1) The description of genetic interaction concepts is unnecessarily long and convoluted and must be consolidated.

We have shortened this section of the Introduction.

2) The screen itself is poorly described. Did the authors use the Bloomington Df kit for their screen? If not, specify. A table with the reagents used here should be provided.

A table is now provided. We respectfully disagree with the criticism that the screen is poorly described.

3) The authors chose to compare a small and a large Df in lieu of examining additive effects for gene heterozygosity (Figure 3). These arguments are weak. Instead, a more detailed description of the screen results would be more meaningful and should be included here.For example, how many of the Dfs tested contain known PHP modulators? What was the outcome of the screen for those Dfs? How does that outcome compare with prior single gene analyses?

The reviewer is factually incorrect at several levels. This is perhaps understandable, since this type of analysis is rarely performed on a systematic basis.

a) First, we present every single deficiency in the graph (Figure 3). We provide representative examples of one large and one small deficiency. The reviewer seems to have misread the figure and our text in this regard.

b) Our arguments are not weak. This is the first time that a collection of small deficiencies has been analyzed electrophysiologically. We demonstrate a clear lack of correlation between the number of genes deleted by a deficiency versus effects on synaptic transmission. We state in our text,

“…we assessed whether there was any relationship between the number of genes that were deleted within a given deficiency and the robustness of PHP. One hypothesis is that the additive effects of multiple, heterozygous gene mutations would increase for larger deficiencies and PHP would be increasingly compromised. That was not the case (Figure 3). There was no correlation between the number of genes uncovered by a given deficiency and EPSP amplitude recorded in the presence of PhTx (R^2^ = 0.003; Figure 3A). Thus, impaired PHP cannot be accounted for by a simple additive accumulation of genetic mutations within a given deficiency.”

This is a simple analysis and a straightforward conclusion based on the lack of a statistical correlation.

c) In a formal genetic analysis, if two heterozygous genes do not interact, then nothing can be concluded. This is because it remains unknown how each heterozygous mutation affects the function of the protein within the cell. The fact that we did not identify homeostatic genes residing on the 3^rd^ Chromosome in our screen is, therefore, an un-interpretable event and not reported. With this fact stated, we did report (in the original text) precisely which genes were identified by the screen that were previously demonstrated to interact with *RIMS1*/+. We stated in our original text:

“The screen was empirically validated by the identification, blind to genotype, of deficiencies that uncovered the RIMS1 locus, as well as the Pi3K68D locus (not included in hit list), previously shown to interact as a double heterozygous mutant with *RIMS1/+* (44). Furthermore, the rim binding protein (RBP) locus was not identified as disrupting PHP, consistent with the previously published observation that a rbp/+ mutant does not interact with *RIMS1/+* for PHP (45). However, rbp/+ did interact with *RIMS1/+* for baseline neurotransmitter release as expected based on previously published data^28^ (not shown)”.

d) Our screen is quite unique so a direct comparison to single gene mutant screens or screens based on RNAi-mediated knockdown is not obvious.

e) A table of all deficiencies used in our screen is now provided.

4) When a Df covering a known PHP player did not show the expected interaction, what does this mean for the results of the screen?

In a formal genetic analysis, if two heterozygous genes do not interact, then nothing can be concluded. This is because it remains unknown how each heterozygous mutation affects the function of the protein within the cell. The fact that we did not identify occasional homeostatic genes residing on the 3^rd^ Chromosome is, therefore, an un-interpretable event and not reported (see above).

5) It is somewhat anticlimactic that the PDPK1 and PPP2R5D are not required for acute PHP. However, the authors hint to a possible role in chronic homeostasis, more similar to the mTOR signaling components. This probably has already been examined and should be included here.

We respectfully disagree. This is a major take home message and it is important. We are NOT attempting to identify new core components of homeostatic plasticity. Our screen is designed to identify double heterozygous gene-gene interactions that weaken the robustness of homeostatic plasticity in previously unexplored ways. Why is this important? As highlighted above, ASD gene mutations are often heterozygous loss of function mutations. These heterozygous mutations will likely interact with standing (heterozygous) genetic variation in the human genome, variation that is unique to each individual. Thus, it is important to investigate and try to understand how heterozygous gene-gene interactions affect the phenotype of an organism. *Drosophila* (and *C. elegans*) are two of the model organisms where such an analysis is theoretically possible. This is what we have achieved.

Finally, it is clear from our new summary (Figure 9A), and the data presented in Figure 2 (with the addition of new data in Figure 1—figure supplement 2) that multiple ASD heterozygous mutations sensitize the homeostatic system to failure, but each mutation alone has no effect. The interacting genes need not be homeostatic genes. Rather, heterozygous mutations in newly identified modifiers potentiate the deleterious effects of the ASD gene mutations, causing homeostatic plasticity to fail. This is a fundamentally different way of thinking about how homeostatic plasticity could be affected by gene mutations. We argue that this may have relevance to the variable phenotypic penetrance of neurological and/or psychiatric diseases caused by heterozygous loss of function mutations.

The reviewer is referred to three excellent publications in this regard, concerning an emerging field of disease resilience (below). This interpretation is the subject of a section of our Discussion termed “Genetic context”.

S. M. Neuner, S. E. Heuer, M. J. Huentelman, K. M. S. O’Connell, C. C. Kaczorowski, Harnessing Genetic Complexity to Enhance Translatability of Alzheimer’s Disease Mouse Models: A Path toward Precision Medicine. *Neuron*. **101**, 399-411.e5 (2019).

R. Chen et al., Analysis of 589,306 genomes identifies individuals resilient to severe Mendelian childhood diseases. *Nat. Biotechnol.***34**, 531–538 (2016).

S. H. Friend, E. E. Schadt, Clues from the resilient. *Science (80-. ).***344**, 970–972 (2014).

6) The RNAseq analysis was performed with adult heads due to understandable limitations. But the qPCR validation of CREG up-regulation must be done in third instar larvae, at the stage when PHP is recorded.

This is a perfectly reasonable concern and we not only performed the requested experiment, but added considerably more information to our study. Now, we document *CREG* expression in motoneurons at four developmental time points including the embryo, first instar, second instar, early third instar and late third instar. We then validate the four-fold increase in *CREG* expression in third instar neurons in the double heterozygous condition (as requested). These data substantially improve our analysis and we thank this reviewer for prompting us to include these data.

7) The authors propose that CREG is a homeostatic repressor. This predicts that overexpression of CREG will block PHP. The authors should directly test this prediction which will strengthen their conclusions.

We generated a new *UAS-CREG* transgene and drove expression either in motoneurons or ubiquitously. In both experiments, PHP is blocked. These data are added to Figure 8.

8) Reducing the dose of CREG would be expected to restore PHP in other trans-heterozygote combinations. This should be examined here since a positive result could expand the relevance of this finding beyond an isolated trans-heterozygous case presented here.

We respectfully disagree. This prediction assumes that the same mechanism is induced in all double heterozygous combinations. We now report that three additional RNAseq experiments fail to show up-regulation of CREG in other double heterozygous mutant combinations. When these data are considered alongside of the new EM data that we present (also showing that the EM phenotype is not conserved across all double heterozygous genotypes), it can be concluded that the CREG result is not universal, but it is none-the-less remarkable. Regardless of the generality, our data emphasize a new way to consider how homeostatic plasticity can fail in the context of human diseases that have a genetic origin.

9) Another elegant solution to link the disconnected parts of this study would be to overexpress CREG in some of the single heterozygotes (RIMS1, CHD8, CHD2, WDFY3, ASH1L – or PDPK1, PPP2R5D) and block PHP.

We demonstrate that CREG overexpression blocks PHP. This effect occurs irrespective of any genetic background. So, the suggested experiment does not make logical sense.

10) Using qPCR to search for increased levels of CREG in (third instar) mutants lacking PHP would be an easy, fast way to implicate CREG in other known PHP pathways and will tremendously benefit the author's arguments.

We disagree with this assertion. There is no reason to assume that CREG is involved in any general mechanism given that it is invoked only at the intersection of two heterozygous gene mutations, neither of which causes CREG over-expression alone. The reviewer is intent upon linking our data to the existing knowledge of homeostatic plasticity mechanism. One reason our work is novel is that we have may have uncovered a fundamentally different way to affect the robustness of homeostatic plasticity. This is emphasized by the fact that the modifiers are not, strictly, essential for PHP when tested as homozygous mutations.

Reviewer #3:Genc et al. present in this study findings arguing that five seemingly unrelated autism associated genes are linked to presynaptic homeostatic plasticity (PHP) and synaptic transmission. They propose through genetic interaction studies in *Drosophila* that phenotypic enhancers could influence the robustness and phenotypic penetrance of these autism associated gene variants. Although several aspects of this study appear to support their conclusion, the reliability and reproducibility of their findings is hampered by conceptual and technical concerns. Addressing these concerns would help to strengthen the study.1) The authors propose that five selected autism genes converge on PHP, but they only examine the genetic interaction between two pairs (RIMS1 and CHD8) and (RIMS1 and ASH1L). It is unclear why the authors do not complete the pairwise interaction studies for the remaining two genes. No rationale is provided to explain this incomplete analysis.

This is a fair point. We now analyze all potential interactions. Three of the four show a clear block (to our surprise) and one (*WDFY3*/+) does not. These data have been added to the manuscript. This greatly strengthens our paper and we thank the reviewer for prompting us to add these data to the study.

2) Due to the possibility of second site mutations contributing to the genetic interaction findings, it is important to assess other loss of function alleles of RIMS1, CHD8, and ASH1L in the genetic interaction studies. As well as for the remaining two selected autism genes. Furthermore, can these findings be ameliorated with genomic rescue of RIMS1, CHD8, ASH1L, etc in the pairwise genetic interaction studies?

The reviewer has seemingly requested many hundreds of experiments. In the context of our work, we argue that this is not justified. First, the *RIMS1* and *CHD8* and *WDFY3* have been previously published and validated at the NMJ. The mutation in *CHD2* was validated in our study by showing rescue experiments in which a *CHD2* gene translocation was placed in the double heterozygous mutant background, demonstrating rescue. Second, our screen essentially is a systematic test for second site (loss of function) interactions on the third chromosome and an interaction is, indeed, rare. Finally, we validate the major finding of our paper by performing several experiments including A) *UAS-CREG* over-expression (new data) replicates the double heterozygous block of PHP and B) by analysis of a triple heterozygous animal (both electrophysiologically and at the EM level) demonstrating that all phenotypes are rescued by simply reducing the dosage of the *CREG* gene.

3) What is the rationale for choosing only five of the 20 deficiencies that showed interactions with RIMS1 to test their interactions with other gene? The authors need to provide a rationale for this selection. Otherwise it raises the question as to whether the remaining deficiencies were tested but did not show as strong of an interaction effect.

We can only test a finite number of interactions because they must be performed by hand. We chose five deficiencies – simple as that. A similar question could be asked about virtually every scientific study – why choose to study a given gene? Here is another example: The laboratory of Dr. Dan Feldman recently published a nice piece of work examining three ASD gene mutations in mice (published in *Neuron*). Why did he choose those three? The point is that one must make choices. Our choices were random.

4) Critical resource and technical information is lacking, which make it difficult to interpret or reproduce the study in the future. In particular, fly stocks (origin, type of mutation, specific breakpoints, stock center, etc) was not reported. What were the specific deficiency lines – including the "20 small deficiencies that cause PHP to fail"?

A new table is provided listing all deficiencies used. Each stock number can be input to the Bloomington website and all relevant information ascertained. We would also like to point out something that we consider obvious, but might be necessary given the antagonistic tone of this reviewer. When mapping individual gene contributions within a deficiency, numerous sub-deficiencies are used followed by testing of individual genes. This represents many, repetitive, examples of verification (positive and negative) before gene identification is achieved. This is standard in the field.

5) It is often unclear in the manuscript and figures if the sample size "n" is referring to animals, neurons, synapses, vesicles, etc. The sample size is also very low (n of only three) for some of the studies, raising the concern as to whether the studies were adequately powered to detect a statistical difference. For example, "we performed multiple (n=3-15) intracellular recordings".

a) In the majority of figures, every data point is plotted. For the EM, we have clarified sample sizes. In tables, as in figures, the sample size is the recording number, given that the averages are for recordings, it cannot be otherwise.

b) The comment regarding sample size is unfair. The reviewer quotes a statement that specifically refers to our screen – yet the reviewer implies that this reflects upon other data in our paper (which is factually incorrect). We stated in our original paper:

“The screen that we performed is diagrammed in Figure 2A. We took advantage of a collection of small chromosomal deficiencies (5-50 genes per deficiency, each with known chromosomal breakpoints; listed in Supplemental Table 1) that tile the 3^rd^ chromosome, uncovering approximately 6000 genes in total. For every double heterozygous combination of *RIMS1/+* with a heterozygous deficiency, we performed multiple (n=3-15) intracellular recordings, quantifying mEPSP amplitude, EPSP amplitude, quantal content (EPSP/mEPSP), resting membrane potential and input resistance.”

A genetic screen, such as ours, is designed to identify “hits” as efficiently as possible. In our genetic screen, as is common practice, we do not make any claims regarding the statistical validity of any individual observation. The only purpose of a screen is to identify “hits”. We establish well-justified parameters to select “hits”. This is done in a conservative fashion in order to protect the experimenter from performing un-necessary follow-up work. Importantly, every “hit” is re-validated. We are now more explicit, stating, “To achieve a final list of 20 hits, each potential hit was re-validated in a second set of experiments, increasing sample sizes (generally 7-16 NMJ). During the process of revalidation, we rule out approximately one third of the potential hits selected from the screen.”

Furthermore, the reviewer is referred to the fact that multiple “hits” from our screen were re-validated, multiple times throughout our manuscript. First, the hits were identified. Second, they were revalidated as a hit. Third, five of the deficiencies were incorporated into a matrix of multiple gene-gene interactions, a further re-validation. Finally, two deficiencies were resolved to individual genes, requiring multiple rounds of re-validation. And, the single genes precisely recapitulate the results of the deficiencies in which they originally resided. Surely, at this point, the extensive nature of revalidation, with statistically relevant sample sizes at every stage, should inspire statistical confidence.

How did we determine adequate sample sizes? We performed a power analysis to estimate necessary sample size. We have an effect size of 150-200% with a low standard deviation. Estimation of sample size suggests a minimal sample size of three to four, which is obviously very small. In our primary screen, when efficiency is emphasized, we use a sample size of 3-15, as reported. None of the data from the screen are compared statistically – the use of sample sizes here is simply for transparent reporting. In our study, when statistical comparisons are made, sample sizes were never less than n=7. In instances where effect size was on the smaller end of the range, sample sizes are generally larger to ensure confidence.

In our study, we analyze more than 40 genotypes electrophysiologically, under two conditions (with and without PhTX), representing more than 80 experimental conditions – not including our screen. In only two experiments out of more than 80 experimental conditions are sample sizes equal to 4 (in the original submission, there were four instances, and new data have been incorporated). These instances were controls that adhered to high effect size and low variance. In every other experiment (>80 conditions) sample size was never less than 7, generally in excess of 10 and often greatly exceeding these sizes. All the data are transparently presented in figures with individual data points and in tables with sample sizes. In every instance, sample sizes were sufficiently powered and all statistical tests and values reported.

6) The authors argue that CREG is a homeostatic repressor that blocks PHP and regulates synaptic ultrastructure (Figure 8). It would be important to show the phenotype of the CREG single mutant. What is the ultrastructural phenotype of the CREG single mutant?

We have performed the requested experiments. The CREG mutant that abolishes gene expression was analyzed. We state in the text:

“Finally, we assessed the consequences of a the heterozygous and homozygous loss of function mutations on baseline neurotransmission and PHP. The CREG^M1^ allele abolished expression (see above). Neither CREG^M1^/+ heterozygous animals (n=8) nor the CREG^M1^ homozygous animals (n=8) and any effect on baseline transmission (quantal content; p>0.5) or the expression of PHP (CREG^M1^/+ = 143% PHP, p<0.05 compared to wild type; CREG^M1^ = 155% PHP, p<0.01 compared to wild type). Thus, loss of CREG is without effect on neuromuscular transmission or PHP. Taken together, our data are consistent with the conclusion that CREG is a homeostatic repressor, one of very few identified to date (Spring et al., 2016).”

We have not performed the ultrastructure of the CREG mutant since it has no effect on synaptic transmission, nor does it effect homeostatic plasticity.

7) EM studies are performed in tissue fixed 1-5 seconds after high frequency stimulation with a sample size as low as only 2 animals. How does the difference between 1 to 5 seconds post-high frequency stimulation affect the findings?

The only way to answer this question is to invoke the collaboration of either Eric Jorgensen or Shigeki Watanabe, who perform modern freeze-slam biology with highly specialized equipment from Leica. Indeed, we have attempted this experiment in collaboration with the Watanabe laboratory but, to date, the technology cannot be applied to third instar larvae due to tissue thickness, which prevents uniform freezing and tissue preservation. We have even attempted this in first instars, but tissue freezing remains a confound. We would like to point out that our manuscript represents an unusually thorough EM analysis by the standards of the field (noting, however, the exceptional and unique work of our *C. elegans* colleagues).

8) The rationale for selecting RIMS1, CHD8, CHD2, WDFY3, and ASH1L is still unclear. The authors state they were selected as they were "category 1 or 2 ASD-associated genes based on SFARI Gene". However these two categories encompass many more than these five genes. Were these the only genes with fly orthologs and available resources? Were there other reasons for choosing these genes?

We thought that we were clear, in this regard, in the text.

Here is the full history. This project began more than five years ago when Prof. Matt State moved to UCSF from Yale University. He supplied us with the identities of many of the most agreed upon de novo mutations that confer risk for ASD at that time. At that time, the list was much shorter than it is now. Nonetheless, the five genes that we decided to study remain category 1 genes, and remain well-acknowledged, high confidence ASD risk factors. It appears that we received good advice and made good choices in retrospect. It would be impossible for us to analyze more than five. No other study has engaged in such a phenotypic analysis. The only study to analyze multiple mutations in ASD, to our knowledge, is the recent work of Dr. Dan Feldman, which was an extraordinary assessment of three mutations in mice (published recently in *Neuron*). We do not known why he chose those three mutations to work on.

We stated in the original text a very clear set of rational for choosing these genes, to which we have added expression analysis. We stated,

“All five of these genes are considered high confidence “category 1” ASD-associated genes based on SFARI Gene (Simons Simplex Collection, 2018)^34^. All five of these genes have clear *Drosophila* orthologs. Further, we demonstrate that all five genes are expressed in *Drosophila* third instar motoneurons based on a Patch-Seq analysis of gene expression (Figure 1—figure supplement 1). The five ASD gene orthologs were also chosen to reflect a broad range of biological activities that are associated with the numerous ASD-associated genes identified to date. The RIMS1 gene is a synaptic scaffolding protein that localizes to and organizes sites of neurotransmitter release, termed active zones. The *CHD8* and *CHD2* genes encode chromatin remodeling factors that localize to the cell nucleus. *WDFY3* encodes a phosphatidylinositol 3-phosphate-binding protein and regulator of autophagy and intracellular signaling. *ASH1L* encodes a member of the trithorax group of transcriptional activators and is found in the cell nucleus. A survey of biochemical and genetic interaction networks in *Drosophila* demonstrates no known interactions among these five genes (Flybase). In humans, there appear to be no known direct biochemical interactions among these genes. Yet, heterozygous LOF mutations in each of these genes are associated with risk for ASD in humans.”

9) The authors need to be careful not to overstate or misstate their findings. Particularly as the premise of the paper hinges upon 5 selected genes associated with autism, and far more than 5 genes are implicated in autism. Providing a better rationale for selecting these 5 genes would help frame their findings better.

Thank you for the criticism. However, our study is one of very few to begin to assess multiple ASD genes in parallel and to explore genetic interactions. We have revisited the text in response to this reviewer, but we argue that highlighting the novelty of our approach is both reasonable and appropriate at this time.

[Editors’ note: what follows is the authors’ response to the second round of review.]

Revisions required for this paper:1) Clarify or modify broad statements about lack of information or claims of "novelty", that falters in the context of existing information in the literature. The study will be strengthened by building connections with existing knowledge, which will minimize the impression that these findings are "synthetic" without anchoring in known phenomena.

Please see individual responses to each specific criticism. In all cases, we quote the changes we make to the text directly in in this document in response to each reviewer query.

a) In discussing the random selection of RIMS1, CHD8, CHD2, WDFY3, and ASH1L the authors state that there are no known biochemical and genetic interactions among these five genes in *Drosophila*, and no known direct biochemical interactions in humans. However, it is curious that the authors fail to also state the potential for interactions between these genes by incorporating many findings in mouse models indicating a potential convergence between these genes on neurodevelopment and function. Presenting this at the outset will further strengthen the study by utilizing prior literature to support the probability of identifying disease relevant genetic interactions.

In many respects, this comment reflects the question that drove us to initiate our study back in 2013. At that time, as now, a major question in the field of ASD research is whether there are genetic or molecular commonalities that give rise to common phenotypes in human. The reviewer would like us to highlight the “potential for interactions between these genes” (the ASD genes). In other words, the reviewer would like us to continue the type of speculation that has been ongoing in the literature for a number of years. We entered this line of research to avoid such speculation and perform forward genetic gene discovery. So, we are hesitant to initiate our study with such speculation. There are many theories, some more robustly defended than others. For example, there is a theory that causal gene mutations in ASD may commonly disrupt the balance of excitation and inhibition, an idea first proposed by Michael Merznich and John Rubenstein (Rubenstein and Merznich, 2003) with recent experimental support (Antoine et al., 2019). There has been speculation that chromatin remodeling factors control the expression of “synaptic” genes (Sullivan et al., 2019). Homeostatic hypotheses have been put forward, including a nice review by T. Bourgeron (Bourgeron, 2015).

Rather than reviewing these many possible theories, or emphasizing phenotypic similarities among genes that may or may not have and biochemical or genetic interaction, we frame our paper from a well-established perspective in the field of genetics and evolutionary biology. At the start of the Introduction, we frame our paper by stating, “Advances in whole genome sequencing and genome-wide association studies have dramatically expanded our understanding of the genetic architecture of ASD. In particular, the identification of rare de novo mutations that confer high risk for ASD has generated new molecular insight (De Rubeis et al., 2014; Iossifov et al., 2014; Sanders et al., 2015). Yet, even in cases where a rare de novo mutation confers risk for ASD, additional processes are likely to contribute to the ASD phenotype including the engagement of adaptive physiological mechanisms that respond to the presence of an ASD risk associated gene mutation (Gaugler et al., 2014; Gibson, 2009; Hartman et al., 2001; Hou et al., 2019; Kitano, 2007; Plomp et al., 1992; Sackton and Hartl, 2016; Sardi and Gasch, 2018).” This is an idea that is also a major thesis of an excellent book on the evolution of complex systems titled, “The Plausibility of Life: Resolving Darwin’s Dilemma” by M.W. Kirschner and J.C. Gerhart (2005).

Out of respect for this reviewer’s comment and the reviewer’s apparent wish to acknowledge that there has been considerable phenotypic work on these genes, we now add a sentence to the first paragraph of our Results that refers the reviewer to a newly added table. We state, “A supplemental table (Supplementary file 1 table S1) includes known disease associations for each of these five human genes, and links to web-based genetic and genomic resources.” This table lists each of the five ASD genes and includes a web-link to the SFARI GENE website for each gene. This site curates the known human mutations in each gene, disease associations for each gene, and evaluates the robustness of the existing literature. It is an excellent resource. Further, for each of the five ASD-associated genes, we list the known disease associations. We hope that the inclusion of this table will satisfy the desire for additional information.

b) By extension, the known disease associations for these genes should be presented and referenced, as well as if they are all impacting similar neurologic functions in humans. All five genes have known disease associations: OMIM: RIMS1 (#603649, Cone-rod dystrophy type 7), CHD8 (#615032, Susceptibility to autism type 18), CHD2 (#615369, Epileptic encephalopathy childhood onset), WDFY3 (#617520, Primary microcephaly type 18), and ASH1L (#617796, autosomal dominant mental retardation type 52).

Please see the addition of a new supplemental table (Supplementary file 1 table S1), as described in the answer to point (a) above. Within the SFARI GENE listing are all of the OMIM information requested by this reviewer.

c) Heterozygous loss of function mutations in CHD8, CHD2, WDFY3, and ASH1L all cause neurodevelopmental disorders, including autism, epilepsy, and microcephaly. In the context of the human data, the authors' finding that PHP in the single heterozygous flies is normal suggests that PHP may not be the most important mechanism driving autism phenotypes in neurodevelopmental disorders. In contrast, although pathogenic variants in RIMS1 have only been reported with a photoreceptor phenotype in humans, there is a recent publication by Peter et al., 2019 (PMID 30949922) presenting evidence in humans for pleiotropic and complex effects involving RIMS1 in sporadic autism spectrum disorder. Therefore, the authors can consider strengthening the conceptual impact of their study from the perspective that variants in RIMS1 may increase the susceptibility to autism spectrum disorder but require other polygenic factors to manifest the disorder.

We thank the reviewer for pointing us to this newer paper on *RIMS1*, one that we had not seen. We now reference this paper in the first paragraph of the Introduction. This is exciting work.

We would also like to make a point of clarification. The reviewer states, “Heterozygous loss of function mutations in *CHD8*, *CHD2*, *WDFY3*, and *ASH1L* all cause neurodevelopmental disorders, including autism, epilepsy, and microcephaly. In the context of the human data, the authors' finding that PHP in the single heterozygous flies is normal suggests that PHP may not be the most important mechanism driving autism phenotypes in neurodevelopmental disorders.”

Although the individual heterozygous mutations do not block PHP, we demonstrate that many of these gene mutations sensitize PHP to fail. Based on this finding, and based on the rest of our analyses, we argue that loss of PHP may contribute to the phenotypic severity of ASD, not the cause of ASD. We consider this distinction between causality and phenotypic severity to be a very important concept, and it is the framework for the entire study (see above). We also state this in our Discussion:

”It is well established that genetic context can profoundly influence the phenotypic severity of disease-causing gene mutations. For example, in mice, it has been shown that genetic context (strain background) influences phenotypic penetrance in an Alzheimer’s disease model (Neuner et al., 2019). In humans, systematic screening of the phenotypically normal population has identified individuals that are resistant to the effects of well-established, debilitating disease causing mutations, an effect termed “resilience” that is attributed to the effects of genetic context (Chen et al., 2016; Friend and Schadt, 2014). It seems plausible that the common phenotypic enhancers, identified in our genetic screen, could represent a mechanism by which genetic context influences the phenotypic penetrance of ASD-associated gene mutations.”

This is a framework that is not dissimilar to the argument of genetic complexity made in a number of recent studies, including that of Peter et al., 2019. But, we take the idea of genetic complexity, which is quite general, and provide evidence for how genetic complexity could manifest as neurophysiological defects through the failure of homeostatic mechanisms.

d) The authors state that the "five ASD gene orthologs do not share any known common biological activity. Therefore, the identification of common genetic modifiers is completely unexpected." This is a confusing statement to make given that CHD8 and CHD2 are in the same subfamily of chromodomain helicase DNA-binding proteins, indicating a common biological activity, and ASH1L localizes to the chromatin. Based on these common functions and subcellular localization it is not entirely unexpected to find common genetic modifiers. These statements should be revised. Noting the potential for common biological activity will not detract from the potential impact of the authors' findings of common genetic modifiers through their screening approach.

We would like to point out that the function of each gene was clearly stated in our original text. We state, “The *RIMS1* gene is a synaptic scaffolding protein that localizes to and organizes sites of neurotransmitter release, termed active zones. The CHD8 and CHD2 genes encode chromatin remodeling factors that localize to the cell nucleus. *WDFY3* encodes a phosphatidylinositol 3-phosphate-binding protein and regulator of autophagy and intracellular signaling. ASH1L encodes a member of the trithorax group of transcriptional activators and is found in the cell nucleus.

We also are quite specific about this statement and topic in our Introduction. We state, “A survey of biochemical and genetic interaction networks in *Drosophila* demonstrates no known interactions among these five genes (Flybase). In humans, there appear to be no known direct biochemical interactions among these genes. Yet, heterozygous LOF mutations in each of these genes are associated with risk for ASD in humans.”

Out of respect for the reviewer’s concern, we have removed that sentence. We agree that the sentence is somewhat redundant and it is less precise that the text that preceded it in our Introduction.

e) The authors state that "there is no clear connection, biochemically or genetically, to the role of PDPK1 or PPP2R5D in the nervous system." The authors need to clarify this statement as there are rodent models demonstrating a genetic role for PDPK1 and PPP2RD5 in the nervous system. There is altered brain development in PDPK1 KO mice (Lawlor et al., EMBO J, PMID 12110585) and abnormal tau pathology in the brains of PPP2RD5 KO mice (Louis et al., PNAS, PMID 21482799). Based on the literature there is clear connection for a role of PDPK1 and PPP2R5D in the nervous system.

With all due respect, the reviewer is confused on this point. This comment is taken out of context and the reviewer infers something that we do not state. We do not make any claim regarding a connection between *PDPK1* and *PPP2R5D*. The sentence refers back to the two other genes that were identified in our RNAseq analyses. To prove this point, we simply copy more of the surrounding paragraph: We state, “… there are only two genes that are commonly down-regulated in all four ASD mutants (FBgn0027578 [Nepl21] and FBgn0037166 [CG11426]) (Figure 7C). FBgn0027578 encodes a metalloprotease of the Neprilysin family, with homology to endothelin converting enzyme 1 in human, of unknown function in the nervous system. FBgn0037166 encodes phosphatidic acid phosphatase type 2, which is expressed in the *Drosophila* nervous system, but of unknown function. There is no obvious means to connect the down regulation of these two genes to impaired homeostatic signaling, although future experiments will explore these genes in greater depth. Furthermore, there is no clear connection, biochemically or genetically, to the role of *PDPK1* or *PPP2R5D* in the nervous system.”

To address any possible confusion that some other reviewer might have, we have revised this sentence. It now reads, “Furthermore, there is no clear connection between FBgn0027578 or FBgn0037166 and the roles of either *PDK1* or *PPP2R5D* in the nervous system.”

f) The authors propose that upregulation of CREG in PPP2R5D/+;CHD8/+ underlies the failure of PHP, which is further supported by the complete block of PHP when CREG is overexpressed ubiquitously or in motor neurons. They identified two CREG mutant alleles with differential effects on normalizing CREG levels in Figure 8, panel D (m2 restored the double mutant to wildtype, but not m1) suggesting that m2 is possibly a stronger loss of function allele. But surprisingly, m2 seems to have a reduced rescue effect compared to m1 (Figure 8, panel F). Is there a difference in CREG expression level between these two alleles? The authors only report the m1 transposon completely abolishes CREG expression and do not show the data. They do not report the effect of the m2 transposon. The effect on CREG expression from both alleles should be examined and shown as this may indicate a dosage sensitivity for CREG.

We acknowledge that additional information would be important. In our prior submission, we stated that RNA expression was abolished in the M1 allele. We have now expanded upon this statement to include precise values, variance and the number of biological and technical replicates. We now state, “The CREG^M1^ transposon completely abolishes CREG expression and a heterozygous CREG^M1^/+ mutant reduces CREG expression (CREG^M1^ = zero expression compared to wild type, 3 biological and 3 technical replicates; CREG^M1^/+ = 51.5 ± 3.05% wild type expression, 3 biological and 3 technical replicates). Next, we generated a triple heterozygous mutant combination (*CHD8+; PPP2R5D/+, CREG^M1^/+*) and find that the CREG^M1^/+ allele attenuates the up-regulation of CREG gene transcript in the triple heterozygous mutant background, a suppression effect of approximately 50%, as predicted (Figure 8D). Then, we repeated this analysis with the CREG^M2^ allele. This allele has a minor effect on baseline CREG expression (73.2±2.7% wild type expression, 3 biological and 3 technical replicates). However, we discovered that this transposon insertion caused a complete block of CREG up-regulation in the triple heterozygous mutant combination, suggesting that this transposon insertion, residing in 3’ UTR, may disrupt a transcription regulatory motif (Figure 8D).”

g) A rationale for selecting only 5 of the 20 deficiencies showing interactions with RIMS1 is still lacking. Were they selected based on the type of genes in the deficiencies (brain specific in mammals), known or unknown function of the genes, prior implications in homeostatic plasticity, availability of reagents, strength of the interaction effect, etc?

This question was asked in the first round of review. In response, we stated: “We are human. We can only test a finite number of interactions because they must be performed by hand. We chose five deficiencies – simple as that.” Perhaps we can add more context with proof regarding the randomness of our selection. We did not select the 5 deficiencies because they were the “best” hits. Note that Df(3)24410 is among the weaker hits identified in our screen (see new Supplementary file 1 table S3-S5). We did not select these five deficiencies because they contain interesting genes. Given the number of genes within each deficiency, it is impossible to guess. Finally, there is some bias against the very largest Df’s, which seems appropriate for this experiment. But, at the same time, we did not actively select for the smallest of the Df’s. So, once again, we randomly selected 5 deficiencies and performed the analysis. As further context, when we began this experiment, we did not expect to observe a high degree of common enhancement. This finding was serendipitous and became a focus of the work from that point forward.

2) Technical concernsa) There appears to be incomplete presentation of some datasets. The authors report that "altered NMJ growth was not commonly observed in the majority of genetic interactions tested (Figure 5—figure supplement 1)", implying that multiple pairwise interactions were examined. In fact, this is not the case. Figure 5—figure supplement 1 shows NMJ morphology data for wildtype, single heterozygotes for PDPK, PPP2R5D, CHD8, CHD2, ASH1, WDFY3. The only genetic interactions shown were for CHD8 with PDPK1 and CHD8 with PPP2R5D. The data actual shows an increase in bouton number for the CHD8/+;PPP2R5D/+ genotype compared to wildtype and PPP2R5D/+ alone, it did not reach statistical significance compared to CHD8/+ alone but perhaps this is due to the sample size? The data for the other pairwise genetic interactions should also be included in order for the authors to make the statement that altered NMJ growth was not commonly observed.

We thank the reviewer for highlighting this point. We agree that the statement highlighted by the reviewer is not a correct representation of the information in the supplemental figure. This information has not changed since the original submission to a different journal more than 24 months ago. We are pleased to change the text here, soften the conclusion and be more specific about the exact data presented. We now state, “We asked whether neuronal morphology was substantially altered in the heterozygous ASD-associated gene mutations and in select double heterozygous genetic interactions (Figure 5—figure supplement 1). We do find evidence that the heterozygous *CHD8/+* mutation predisposes the NMJ to modest overgrowth, consistent with CHD8 influencing brain development in other systems (Gompers et al., 2017). But, but this effect does not become more severe when combined with either *PDPK1* or *PPP2R5D* mutation as double heterozygotes. Thus, we conclude that altered synaptic growth is not highly correlated with the block of PHP in these double heterozygous combinations.”

b) Sample size still remains unclear for Figure 5—figure supplement 2, Figure 5—figure supplement 3, Figure 8F, Figure 8K. Please clarify the sample size so that the interpretation and reproducibility of the data is improved.

We thank the reviewer for pointing out this omission. The sample sizes have been listed in the figure legend for all genotypes in Figure 5—figure supplement 2, Figure 5—figure supplement 3, 8F and 8K.

c) In Figure 8, what does the sample size "n" in panels H and I represent? The authors note in the figure legend that "individual data points shown for indicated genotypes". Clarify if the individual data points represent individual vesicles, average vesicle size per synapse, or average vesicle size per animal. How many synapses were quantified per animal? How many animals per genotype?

We thank the reviewer for pointing out this omission. The sample sizes have been clarified the figure legend of Figure 8.

d) Similarly, in Figure 6, what is the sample size representing? Individual vesicles, average size of vesicles per boutons, per animal? How many animals and synapses per animal were assessed?

We thank the reviewer for pointing out this omission. The sample sizes have been clarified in the figure legend.

e) It is important to provide all the necessary information about reagents and resources used for rigor and reproducibility. The authors now include a table S1 showing all of the deficiency stocks that were used in the study. But it remains unclear which of the listed 168 deficiency stocks were the "20 small deficiencies that cause PHP to fail when combined with RIMS1/+". The authors provide a schematic showing the distribution of these deficiencies in Figure 2C, but it is important to also denote in Table S1 the stock ID number corresponding to these 20 small deficiencies. This should be very straightforward to indicate in table S1.

We are pleased to provide a set of new tables at this stage of the review process. We now present a total of seven supplemental tables. Included are the follow new data: Supplementary file 1 tables S3-S5 list all of the deficiencies that were tested, inclusive of the electrophysiological data acquired during the screen and sample sizes. We also list each of the deficiencies that were selected as “hits” in our screen. We list the stock number and full genotype. Please note: The values presented in graphical format in Figure 2 from our screen are the values that were initially recorded during our high-throughput screening mode and these values are listed in Supplementary file 1 table S2. In Supplementary file 1 table S3, we report values only for the “hits” and these values are different because they include larger sample sizes that were achieved when we verified each hit. Thus, Supplementary file 1 table S3 includes additional information, inclusive of our verification efforts. Finally, (Supplementary file 1 table S5), we list all of the genes that are uncovered by each of the deficiencies selected in our screen, according to information provided in Flybase. This should facilitate reader use of the information. These tables are cited in the Results section. We state, “(see Supplementary file 1 tables S3-S5 for further detailed information on the screen results).” We believe that we have exceeded the request for new information in this regard. We thank the reviewers for prompting us to provide these additional data.

f) The authors assessed the consequences of CREG heterozygous and homozygous loss of function mutations on baseline neurotransmission and PHP and describe the findings in the text. This is an important control data. For comparison to the other genotypes it would be helpful to also include the data graphically in Figure 8, panel F.

We have added data for homeostatic plasticity in the CregM1/+ and CregM1 to Figure 8F. We think that this now represents a full data set, as is commonly presented in homeostatic plasticity papers in the field. We also report the precise values for baseline neurotransmitter release in the text. We now state, “Finally, we assessed the consequences of the heterozygous and homozygous loss of function mutations on baseline neurotransmission and PHP. The CREG^M1^ allele abolished expression (see above) and is the focus of these analyses. Neither the heterozygous nor homozygous animals affected expression of PHP (Figure 8F). The CREG^M1^/+ heterozygous animals had no effect on baseline transmission compared to wild type (wild-type QC = 40.8±2.2 n=10; CREG^M1^/+ QC = 38.6 (±3.0) n=8; Student’s t-test; p>0.5). The CREG^M1^ homozygous allele decreased baseline transmission by ~18% (CREG^M1^ QC = 33.5±3.0 n=8; p=0.02). Clearly, neither baseline release nor PHP are potentiated, demonstrating that the rescue of PHP in the triple heterozygous mutant condition cannot be considered an additive effect of the heterozygous CREG^M1^ mutation. Taken together, our data are consistent with the conclusion that CREG is a homeostatic repressor, one of very few identified to date (Spring et al., 2016). “

g) You stated: "First, we present every single deficiency in the graph (Figure 3). We provide representative examples of one large and one small deficiency. The reviewer seems to have misread the figure and our text in this regard." The reviewers did not find the data, legend or text presenting the interacting deficiencies. Figure 3 has no such graph. They are wondering whether you refer to the drawing in Figure 2C. Do the authors imply that this is the summary result that presents every single deficiency? Are we missing something? Moreover, the stock numbers for the Bloomington deficiencies provided in the table do not match the ones described in the text.

The original reviewer comments stated, “The authors chose to compare a small and a large Df in lieu of examining additive effects for gene heterozygosity (Figure 3).” We interpreted this statement to mean that the reviewers had missed the fact that each data point in Figure 3A represents averaged data for every individual deficiency tested in our screen. In Figure 3A, we plot quantal content versus the number of genes deleted within each deficiency and note that there is no correlation. The legend of Figure 3A stated, “Scatter plot showing the number of genes deleted (y axis) versus quantal content (x axis) in the presence of PhTx for all deficiencies tested.”

In our response to reviewer criticism in the first round of review, we stated, “First, we present every single deficiency in the graph (Figure 3A – the entire figure is copied here for clarity). We provide representative examples of one large and one small deficiency. The reviewer seems to have misread the figure and our text in this regard.” We are sorry that there seems to remain some confusion. Currently, the reviewers state, “Figure 3 has no such graph”. Perhaps the graph was somehow deleted in the figure the reviewer received, or there remains some confusion about the precise graph that we are each referring to?

Finally, we re-checked the deficiency call-outs. The representative deficiencies (Figure 3B) are not “hits” from our screen (we never suggested that they were) and, therefore, were not represented in the last submission. The representative examples were chosen, simply, as representative deficiencies that had either many or few genes deleted, with remarkably similar quantal contents. We now supply, upon reviewer request, information for all of the deficiencies in our screen.

Information is needed for why data sets are not shown (ie CREG expression level in m1 allele), clarity on what the sample size represents in multiple datasets, details as to how many synapses, boutons, animals are included in the NMJ and EM datasets.

Please see above.